# TRPV1 channels are critical brain inflammation detectors and neuropathic pain biomarkers in mice

Maria Cristina Marrone[1,2,*], Annunziato Morabito[1,3,*], Michela Giustizieri[1,2], Valerio Chiurchiù[2,4], Alessandro Leuti[2], Marzia Mattioli[1,2], Sara Marinelli[1,5], Loredana Riganti[6], Marta Lombardi[7], Emanuele Murana[8], Antonio Totaro[2], Daniele Piomelli[9], Davide Ragozzino[8,10], Sergio Oddi[2,11], Mauro Maccarrone[2,4], Claudia Verderio[5,6] & Silvia Marinelli[1,2]

The capsaicin receptor TRPV1 has been widely characterized in the sensory system as a key component of pain and inflammation. A large amount of evidence shows that TRPV1 is also functional in the brain although its role is still debated. Here we report that TRPV1 is highly expressed in microglial cells rather than neurons of the anterior cingulate cortex and other brain areas. We found that stimulation of microglial TRPV1 controls cortical microglia activation per se and indirectly enhances glutamatergic transmission in neurons by promoting extracellular microglial microvesicles shedding. Conversely, in the cortex of mice suffering from neuropathic pain, TRPV1 is also present in neurons affecting their intrinsic electrical properties and synaptic strength. Altogether, these findings identify brain TRPV1 as potential detector of harmful stimuli and a key player of microglia to neuron communication.

[1] European Brain Research Institute-Fondazione Rita Levi Montalcini, 00143 Rome, Italy. [2] IRCCS Fondazione Santa Lucia, 00143 Rome, Italy. [3] Department of Physiology and Pharmacology Vittorio Erspamer, Sapienza University of Rome, Rome 00185, Italy. [4] Department of Medicine, Campus Bio-Medico University of Rome, Rome 00128, Italy. [5] CNR-National Research Council, Institute of Cell Biology and Neurobiology, 00143 Rome, Italy. [6] CNR Institute of Neuroscience, via Vanvitelli 32, 20129 Milan, Italy. [7] IRCCS Humanitas, via Manzoni 56, 20089 Rozzano, Italy. [8] Department of Physiology and Pharmacology, Istituto Pasteur-Fondazione Cenci Bolognetti, Sapienza University of Rome, Rome 00185, Italy. [9] Drug Discovery and Development, Fondazione Istituto Italiano di Tecnologia, Via Morego 30, I-16163 Genova, Italy. [10] IRCCS Neuromed, Via Atinese, Pozzilli 86077, Italy. [11] Faculty of Veterinary Medicine, University of Teramo, Teramo 64100, Italy. * These authors contributed equally to this work. Correspondence and requests for materials should be addressed to S.M. (email: s.marinelli@ebri.it).

The transient receptor potential vanilloid type 1 (TRPV1), also known as vanilloid receptor, is a non-selective cationic channel that is exogenously activated by capsaicin, resiniferatoxin and some venom toxin[1–3]. Endogenously, TRPV1 is opened by high temperatures ($>43\,°C$), acid pH, anandamide (N-arachidonoylethanolamine), 2-arachidonoylglycerol, N-arachidonoyl dopamine, N-oleoyldopamine, ATP, lipoxygenase products and monoacylglycerols[4–6]. TRPV1 behaves as a molecular integrator of chemical and noxious heat stimuli in the peripheral[7] and central nervous system, determining nociceptive responses[8,9]. For instance, TRPV1 behaves as thermosensory transducer in central vasopressin neurons of the hypothalamus and modulates dopaminergic mesolimbic system upon noxious/stressor stimuli[10].

With regards to its role in the central nervous system, it is widely accepted that TRPV1 activation mostly modulates synaptic transmission by a presynaptic mechanism[11–14]. Neurobehavioral studies indicate that TRPV1 is critically involved in neurological and psychiatric disorders such as epilepsy, anxiety and depression as well as drug-addiction disorders[15]. Concerning TRPV1 expression, although still controversial[16], much evidence indicates that TRPV1 is expressed in neuronal and glial cells of various brain regions, although at lower levels than in peripheral structures and spinal cord[17–20]. The somatodendritic distribution of TRPV1 in neurons did not always account for the presynaptic effect mediated by this channel. However, compelling studies also indicate postsynaptic TRPV1 activation as a trigger of some forms of long-term synaptic depression[21–24].

Some work on cultured microglia and microglial cell lines shows that TRPV1 is also functionally expressed in brain resident immune cells. Once activated, TRPV1 causes a variety of effects ranging from microglia cell death to phagocytosis, cell migration, cytokine production and ROS generation[25–30].

In the present study, we investigated the roles of TRPV1 in the anterior cingulate cortex (ACC) of the rodent brain[31] and we evaluated the impact of neuropathic pain on the channel function. First, we found that TRPV1 is mainly functional in the microglia rather than in neurons of both naïve and sham-operated mice. Its activation, beyond controlling microglia reaction *per se*, modulated microglia-neuron communication, by promoting release of extracellular vesicles (EVs) from microglia. Indeed, EVs are important mediators of intercellular communication between microglia and brain cells[32], and selectively enhance neurotransmission by promoting sphingolipid metabolism in excitatory neurons[33].

Finally, we observed that in mice rendered neuropathic by peripheral nerve ligation, TRPV1 is also expressed in cortical neurons, and its activation directly affects both the intrinsic membrane properties and the synaptic strength of cortical pyramidal neurons (PNs), likely accounting for the increased cortical excitability that characterizes chronic pain states.

## Results

### TRPV1 is expressed in microglia of ACC and other brain areas.
So far, immunohistochemical evidence (in which a variety of polyclonal antibodies-pAbs-has been employed) showed that in many brain areas TRPV1 is mostly present in neuronal cell bodies, and rarely expressed in astrocytes and microglia (Tables 1 and 2).

In the present study we found that a monoclonal anti-TRPV1 antibody (anti-TRPV1 MAb) stained fibers and cell bodies in ACC sections of both adult ($n=9$ mice; Fig. 1a,b) and young mice ($n=8$, Supplementary Fig. 1). The antibody specificity was tested in tissue samples from $TRPV1^{-/-}$ mice ($n=6$ mice). Unlike the anti-TRPV1 pAbs, neither cytoplasm nor fibers of cells

in tissues from $TRPV1^{-/-}$ mice were marked by the anti-TRPV1 MAb (Supplementary Fig. 2). Double immunofluorescence for TRPV1 and the neuronal marker NeuN showed that the TRPV1 positive processes mainly surrounded neuronal bodies (Fig. 1a, $n=4$). Sparse overlapping between the two antibodies was evidenced in confocal maximum projection (Fig. 1a, NeuN/TRPV1 panel). However single $1–2\,\mu m$ z-stacks showed that merged signal did not result from neuronal staining but could be attributed to the slight overlap of TRPV1 fibres surrounding neurons. The amount of colocalization between TRPV1 and NeuN was 'very weak' as indicated by the Pearson coefficient of correlation (PCC; $r=0.125\pm0.012$, $n=4$; Fig. 1c)[34]. Subsequent double-immunostaining with the anti-TRPV1 MAb and the microglial marker Iba1 revealed that TRPV1 was mainly expressed in microglia (Fig. 1b), as confirmed by the high degree of colocalization of TRPV1 and Iba1 signals ($r=0.720\pm0.010$, $n=5$, Fig. 1c). These fluorescence signals were higher in ACC sections of young than adult mice (Supplementary Fig. 3). Thus, the largest part of TRPV1 was expressed in microglia although not all microglial cells were TRPV1-embed (Supplementary Fig. 3).

To validate this hypothesis, analyses on highly purified neurons and microglial cells from cortical wt and $TRPV1^{-/-}$ tissues were performed. Although mRNA transcript levels of TRPV1 were comparable between neurons and microglia (Supplementary Fig. 4a), protein analysis evaluated by flow cytometry revealed low mean fluorescence intensity levels of surface TRPV1 expression in NeuN positive neurons ($MFI=48.2\pm7.4$, $n=3$ from 3 mice pools), whereas CD11b positive microglial cells showed intense levels ($MFI=124.8\pm15.2$, $n=3$ from 3 mice pools) (Fig. 1e,f). These findings were further corroborated through western blot analysis on protein extracts of ACC and postnatal cortical microglia cultures. TRPV1 was immunodetected as a band of 100 kD in the ACC extracts from wild type but not from $TRPV1^{-/-}$ mice (Supplementary Fig. 4b). We also detected a second band of 75 kDa that was still revealed in $TRPV1^{-/-}$ tissues and was absent in cultured microglia cells (non specific band, Supplementary Fig. 4b,c).

Consistent with a dominant microglial localization, we found that, for equal total protein amounts, postnatal cortical microglia cultures expressed higher level of TRPV1 compared with ACC tissues (Supplementary Fig. 4d).

Finally, we performed patch-clamp recordings of GFP-expressing microglial cells from acute $Cx3cr1^{+/-}$ ACC slices, to directly detect the presence of functional TRPV1s in these cells. A voltage ramp protocol was used to measure the current–voltage relationship before and after the application of capsaicin. In these conditions, we observed the activation of a capsaicin-evoked outward rectifying current (Supplementary Fig. 4e), consistent with the typical properties of TRPV1 channels[1].

Altogether these data strongly indicate that functional TRPV1 is present in the microglia of mouse cortex.

The anti-TRPV1 MAb also labelled astrocytes of the ACC (Supplementary Fig. 5a,c). Compared with microglia, the TRPV1 expression was significantly lower in astroglial cells ($r=0.084\pm0.006$, $n=3$, for anti-TRPV1/anti-GLAST and $r=0.0753\pm0.004$, $n=4$, for anti-TRPV1/anti-GLT1; $P<0.01$ for both markers; Two Sample $t$-Test). To test if the expression level in astrocytes was dependent on the brain area, the hippocampus was chosen for comparison, to avoid a general underestimation of TRPV1 in astrocytes (Supplementary Fig. 5b,d). A low degree of TRPV1-GLAST and TRPV1-GLT1 colocalization was also detected in hippocampus ($r=0.176\pm0.019$, $n=3$; $r=0.219\pm0.015$, $n=4$, respectively). These data indicated that TRPV1 is expressed at much lower levels in astrocytes than in microglial cells.

**Table 1 | TRPV1 antibodies employed in brain studies.**

| Species | N-terminal | C-terminal | Staining pattern | TRPV1$^{-/-}$ | Company | Authors |
|---|---|---|---|---|---|---|
| Rabbit Polyclonal | + | − | Cell bodies (nucleus and cytoplasm), dendrites, and terminals | − | Immunological Sciences | Marinelli et al.[70] |
| Rat polyclonal | − | + | Neuronal cytoplasm and microglial branches | − | Sensory Research Center Seoul National University (Seoul, Korea) | Kim et al.[25] |
| Rabbit Polyclonal | − | + | Neuronal cell bodies and dendrites | − | Chemicon (Temecula, CA, USA) | Tòth et al.[18] |
| Rabbit Polyclonal | − | + | | − | Calbiochem (San Diego, CA., USA) | |
| Guinea pig | − | + | Neuronal cytoplasm | + | Dr Julius Lab, UCSF | Cavanaugh et al.[16] |
| Guinea pig | − | + | Neuronal cytoplasm | − | Harlan (Indianapolis, USA) | Mezey et al.[17] |
| Rabbit | + | − | | − | New Zealand White | |
| Goat Polyclonal | + | − | Neuronal cytoplasm and processes | + <br> − <br> − | Santa Cruz Biotechnology, (Santa Cruz, CA, USA) | Cristino et al.[19]; Maione et al.[71]; Fogaça et al.[72]; Casarotto et al.[73]; |
| Rabbit Polyclonal | + | − | microglial processes and cell bodies | − | Novus Biologicals (Littleton, CO) | Sappington and Calkins[26] |
| Guinea pig Polyclonal | − | + | Neuronal cytoplasm | − | Abcam (Cambrige, UK) | Maione et al.[74] |

**Table 2 | TRPV1 antibodies employed in the present study.**

| Species | N-terminal | C-terminal | Staining pattern | TRPV1$^{-/-}$ | Company |
|---|---|---|---|---|---|
| Rabbit Polyclonal | − | + | Neuronal cytoplasm | + | Neuromics |
| Rabbit Polyclonal | + | − | Neuronal cytoplasm | + | Immunological Science |
| Mouse Monoclonal | − | + | Microglial branches and neuronal cytoplasm | + | Millipore Bioscience Research |

Note that '+' in the TRPV1$^{-/-}$ columns indicates that the antibody has been tested in tissue from TRPV1$^{-/-}$ mice.

The microglial TRPV1 expression pattern was also found in other brain 'painmatrix' areas i.e thalamus, somatosensory cortex, periaqueductal gray (PAG) and hippocampus (Supplementary Fig. 6) while in the spinal cord and dorsal root ganglia TRPV1 was detected also in neurons (as well established) (Supplementary Figs 7–9), pointing to a selective microglial expression of the receptor only in brain areas.

**TRPV1s are expressed in cortical neurons of mice with pain**. Many studies demonstrated the fundamental role of TRPV1 in pain induction and detection at peripheral structures and spinal cord[35], and compelling evidence indicates that TRPV1 is involved in nociceptive processing also at brain level[36–42]. Our new evidence on TRPV1 localization in brain microglia, prompted us to assess the expression pattern of this channel in the ACC of adult mice suffering from neuropathic pain at different time points after the ligature of the sciatic nerve (chronic constriction injury, CCI).

Three days after the nerve constriction, TRPV1 was expressed in cortical microglia similarly to sham-operated mice ($n = 3$; Supplementary Fig. 10). One week after CCI, anti-TRPV1 mAb stained also the cytoplasm and the apical dendrite of Iba1 immunonegative cells whose morphology resembled neurons ($n = 5$; Fig. 2a–h). This immunoreactivity pattern was limited to layer 2/3 of both hemispheres and was less marked in the ispilateral than in the contralateral hemisphere. At 4 weeks after CCI, the cytoplasmic expression of TRPV1 in Iba1-negative cells was broadly distributed throughout all layers of the contralateral hemisphere (Fig. 2i–p; $n = 7$). In line with these observations,

TRPV1 microglial expression decreased in the contralateral ACC of 1 week and 4 week CCI mice. Indeed, the Pearson correlation coefficient for anti-TRPV1/Iba1 signal colocalization was weaker in the contralateral ACC of 1 week and 4 week CCI mice compared with ispilateral cortices (Fig. 2q–t). Double immunofluorescence staining with NeuN and anti-TRPV1 revealed that TRPV1 was expressed in both cytoplasm and branches of NeuN-positive neurons ($n = 3$; Fig. 3a), thus suggesting that in CCI adult mice TRPV1 was expressed in cortical neurons, besides microglial cells. Triple immunofluorescence labelling for NeuN, TRPV1 and EAAC1 (the neuronal glutamate transporter marker) showed that vanilloid receptor type 1 was mostly expressed in pyramidal neurons. Indeed, whereas almost all EAAC1 immunopositive neurons expressed TRPV1 (Fig. 3b), only a small fraction of EAAC1 negative neurons were immunoreactive for TRPV1 ($n = 3$).

The neuronal expression pattern was prominent in ACC, although other areas of the painmatrix such as the thalamus, somatosensory cortex, PAG and hippocampus displayed sporadic TRPV1 neuronal expression. The anti-TRPV1 MAb labelling of neurons was absent in cortical sections from CCI TRPV1$^{-/-}$ (Supplementary Fig. 11).

In cortical sections from young CCI mice, TRPV1 was never detected in neurons, at none of the time points tested (3,7 and 14 days post-CCI; Supplementary Fig. 1b,c).

**Microglial TRPV1s control glutamatergic neurotransmission**. Growing evidence shows that activation of brain microglia modulates excitatory neurotransmission and increases synaptic

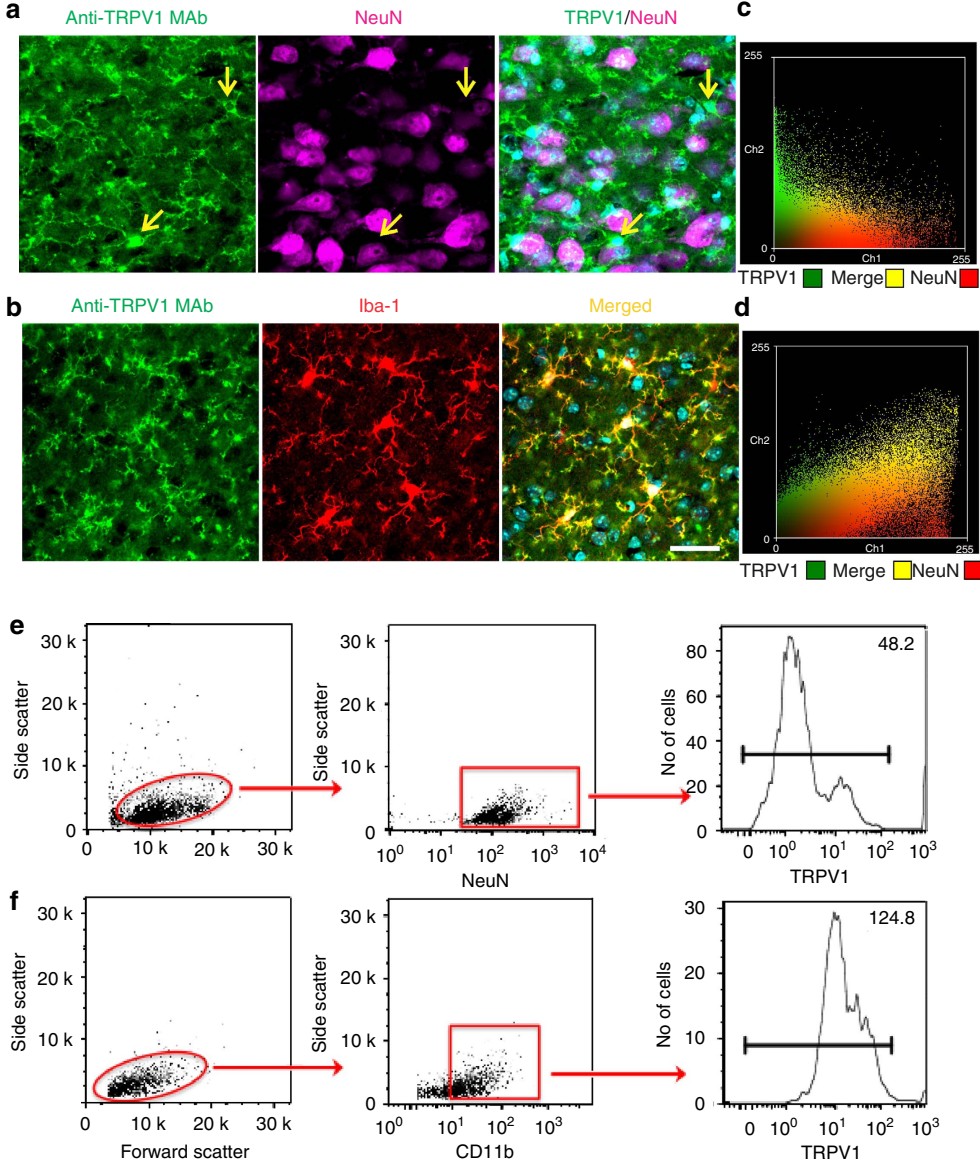

**Figure 1 | TRPV1 protein expression in microglial and neuronal cells of the ACC of adult naive mice. (a)** Photomicrograph of double immunofluorescence for anti-TRPV1 MAb (green) and NeuN (magenta). The anti-TRPV1 MAb labels mainly fibers that surround neurons (NeuN positive cells) and sparse NeuN negative cytoplasm (arrows). **(b)** Anti-TRPV1 positive processes highly overlap with the microglial marker iba1 (in red) in the layer 2/3 of ACC. **(a,b)** DAPI, nuclear marker (in blue). **(c,d)** Graphic representation (scatter plot) of the correlation coefficient of Pearson (PCC) for quantifying the colocalization between the anti-TRPV1 and NeuN (in yellow, PCC = 0.13) and anti-TRPV1 and Iba-1 (PCC = 0.77). Note that higher PCC values correspond to a strong colocalization[32]. **(e,f)** Neuronal and microglial cells were isolated from cortical tissue using specific magnetically labelled kits. According to physical parameters, obtained cells were gated on NeuN and CD11b for identification of neurons **(e)** and microglial cells **(f)** respectively. Surface expression of TRPV1 was analysed by means of flow cytometry. Data are representative of a single experiment and show the mean fluorescence intensity (MFI) of 3 independent pools of at least 3 mice.

strength[43–45]. Given our data on TRPV1 expression in brain microglia, we hypothesized that the well-characterized presynaptic modulation of neurotransmission by TRPV1 may be indirect, due to stimulation of TRPV1 on microglia rather than on neurons.

To address this possibility, we first explored the action of TRPV1 stimulation in the ACC. As reported for other brain areas, capsaicin (1 μM) selectively increased the frequency of miniature excitatory postsynaptic currents (mEPSCs) onto PNs of ACC without affecting their amplitude (Fig. 4a). This effect is due to α-amino-3-hydroxy-5-methyl-4-isoxazolepropionic acid receptors activation (AMPA; Supplementary Fig. 12).

Both genetic deletion and pharmacological blockade of TRPV1 by the non-competitive antagonist IRTX inhibited the enhancement of mEPSC frequency by capsaicin (Fig. 4b).

Among the signalling lipids that are able to activate TRPV1, we tested the lysophosphatidic acid (LPA). This bioactive and proinflammatory lysophospholipid is highly produced during inflammation and activates microglial cells which, in turn, self-sustains LPA synthesis[46,47]. Beyond acting on its high-affinity metabotropic receptors LPA1–4, LPA also binds to the TRPV1 C-terminal[48]. Similarly to capsaicin, LPA significantly enhanced the frequency of mEPSCs (Fig. 4c) and did not affect mEPSC amplitude ($P < 0.9$). The enhancement of

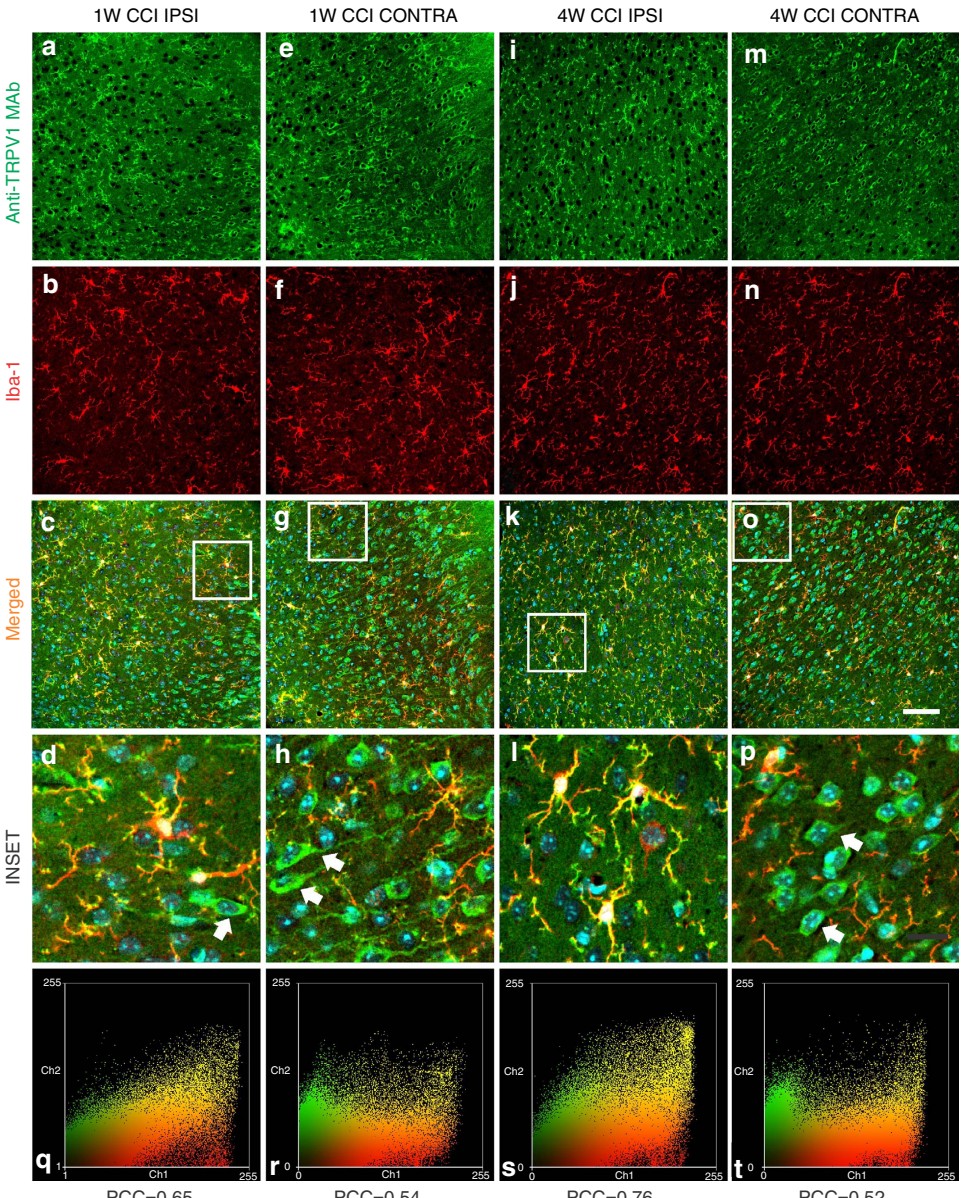

**Figure 2 | TRPV1 distribution pattern in the ACC of CCI adult mice.** Beyond microglia cells (iba-1 positive, in red), the anti-TRPV1 MAb stains both the cytoplasm and apical processes of some iba1 immunonegative cells (white arrows in the INSET). This expression pattern was minor in the superficial cortical layers of the ipsilateral (IPSI; **a**–**d**) than contralateral (CONTRA, **e**–**h**) hemisphere of mice that underwent surgery from 1 week (1 W CCI mice). In 4-week CCI mice (**i**–**l**, **m**–**p**), when the chronicization of pain becomes established, the anti-TRPV1 MAb labels both cytoplasm and apical processes of iba1 negative cells (green) of the CONTRA hemisphere (**o**,**p**). Note that this staining pattern is also present in the deeper layers of the ACC (**o**) while is absent in the IPSI hemisphere of 4 W CCI mice (**k**,**l**). (INSET) High-power views of cells from the box areas in all merged panels. (**q**–**t**) Colour scatter plots representing the amount of colocalization (yellow) between the anti-TRPV1 (green) and iba-1 (red) in each condition. Smaller PCC values denote a higher expression of TRPV1 in the cytoplasm of iba1-immunonegative cells.

mEPSC rate by LPA was significantly dampened in slices incubated with IRTX (Fig. 4e) and was completely abolished when both TRPV1 and LPA metabotropic receptors were blocked (Fig. 4g). Taken together, these data suggest that LPA may behave as a potential endogenous activator of TRPV1 in the brain.

Then, we performed a set of experiments in the presence of the tetracycline minocycline. This antibiotic exerts anti-inflammatory and antimicrobial effects by preventing microglia activation[49,50]. Minocycline (100 nM) did not affect basal mEPSCs (from $2.34 \pm 0.59$ to $2.28 \pm 0.66$ Hz, $P = 0.85$ and from $19.54 \pm 1.49$ to $19.29 \pm 1.62$ pA, $P = 0.82$; $n = 9$) but fully inhibited the capsaicin-induced increase of mEPSC rate (Fig. 5a–c). These data strongly suggest that the enhancement of glutamatergic neurotransmission by capsaicin is due to the activation of TRPV1 expressed by microglia cells.

To assess whether astrocytes work as mediators of microglia-neuron communication upon TRPV1 activation, capsaicin was tested in the presence of the glial-specific toxin fluoroacetate (FAc)[43,51]. We found that capsaicin still enhanced mEPSCs rate (Fig. 5d,e,f), thus suggesting that, at least in these experimental conditions, astrocytes are not required for this type of TRPV1-mediated microglia-to-neuron communication.

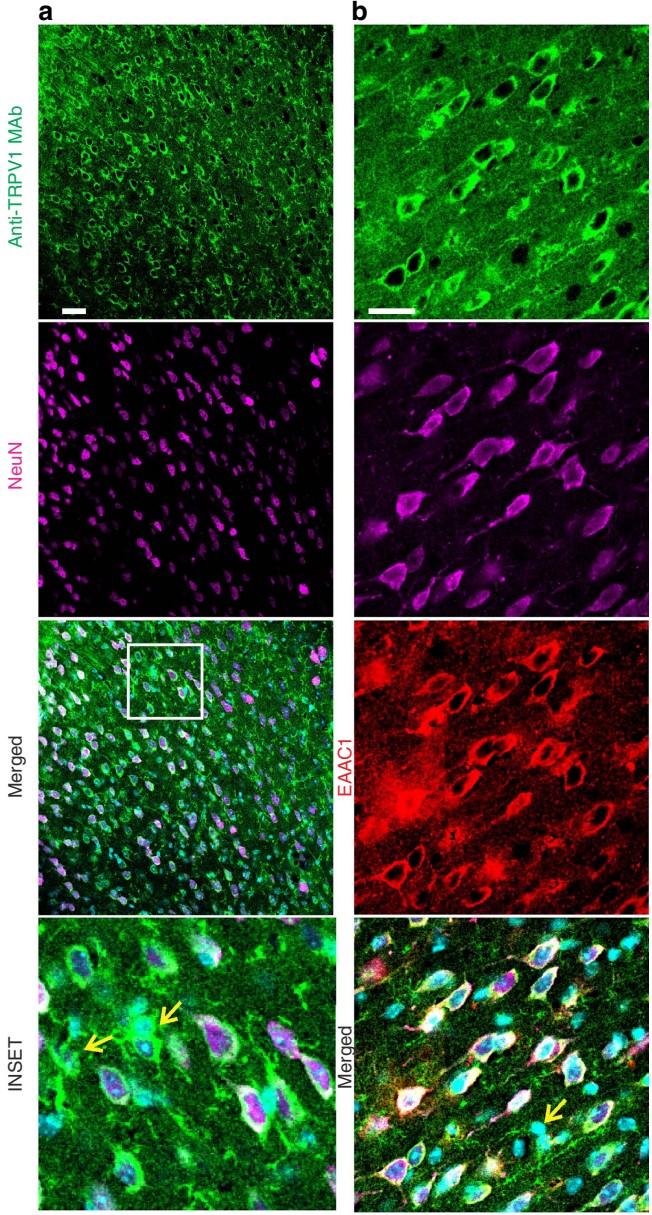

**Figure 3 | TRPV1 is also expressed in principal neurons of the ACC from CCI adult mice.** Contralateral ACC photomicrographs of 4 weeks CCI mice, from two independent experiments (**a,b**). (**a**) Anti-TRPV1 MAb labels both the cytoplasm and the apical dendrite of NeuN positive cells. The remnant TRPV1 positive fibers and cytoplasms probably belong to glial cells (yellow arrows). (**b**) Anti-TRPV1 MAb labels EAAC1 positive neurons and NeuN negative cells (yellow arrow).

**TRPV1 activation stimulates microglial microvesicle release.** Next step was to define the molecular mechanism underlying the increase of mEPSC frequency driven by microglial TRPV1 activation. Among several factors, microglia-to-neuron communication occurs through EVs shed from the plasma membrane of microglia, called microvesicles (MVs) or ectosomes[33,52]. Production of MVs increases in reactive microglia[53], and is strongly enhanced by ATP through activation of P2X$_7$ receptors[54]. Hence, TRPV1 stimulation on microglia may promote MV shedding as well. To verify this hypothesis, cultured murine microglia were challenged with capsaicin (300 nM) for 10 min and EVs, both shed MVs and smaller

vesicles, were quantified and sized by nanoparticle tracking analysis (NTA). NTA showed a multimodal size distribution of quite large vesicles, presumably MVs (mean diameter control $= 298.2 \pm 21.2$ nm; capsaicin $= 305.0 \pm 16.32$ nm; $n = 3$), with a major peak consisting of vesicles about 200 nm in size and few smaller peaks of larger MVs (Fig. 5g). EV concentration increased by 1.5 fold under TRPV1 or ATP stimulation (Fig. 5h, $n = 3$). Selective quantification of MVs shed from microglia that have incorporated fluorescent phosphocholine (NBD-C6_HPC) in the plasma membrane, confirmed that both ATP and capsaicin enhance MV production to the same extent (Fig. 5i, $n = 3$; $P < 0.05$).

These data suggest that capsaicin may increase mEPSC rate by triggering release of MVs upon TRPV1 activation on microglia.

To this aim, we employed an inhibitor of acid ceramidase, the ARN14988 (ref. 55), to prevent neuronal sphingosine production, essential for MV-induced stimulation of synaptic activity[33]. This compound did not affect, *per se,* basal mEPSC frequency, although it slightly reduced their amplitude (Supplementary Fig. 13a). In slices pre-treated with ARN14988, capsaicin did not affect either frequency or amplitude of mEPSCs (Fig. 5j). To exclude possible inhibitory effects of ARN14988 on MV production or activity, microglia cultures were treated with the ceramidase inhibitor before exposure to capsaicin. NTA quantification showed that ARN14988 did not inhibit capsaicin-dependent MV release (Supplementary Fig. 13b). Furthermore, patch-clamp recordings revealed that MVs from ARN14988-treated cells retained the capability to increase mEPSC frequency in cultured neurons (Supplementary Fig. 13c).

To strengthen our hypothesis, we pharmacologically blocked MV shedding from microglia with the p38 MAPK inhibitor SB203580. p38 MAPK activates downstream P2X7 ATP receptor and is essential for MV shedding[56]. Given that p38 MAPK is activated downstream TRPV1 receptors[57], we reasoned that inhibition of p38 MAPK could block capsaicin-induced MV shedding as well. Indeed, capsaicin-induced MV shedding from cultured microglia treated with SB-203580 was significantly reduced (Supplementary Fig. 13d). Finally, capsaicin failed to affect the frequency and amplitude of mEPSCs in slices incubated with SB-203580 (Fig. 5k).

Altogether, these data suggest that capsaicin possibly increases spontaneous glutamatergic synaptic activity by promoting shedding of microglial MVs, which in turn fosters sphingosine metabolism in neurons and enhances presynaptic release probability.

**TRPV1 controls cortical microglia activation.** In the healthy mature brain, microglial cells play a role in immune surveillance and ensure the maintenance of brain homeostasis. Upon injuries these cells shift to an activated state characterized by drastic changes in the cellular shape, functional behavior and by the release of different proinflammatory and immunoregulatory factors[58,59]. Conforming to the capsaicin-mediated induction of microglial chemotaxis[29], we investigated whether TRPV1 stimulation regulates the morphology of microglial cells.

Morphometric analyses (ImageJ plugin, Supplementary Fig. 14a) showed that the majority of microglial cells from acute slices incubated in ACSF had a ramified shape, displaying small cell bodies with thin and elongated processes (Fig. 6a,g). Compared with controls (vehicle and ACSF treated slices) (Fig. 6a,b), sections incubated with 1 µM capsaicin for 10 min displayed a bulk of hypertrophied iba1 positive cells showing larger cell bodies, and thicker, shorter and highly branched projections (Fig. 6c,g). Of note, capsaicin-induced morphological changes were absent in sections from $TRPV1^{-/-}$ mice

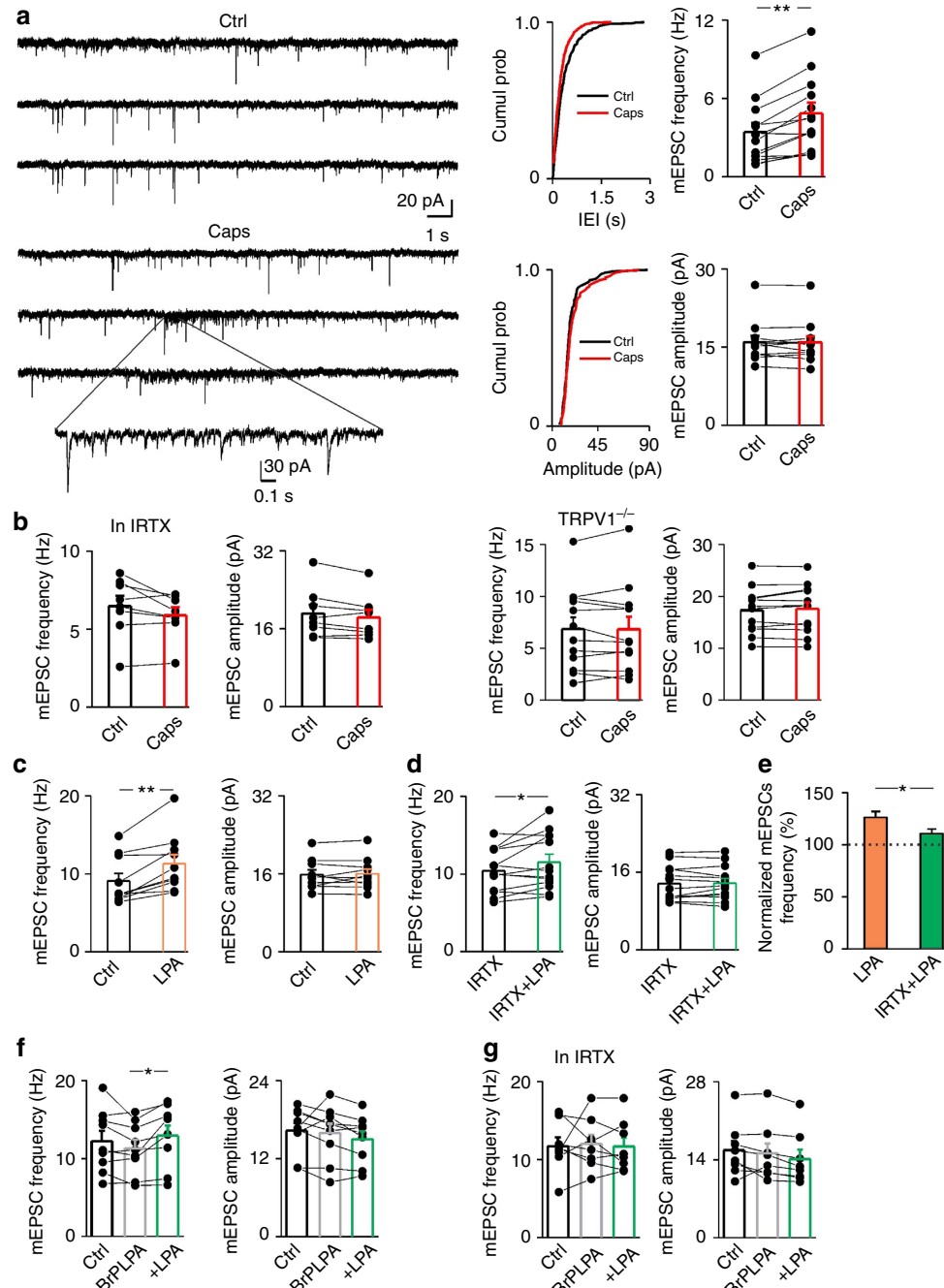

**Figure 4 | TRPV1 activation increases mEPSCs onto pyramidal neurons of the ACC.** (**a**) Left, example traces of AMPAR mEPSCs recorded from a PN at −70 mV, in control and after 1 μM capsaicin (caps), in the presence of picrotoxin (100 μM). Below trace shows expanded trace recording during caps. Middle, cumulative distributions of mEPSC interevent interval (IEI) and amplitude from cell shown on the left (P < 0.02 and P = 0.32 respectively). Right, summary histograms and line series plots of mEPSC frequency and amplitude showing that capsaicin significantly increases the frequency but not the amplitude of mEPSCs (from 3.40 ± 0.71 to 4.83 ± 0.84 Hz, n = 12, P < 0.001; from 15.97 ± 1.13 to 15.90 ± 1.60 pA, n = 12, P = 0.83). Data are values from single cells (black filled circle) and mean ± s.e.m. (**b**) Group data of mEPSC frequency and amplitude showing the lack of effect of capsaicin in the presence of the TRPV1 antagonist IRTX (left; from 6.46 ± 0.68 to 5.88 ± 0.50 Hz, n = 8, P = 0.11) or in cortical tissue from TRPV1$^{−/−}$ mice (right; 6.84 ± 1.15 to 6.81 ± 1.20 Hz, n = 12, P = 0.9). (**c**) Left, summary graph of mEPSC frequency and amplitude before (black bar; 9.10 ± 0.96 Hz and 15.43 ± 0.95 pA, n = 10) and during 5 μM LPA (orange bar; 11.32 ± 1.1 Hz and 15.38 ± 1.07 pA, n = 10). LPA significantly boost mEPSC frequency (P < 0.002; Wilcoxon Signed Ranks Test). (**d**) Same as in previous graphs but in the presence of IRTX (IRTX 10.41 ± 0.76.14, IRTX + LPA 11.54 ± 1 Hz, n = 13; P < 0.05 Paired Sample T Test; IRTX 12.92 ± 0.93,IRTX + LPA 13.01 ± 0.93 pA, n = 13). (**e**) Note that LPA is significantly less effective when TRPV1 is inhibited (P < 0.05, Two-sample T test) (*P < 0.05, **P < 0.001). (**f,g**) In the presence of LPA1-4 metabotropic receptors antagonist BrP-LPA, LPA increases glutamatergic transmission by TRPV1 activation. (**f**) Left, summary graphs of mEPSC frequency and amplitude before (black bar; 12.21 ± 1.32 Hz and 16.31 ± 1.9 pA, n = 9), during BrP-LPA (5 μM; grey bar; 11.32 ± 1.1 Hz and 15.95 ± 1.4 pA, n = 9) and with LPA (green bar; 12.94 ± 1.3 Hz and 15.00 ± 1.2 pA, n = 9). (**g**), same as in **f** but with the addition of IRTX (ctrl 11.71 ± 1.14 Hz, BrLPA 12.03 ± 1.15 Hz, BrLPA + LPA 11.67 ± 1.16 Hz, n = 8; ctrl 15.70 ± 1.70 pA, BrLPA 15.16 ± 1.80 pA, BrLPA + LPA 14.31 ± 1.60 pA, n = 8). (*P < 0.05, Paired Sample *t*-Test).

(Fig. 6f,h). In the latter, microglia displayed already a hypertrophied morphology under baseline conditions. (Fig. 6d–f,h; Supplementary Fig. 14b). Subsequent flow cytometry experiments revealed that WT microglia challenged with capsaicin produced significantly higher levels of TNF-α ανδ lower levels of IL-10 (Fig. 6i,j), whereas $TRPV1^{-/-}$ microglia expressed equal amounts of TNF-α with compared with WT cells and

significantly greater amounts of IL-10 (Fig. 6i,j). Thus, stimulation of TRPV1 induced a pro-inflammatory phenotype of microglia from WTs. Conversely, microglia shifted toward an anti-inflammatory phenotype when TRPV1 is lacking.

We next asked whether TRPV1 might be endogenously active under inflammatory conditions. Lipopolysaccharide (LPS), the most abundant component within the cell wall of Gram-negative

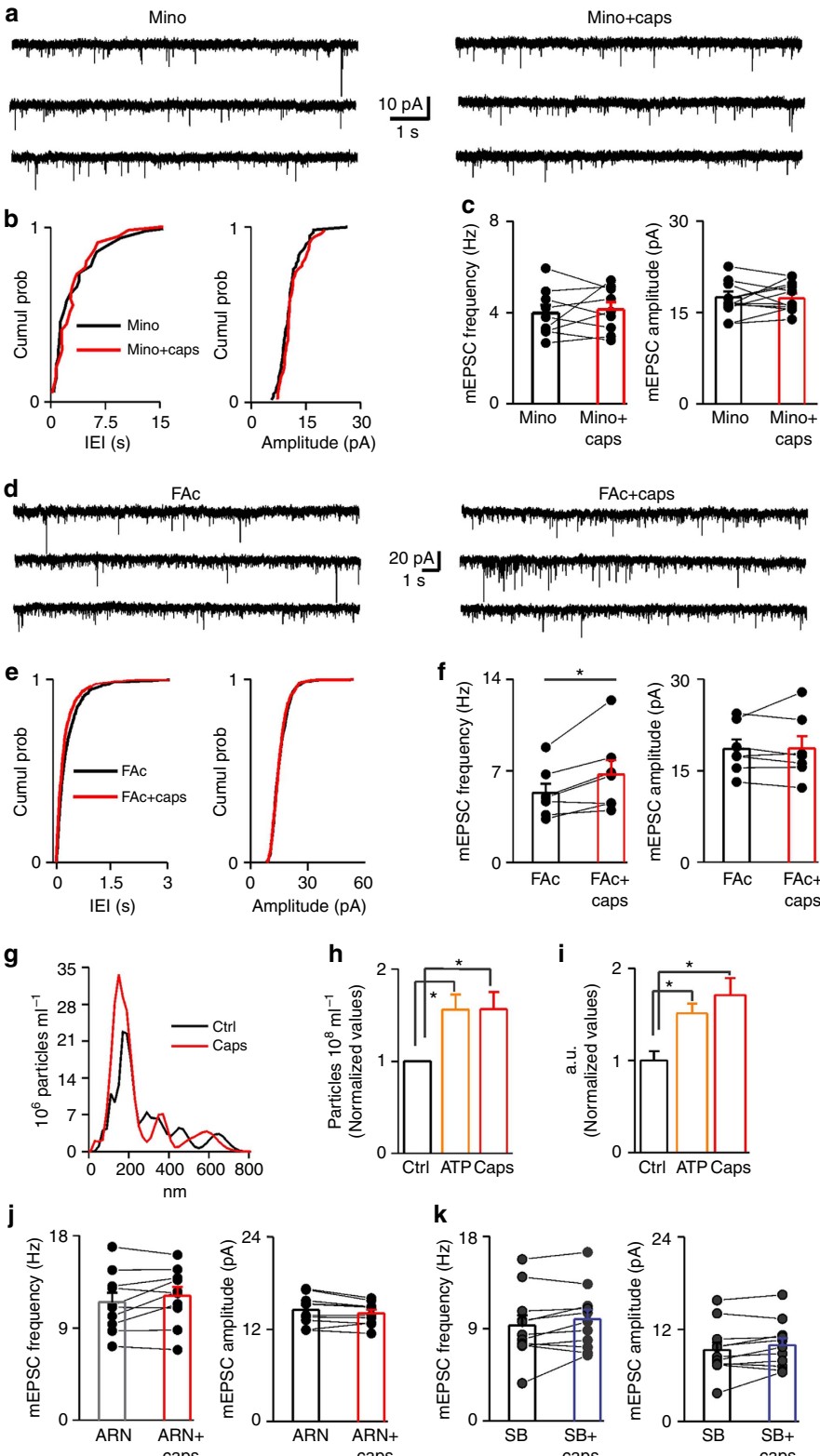

bacteria, exerts inflammatory action by activation of microglia. $500 \, ng \, ml^{-1}$ LPS increased the mEPSC frequency without affecting their amplitude (35% of mEPSC frequency enhancement, Fig. 7a,b)[43]. This effect was tightly restricted to activation of glial cells, since it was consistently blocked by minocycline and by the astroglia blocker FAc (1 mM) (from $5.73 \pm 1.13$ to $5.80 \pm 1.24 \, Hz$, $n = 11$, $P = 0.49$; from $18.52 \pm 1.48$ to $18.01 \pm 1.40 \, pA$, $n = 11$, $P = 0.22$).

The overall increase of glutamatergic neurotransmission by LPS was significantly reduced by IRTX ($n = 14$, $P < 0.05$; Fig. 7c right panel). Likewise, both the enhancement of mEPSCs and morphological changes induced by LPS were absent in $TRPV1^{-/-}$ mice (Fig. 7d and Supplementary Fig. 14c, respectively).

Along with these data, application of capsaicin following saturating LPS failed inducing any further potentiation of mEPSCs onto PNs (Fig. 7e,f). These results suggest that the lack and/or block of TRPV1s prevent the inflammatory effects of LPS by keeping microglia in an M2 anti-inflammatory state.

**TRPV1s affect neuron excitability and synaptic strength.** TRPV1 expression in the soma and dendrites of cortical pyramidal neurons (PNs) of CCI mice may directly affect neuronal excitability, thereby enhancing cortical hyperexcitability, a distinct feature of chronic pain.

In baseline conditions, neither input resistance (shams $137.37 \pm 9.9 \, M\Omega$ $n = 24$, CCI $147.12 \pm 10.6 \, M\Omega$ $n = 23$, $P = 0.5$, Two Samples $t$-Test) nor resting membrane potential (shams $-70.96 \pm 1.52 \, mV$ $n = 24$, CCI $-67.93 \pm 2.65 \, mV$ $n = 23$, $P = 0.32$, Two Samples $t$-Test) of PNs neurons from sham and CCI mice were different. However, the number of action potentials evoked by current injections of increasing amplitude was significantly higher in CCI than in shams (Supplementary Fig. 15a left and right input-output curves, respectively). The increase of firing can be ascribed, at least partly, to TRPV1 expression in CCI mice neurons, since IRTX caused a rightward shift in the input current-action potential rate curves (Supplementary Fig. 15b). In parallel, mEPSC frequency was enhanced in CCI mice compared with control animals (Supplementary Fig. 15c). Unexpectedly, the amplitude of mEPSCs was greatly reduced (Supplementary Fig. 15d). The amplitude drop of spontaneous glutamatergic currents could be attributed to AMPA receptor trafficking, as a neuronal homeostatic mechanism subsequent to the persistent increase of synaptic activity[60].

Application of capsaicin, besides frequency (as in control mice, see Fig. 4) significantly scaled up both amplitude and charge transfer of mEPSCs in 10 out of 18 PNs recorded from CCI mice (12% and 16% increase, respectively; Fig. 8a–c). To dissect out microglial versus neuronal TRPV1 contribution in the capsaicin-induced increase of synaptic strength, PNs from CCI mice were recorded in the presence of minocycline. In a similar proportion of neurons recorded in these conditions (4 out of 13 neurons, $P = 0.39 \, \chi^2$-test), capsaicin significantly enhanced the mEPSC amplitude and the charge transfer, indicating that only neuronal TRPV1s account for capsaicin effect on miniature synaptic events (Supplementary Fig. 15e).

Beside its effects on the glutamatergic transmission, capsaicin (1 μM) induced a slight but consistent membrane depolarization associated with a significant reduction of the input resistance in PNs of CCI mice (Fig. 8d,e). This effect was independent from postsynaptic glutamatergic or GABAergic ionotropic receptors stimulation, as it was obtained in the presence of DNQX, APV and picrotoxin. Conversely, in neurons from shams (Fig. 8e) we found no change in the same intrinsic membrane parameters. In voltage-clamp experiments, capsaicin elicited inward currents associated with a significant enhancement of membrane conductance in CCI PNs, but not in shams (Fig. 8f,g). In capsaicin-responding neurons ($n = 4$), IRTX successfully reverted capsaicin-induced inward current (Supplementary Fig. 15f–g).

## Discussion

The present study shows that TRPV1 is expressed and functional in the microglia of the anterior cingulate cortex of healthy mice. Notably, TRPV1 activation directly affects microglia function, which in turn modulates synaptic neurotransmission. This evidence arises from: (a) mRNA TRPV1 expression in cortical microglia, validated by the lack of expression in cells from knock out tissue as negative control; (b) high levels of TRPV1 protein expression in cortical microglia cells as indicated by western blotting, flow cytometry and immunofluorescence assays in cortical tissue; (c) the outward rectifying current induced by capsaicin in microglia cells from cortical slices; (d) the shedding of microvesicles from microglial surface upon capsaicin application; and (e) morphological and phenotypic changes of microglial cells upon TRPV1 activation. Importantly, TRPV1-mediated increase of cortical spontaneous glutamatergic neurotransmission is absent when microglia is switched off by minocycline and is occluded by the inflammatory agent LPS. Conversely, we found that in the cortex of mice suffering from neuropathic pain, TRPV1 is expressed also in neurons, affecting their intrinsic electrical properties and synaptic strength. This different TRPV1 expression pattern under chronic pain conditions places

**Figure 5 | TRPV1 is functionally expressed in cortical microglia and modulates synaptic transmission by microvesicles shedding.** (**a**) Example recording of mEPSCs before (left) and during (right) capsaicin in the presence of minocycline. (**b**) Cumulative probability plots comparing minocycline (black line) and minocycline plus capsaicin (red line) on mEPSC IEI and amplitude of the recording showed above ($P = 0.25$ and $P = 0.31$, KS test). (**c**) Bar histograms of group data showing the lack of capsaicin-mediated increase of mEPSC rate when microglia activation is blocked by minocycline (from $3.98 \pm 0.35$ to $4.13 \pm 0.33 \, Hz$, $n = 9$, $P = 0.53$; from $17.54 \pm 0.94$ to $17.34 \pm 0.79 \, pA$, $n = 9$, $P = 0.70$, Paired Sample $t$-Test). Group data are presented as single value and mean $\pm$ s.e.m. (**d**) Example recording of mEPSCs before (left) and during (right) capsaicin in the presence of 1 mM FAc. (**e**) Cumulative probability plots comparing FAc (black line) and minocycline plus capsaicin (red line) on mEPSC IEI and amplitude of the recording showed above ($P < 0.01$ and $P = 0.28$, KS test). (**f**) Summary data of both mEPSC frequency and amplitude recorded in the presence of FAc before and during capsaicin application (FAc $5.30 \pm 0.72 \, Hz$, FAc + caps $6.70 \pm 1.10 \, Hz$, *$P < 0.05$ Paired $t$-test; FAc $18.58 \pm 1.55 \, pA$, FAc + caps $18.63 \pm 1.98 \, pA$, $n = 7$, $P = 0.89$ Paired Sample $t$-Test). (**g**) Representative size profile of EVs released constitutively (black line) or upon capsaicin challenge (red line) from $5 \times 10^5$ murine microglia in 500 μl of serum free medium during a period of 10 min. (**h**) Histogram showing the fold increase of total EV concentration detected by Nanosight upon stimulation with ATP or capsaicin (Kruskal-Wallis test, $P = 0,035$). (**i**) Histogram showing the fold increase of MVs shed selectively from the plasma membrane and measured by a spectrophotometric assay under the same stimuli (Holm-Sidak's, $P < 0,05$). (**j**) Group data showing that addition of ARN14988 reduces capsaicin-mediated increase of mEPSC frequency in cortical slices (from $11.51 \pm 0.92$ to $12.12 \pm 0.88 \, Hz$, $P = 0.15$; from $14.52 \pm 0.61$ to $14.10 \pm 0.46 \, pA$, $P = 0.06$, Paired Sample $t$-Test, $n = 10$). (**k**) Summary graph of mEPSC frequency (left) and amplitude (right) in the presence of 2 μM SB203580 before (black bar) and after capsaicin (blue bar) (from $9.30 \pm 1.00$ to $9.93 \pm 0.90 \, Hz$, $P = 0.08$; from $10.67 \pm 0.47$ to $10.75 \pm 0.49 \, pA$, $n = 11$, and $P = 0.89$, Paired Sample $t$-Test and Wilcoxon Signed Rank Test, respectively). Data are values from single cells and/or mean $\pm$ s.e.m. *$P < 0.05$.

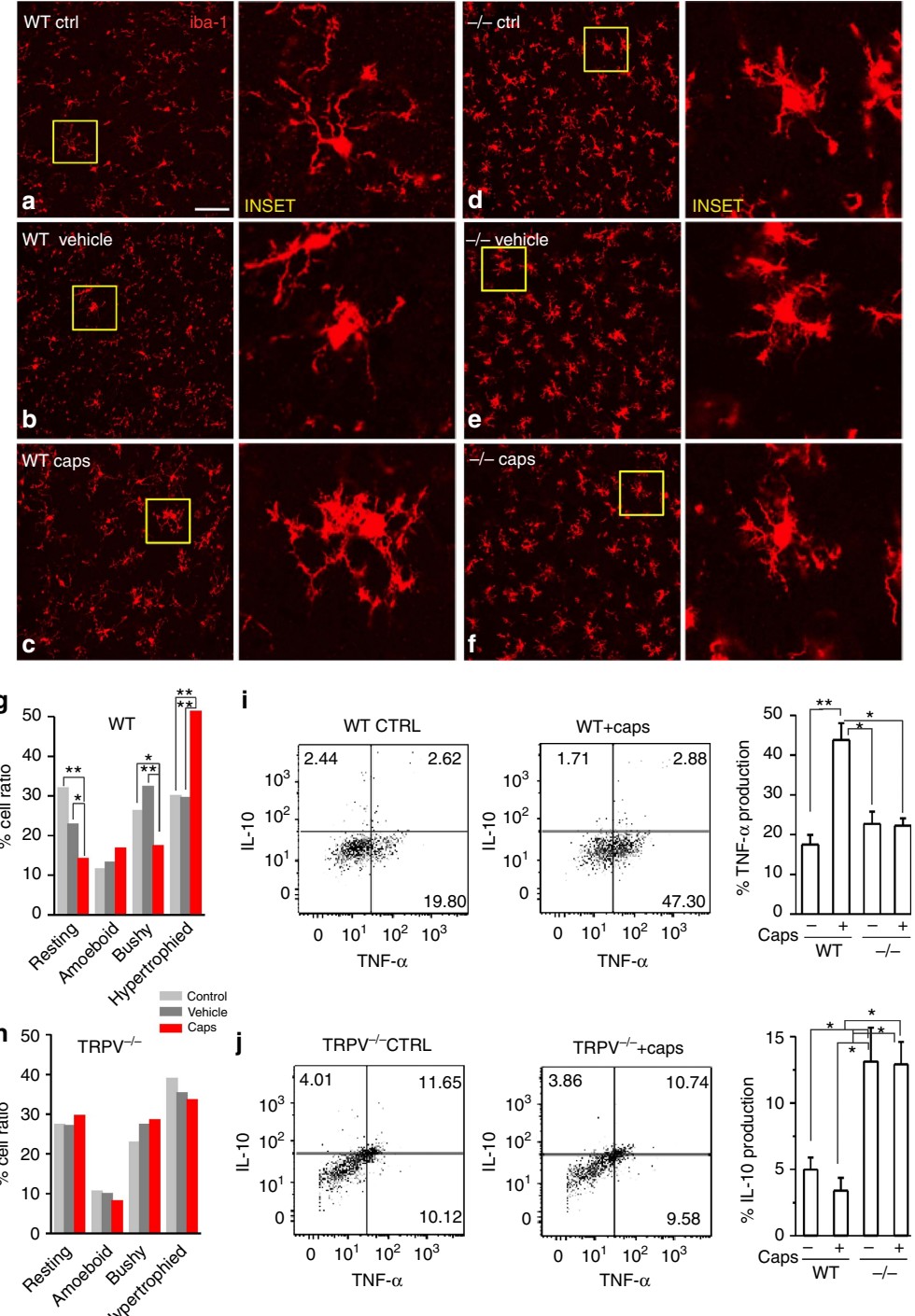

**Figure 6 | The presence or absence of TRPV1 influences microglia phenotype.** (**a**–**f**) Sections of cortical tissue from WT and TRPV1$^{-/-}$ mice, fixed after exposure to ACSF (**a**,**d**), ACSF plus vehicle (DMSO; **b**,**e**) and ACSF plus 1μM capsaicin (**c**,**f**) and immune-processed for iba-1 to stain microglia cells (in red). INSETs are zoom images taken from an area delimited by the yellow square for each condition. (**g**), Bar graph of percentage of cortical microglia cell phenotype (resting, ameboid, bushy and hypertrophied), in control (grey bars), vehicle- (dark grey bars) and capsaicin- treated (red bars) cortical sections from WT mice. Capsaicin treatment causes a significant shift from ramified and bushy to hypertrophied morphology. (**h**), Same as in 'g' but in cortical sections from TRPV1$^{-/-}$ mice. In these tissues capsaicin fails to induce morphological changes of microglia cells. Note that microglia cells in $-/-$ tissues are already hypertrophied in control conditions (**d**–**f**,**h**). *$P < 0.05$, **$P < 0.01$ Fisher exact test. Acutely isolated CD11b+ microglial cells from WT (**i**) and TRPV1$^{-/-}$ mice (**j**) were cultured in the absence ($-$) or presence ($+$) of capsaicin (1μM) for 10 min, washed and then incubated with Brefeldin A for 6 h to allow cytokine accumulation in vesicles. Intracellular production of TNF-α and IL-10 was analysed by means of flow cytometry. Data represent percentage of cytokine production ± s.e.m. of 3 independent pools of at least 3 mice. *$P < 0.05$, **$P < 0.01$ (Bonferroni's multiple comparison test following parametric one-way Anova). Note that 'Percentage of TNFalpha/IL-10 production' means the 'Percentage of TNFalpha producing cells'.

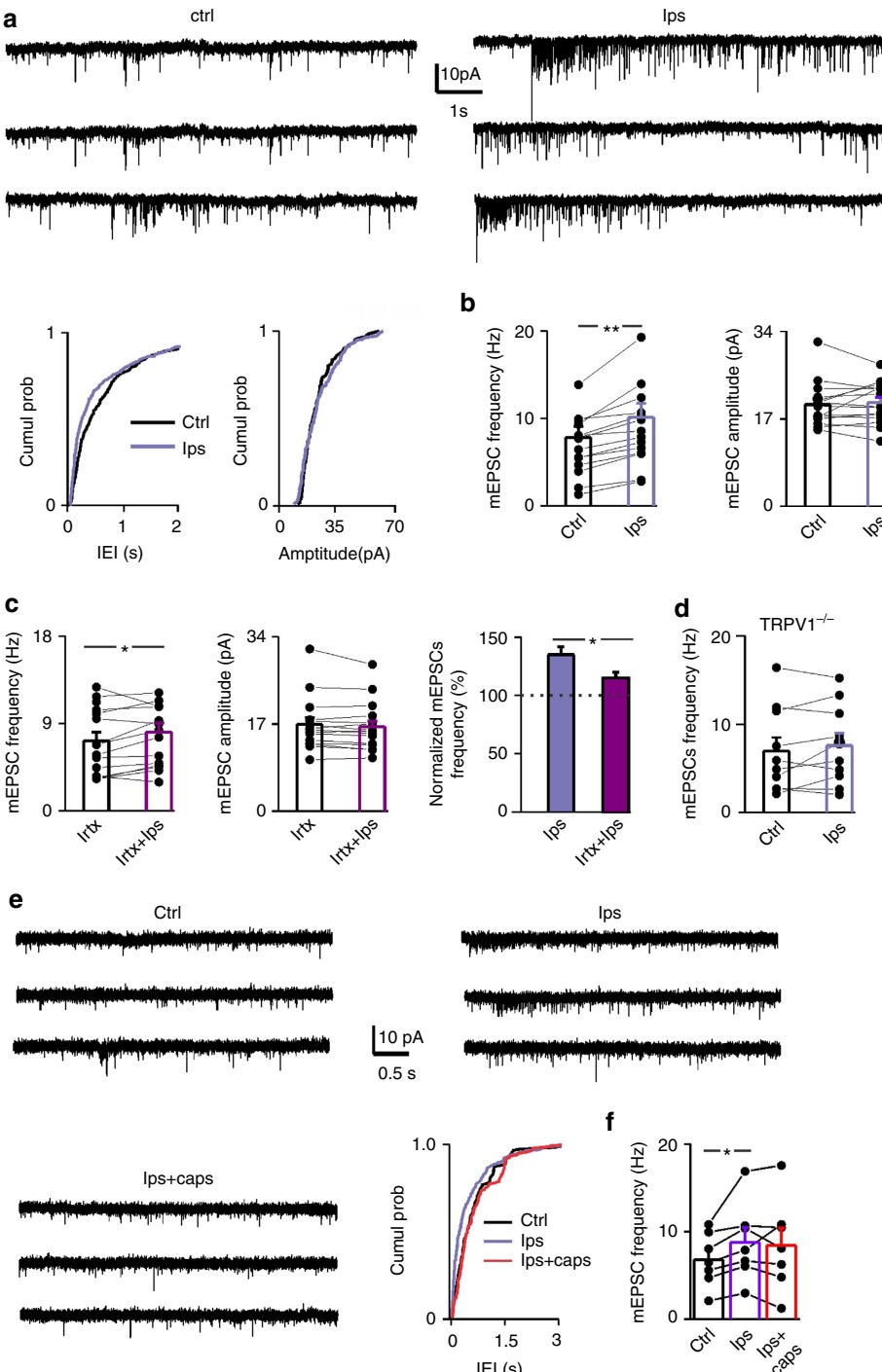

**Figure 7 | TRPV1 tonically controls LPS-mediated effects.** (**a**) Example recording of mEPSCs in control (ctrl) and during LPS exposure (500 ng ml$^{-1}$). Below the cumulative probability plot of mEPSC interevent interval and amplitude ($P < 0.01$ and $P = 0.24$, respectively, KS test). (**b**) Population plot of mEPSC frequency before (black bar and filled circle) and during LPS (violet bar and filled circle), showing that LPS, similarly to capsaicin, increases selectively the mEPSC frequency (from $7.67 \pm 1.32$ to $10.09 \pm 1.73$ Hz, $n = 14$, $P < 0.001$ Paired Sample Wilcoxon Signed Rank test) without a significant effect on the amplitude (from $20.05 \pm 1.30$ to $19.76 \pm 1.32$ pA, $P = 0.5$ Paired Sample $t$-Test). (**c**) As in b, but in the presence of 300 nM IRTX (from $6.8 \pm 1.06$ to $7.78 \pm 1.19$ Hz, $n = 14$, $P < 0.01$ Paired Sample Wilcoxon Signed Rank test; from $17.47 \pm 1.17$ to $17.11 \pm 1.23$ pA, $P = 0.09$ Paired Sample Wilcoxon Signed Rank test). Right, the percentage increase of mEPSC frequency by LPS in the presence of the TRPV1 antagonist IRTX, is significantly reduced compared with the rate enhancement induced by LPS applied alone ($P = 0.01$ Mann–Whitney test). (**d**) Population data of mEPSC frequency in control and during LPS obtained from PNs of TRPV1$^{-/-}$ mice (from $7.36 \pm 1.90$ to $7.69 \pm 1.81$ Hz, $n = 9$, $P = 0.52$ Two Samples $t$-Test). (**e**) Trace records from a PN in control condition (ctrl), in LPS and LPS plus capsaicin. Following LPS perfusion, capsaicin is no longer able to further facilitate glutamatergic neurotransmission. Right below, cumulative distribution probability of the interevent intervals from the same neuron (ctrl versus LPS $P > 0.0001$, LPS versus caps $P < 0.0001$, ctrl versus caps $P = 0.54$). (**f**) Summary graph of mEPSC frequency before (black bar), during LPS (violet bar) and LPS + caps (red bar) (ctrl $6.69 \pm 1.15$ LPS $8.78 \pm 1.68$ Hz, LPS + caps $8.45 \pm 1.97$ Hz, $n = 7$, ctrl versus LPS $P < 0.05$, LPS versus LPS + caps $P = 0.68$, ctrl versus LPS + caps $P = 0.15$ Paired Sample $t$-Test;. Data are values from single cells and/or mean ± s.e.m. *$P < 0.05$, **$P < 0.001$.

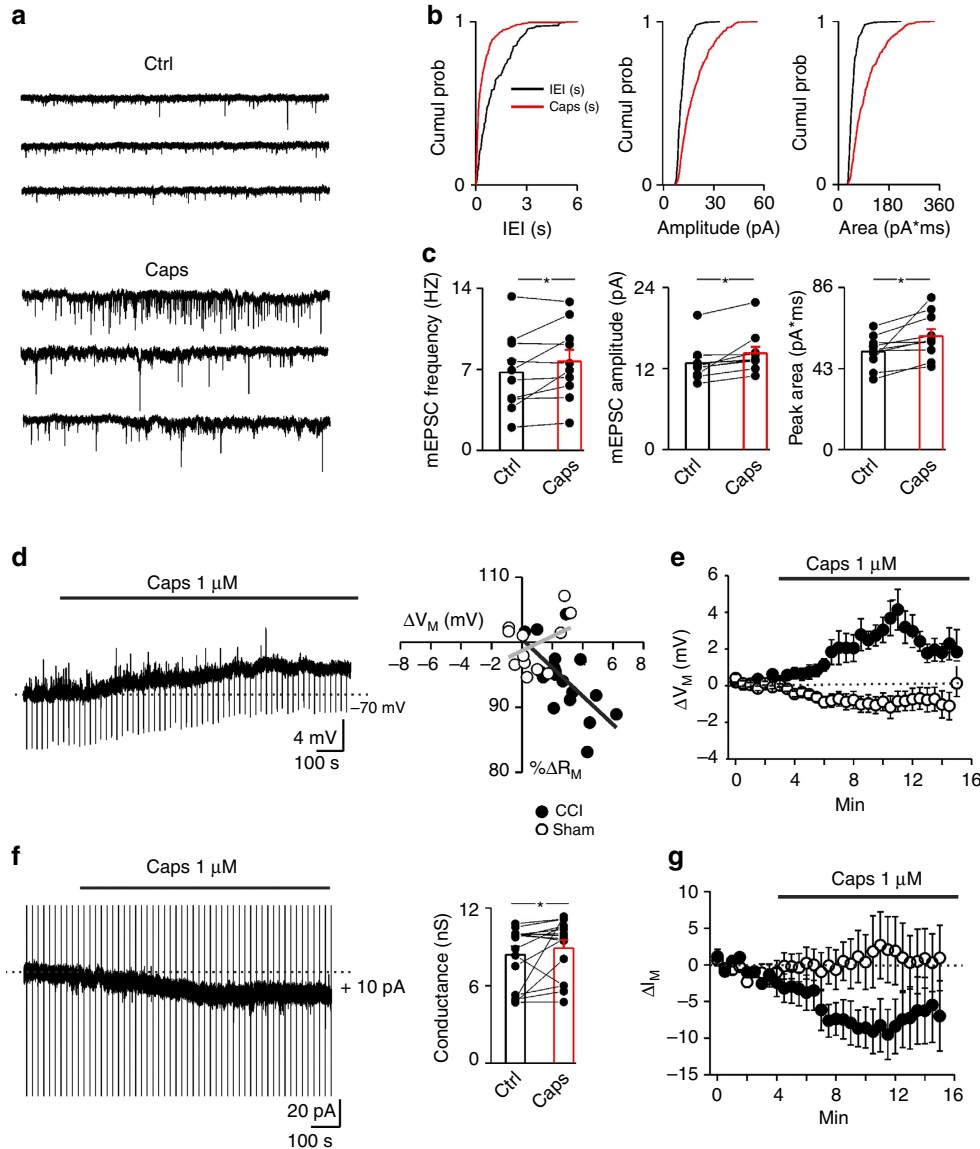

**Figure 8 | Neuronal TRPV1 stimulation directly modulates layer 2/3 pyramidal neuron synaptic strength in CCI mice.** (**a**) Example recording of mEPSCs before and during capsaicin from a CCI pyramidal neuron. Note that capsaicin increases both the frequency and amplitude of mEPSCs. (**b**) Cumulative probability plots comparing the IEI (left panel), amplitude (middle panel) and area (right panel) of control mEPSCs with those recorded in capsaicin, (KS-test $P < 0.001$). (**c**) Summary graph of mEPSC frequency (left), amplitude (middle) and peak area (right) before (black bar, solid circle) and during capsaicin (red bar, solid circles) (frequency from $6.73 \pm 1.05$ to $7.66 \pm 1.02$ Hz, $P = 0.05$ Paired Sample $t$-Test; amplitude from $12.90 \pm 0.84$ to $14.26 \pm 0.95$ pA, $P < 0.01$ Paired Sample Wilcoxon Signed Rank test; Area from $52.03 \pm 2.67$ to $60.20 \pm 3.74$ pA*ms, $P < 0.05$ Paired Sample $t$- test; $n = 10$). (**d**) Left, current-clamp recording of a pyramidal neuron responding to capsaicin with depolarization and reduced $R_M$. Negative deflections are membrane potential responses to negative current injections (60pA). Right, plot of $\%\Delta R_M$ (normalized to pre- capsaicin values) versus $\Delta V_M$ (membrane potential values in capsaicin subtracted from baseline ones) in all tested neurons from CCI and sham mice ($n = 18$ and $n = 12$, filled and white symbols, respectively). These data were analysed by performing a linear fit to evaluate the strength of association between the two variables, through the coefficient of determination $r^2$. In sham, $\Delta R_M$ values do not correlate with variations in $\Delta V_M$ ($r^2 = 0.07$) while in PNs from CCI mice, capsaicin-induced changes of $V_M$ correlate with a $R_M$ with a value of $r^2 = 0.36$. (**e**), Summary time plot of subtracted membrane potential ($\Delta V_M$) of CCI and sham PNs in response to capsaicin ($P < 0.001$, Two Sample $t$-Test). (**f**) Left, example of voltage-clamp recording (at $-70$mV) of a CCI PN responding to capsaicin with an inward shift of the holding current. Vertical deflections (truncated in the figure) are membrane current responses to voltage steps to test passive properties and recording stability. Right, population graph of membrane conductance in control (black bar) and during capsaicin perfusion (red bar; $n = 12$; *$P < 0.05$ Paired Sample Wilcoxon Signed Rank test). (**g**), Summary plot of $\Delta I_M$ (membrane current values during capsaicin subtracted to baseline values) versus time for PNs from sham and CCI mice ($n = 12$ and $n = 14$, respectively; $P < 0.001$, Mann–Whitney test).

microglial TRPV1 at the center of a new important mechanism, providing a link between physiological and pathological states.

TRPV1 is functional in *in vitro* microglia[26–29] and growing evidence indicates a modulatory role of excitatory neurotransmission by microglia cells[43–45]. Accordingly, our study demonstrates that, in resting conditions, acute activation of TRPV1 expressed in microglia accounts for the well-known presynaptic modulation of glutamatergic neurotransmission by this channel observed in several rodent brain areas, and adds new insights on the apparent paradox of a postsynaptic TRPV1

localization inducing presynaptic effect on glutamate release[15,61]. In addition to the glutamatergic hypothesis, microglia affects neuronal excitability also by disinhibition mechanisms[62], thus likely explaining the TRPV1 regulation of inhibitory neurotransmission at dentate gyrus synapses[63]. Differently, in 'activated' conditions, our evidence on TRPV1 functional expression in neurons may argue for the postsynaptic TRPV1 modulation of some forms of long term depression[21–24].

The different expression pattern revealed by the anti-TRPV1 MAb and the polyclonal antisera lies in the well-known uniqueness of antibodies whose ability to recognize the epitope relies heavily on the experimental technique employed (in which proteins can be variously fixed or unfixed, denatured or undenatured). Indeed, the anti-TRPV1 MAb also recognized the native neuronal receptor in the cytofluorimetry assay on living neurons. This demonstration of TRPV1 cortical microglia expression is complemented by the evidence at the mRNA level, exploiting probes covering the exon sequence lacking in the TRPV1 mutant mice: TRPV1 mRNA is copiously expressed in purified cortical microglial cells, as well as in neurons, while no significant amount is detected in $TRPV1^{-/-}$ tissue and cells.

Hence, our data do not exclude the existence of a functional TRPV1 postsynaptically expressed, but posit that this channel in neurons could be activated upon specific stimuli, thus not showing up under our experimental conditions of the immuno-fluorescence and electrophysiological analysis. Indeed, only models of long term synaptic plasticity unveil a TRPV1-mediated postsynaptic effect[21–24].

In contrast to our and other studies[17–20], evidence using TRPV1 reporter transgenic mice, led to the conclusion that TRPV1 expression is low and restricted to few brain areas[16]. Different reasons could underlie this discrepancy. First, the protein expression levels downstream of IRES sequences are variable, and can be lower than that from the cap-dependent translation of the primary mRNA[64,65]. This might lead to artificially low reporter levels in brain areas that do express TRPV1. Second, the putative reporter mRNA is very long, encompassing TRPV1, PLAP and lacZ coding sequences, which may well have a different stability and shorter half-life. Third, the precise genomic context around a reporter gene can significantly affect the chromatin organization of that locus and thus the expression pattern of the reporter gene. Thus, a systematic comparative study revealed significant differences between Cre recombination patterns and the expression pattern of the endogenous genes. In conclusion, the expression pattern of the reporter genes in a transgenic mouse might not always precisely mimic the real expression of the endogenous gene. Intriguingly, since the identity of the TRPV1-positive cells stained in the brain by the lacZ reporter was not investigated[16], these cells could also include the glial cells immunopositive for the TRPV1 monoclonal antibody identified in the present study.

Remarkably, the present data clarify the TRPV1 role in the brain with respect to that held by the same receptors in the peripheral somatosensory system. Indeed, TRPV1s expressed in the main immune cells of the brain may similarly function as detectors of nociceptive and inflammatory stimuli/agents.

Accordingly, we found that capsaicin shifted microglia to a hypertrophic and pro-inflammatory state by increasing the production and release of cytokines that is, TNFα. Other studies showed increase production of L6 and ROS formation following TRPV1 activation in microglial cells[26,27].

Notably, during inflammation or upon more generic environmental cellular insults, potential endovanilloids such as n-acyl amides, monoacylglycerols and bioactive lipids markedly increase[46,66,67]. Among these candidates, LPA seems to be a suitable endogenous agonist since, similarly to capsaicin, it

activates potassium outward currents in microglia cells and induces microglial microvesicles shedding[68,69]. Our present data unveil a TRPV1-mediated enhancement of excitatory neuro-transmission by LPA. Certainly, other candidates cannot be excluded i.e. the N-acylethanolamides anandamide, palmitoyl-ethanolamide, oleoylethanolamide and monoacylglycerols[6].

Remarkably, we found that when TRPV1s are defective, microglia shift to anti-inflammatory phenotype, by producing large amounts of IL10. Given the substantially reduced effects of the endotoxin LPS in $TRPV1^{-/-}$ slices, we hypothesize that loss-of-function mutations of TRPV1 may keep these cells in a refractory status and decrease the immunological response to pathogens.

The morphology of microglial cells activated by capsaicin is in line with the observed enhancement of excitatory neurotransmission by microglial TRPV1 stimulation. Previous findings under-scored how the amount of contact/interactions between microglia processes and neuronal compartments shapes neuronal excitability. In particular, the fewer microglial processes contact neurons (activated microglia), the greater is the synaptic neuronal activity and the surface expression of AMPA receptors.

Similarly to other proinflammatory agents[53], we found that capsaicin promotes the shedding of microglial MVs, suggesting that endovanilloid signalling may represent another route for a rapid propagation of inflammatory signals from microglia to neurons. The potential involvement of MVs in TRPV1-mediated microglia-to-neuron communication is indicated by a significant reduction of capsaicin-induced increase of mEPSC frequency under inhibition of p38 MAPK phosphorylation and block of sphingosine metabolism. While the first is essential for MV shedding, since its activation causes the translocation into the plasma membrane of a key enzyme (acid sphingomyelinase) mediating MV budding[56], the latter is a crucial step for the fusion of synaptic vesicles with the neuronal plasma membrane and, therefore, is a key effector of presynaptic stimulation of excitatory transmission mediated by MVs[33].

Recent evidence points to astrocytes as a relay between microglia to neuron communication[43]. However this would not be the case of our study because, despite the block of astroglia, acute stimulation of TRPV1 still affects neurotransmission.

An additional potential mechanism underlying TRPV1-mediated microglia-to-neuron communication could be the release of microglial factors from MV breakage or from microglia cells themselves, which modulate glutamatergic neurotransmission via the interaction with their downstream receptors (Supplementary Fig. 17). Further experimental work is needed to address this possibility.

A major finding of this study is the expression of TRPV1 also in neurons of adult mice suffering from neuropathic pain. This new TRPV1 cellular distribution might be part of a homeostatic response that links physiology to pathology and is time-dependent from pain onset, being absent at early stages and gradually overspreading in all cortical layers of the contralateral hemisphere once the chronicization endures. Since our data indicate that under control healthy mice conditions TRPV1 mRNA is present in cortical neurons, we assume that a post-transcriptional regulation mechanism of gene expression may account for the new functional expression of TRPV1 protein in the cortex of mice affected by neuropathic pain, that is, an increase of translational efficiency or a decrease of protein degradation in neurons, activated by the persistent central sensitization in chronic pain conditions. The possibility of translational control of TRPV1 mRNA would call naturally into play TRPV1-selective miRNAs in neurons, that might release their translational repression in an activity dependent manner, under the chronic pain conditions that we demonstrated to induce ACC neuronal expression of TRPV1.

Similarly, changes of TRPV1 cellular expression were shown in DRG neurons during inflammation, reasoning for a different role of this protein at different stages of hyperalgesia.

Importantly, we never found anti-TRPV1 positive neurons in the cortex, thalamus and hippocampus of young mice (two/three weeks old), at different time points from neuropathic pain onset. Given that neuropathic pain is constitutively suppressed during early life by anti-inflammatory neuroimmune regulation, the shift of TRPV1 expression in cortical neurons of adult mice may be linked to glial released inflammatory factors during the central sensitization in neuropathic pain. Once released at the site of injury, cytokines, growth factors and prostaglandins, in addition to contributing to peripheral and central sensitization mechanisms, are capable of signaling to the brain via both neural and blood-borne routes and, once within the brain, they result in de novo production by glial cells.

Based on the above observations, we hypothesize that inflammatory molecules may regulate post-transcriptional mechanisms of gene expression by enhancing neuronal TRPV1 protein translation, possibly via activity-dependent miRNA-mediated mechanisms.

At least in our mice with neuropathic pain, neuronal TRPV1 expression is consistently restricted to the ACC, rather than other brain areas in which the neuronal distribution of this receptor is only occasionally observed. To note, cortico-spinally projecting neurons play direct effect on pain regulation, undergoing synaptic pontentiation following nerve injured. Probably these cortically spinally projected neurons may be our TRPV1 immunopositive neurons in mice whose nerve was injured. In fact, TRPV1 activation modulates both intrinsic and extrinsic excitability of these cells by increasing their firing rate, depolarizing membrane potential and enhancing synaptic strength. Together, these data suggest that TRPV1 in cortical PNs of neuropathic mice may be crucial for the enhanced cortical activity and alteration of the emotional component in subjects affected by chronic pain. Actually, TRPV1 mediates both anxiety and emotional alterations.

The contribution of microglial TRPV1 to the hyperexcitability and synaptic strength of CCI PNs appears to be, on the other hand, less relevant, as shown by the persistence of these features when microglia function is blocked. This conclusion is supported by the reduced TRPV1 expression in cortical microglia of CCI mice as well as by the well-standardized presynaptic effect of MVs.

Hence, neuronal TRPV1 maybe the effective player in changing the electrophysiological properties of PNs, once pain becomes chronic.

On a final note, our study suggests that future strategies aimed at imaging TRPV1 cellular expression and distribution may provide differential diagnostic tools for pathologies associated with an inflammatory brain component.

## Methods

All experiments were performed in accordance with Directive 2010/63/EU of the European Parliament and of the Council on the protection of animals used for scientific purposes, and approved by both the animal-welfare body (article. 25 of Legislative Decree 26/2014), established at the registered facility 'Fondazione Santa Lucia' and the Italian authority veterinary service (Italian Ministry of Health). All efforts were made to avoid animal suffering and to minimize the number of animals used.

**Animals.** Both adult (P80-120) and young (P20-30), C57BL/6 (Charles River Laboratories, Como, Italy), B6.129S4-Trpv1$^{tm1Jul}$/J mice (TRPV1$^{-/-}$; The Jackson Laboratory, Bar Harbor) and CX3CR1$^{+/GFP}$ male mice (P18-30) from D. Ragozzino (Sapienza University) were employed in this study.

**Chronic constriction injury and testing conditions.** All animals were housed in standard cages and kept under a 12-h light–dark cycle in an air-conditioned facility. Following the procedure originally proposed by Bennett and Xie (1988),

adapted for mice, chronic constriction injury of the sciatic nerve (CCI) was used as murine model of neuropathic pain. CCI was obtained by three unilateral ligatures of sciatic nerve. In the following, injured and uninjured hindpaws are indicated as ipsilateral and contralateral hindpaws with respect to the sciatic nerve ligation. Anesthesia was performed intraperitoneally with zoletil (tiletamine and zolazepam, 100 mg ml$^{-1}$, 0.5 ml kg$^{-1}$) and rompun (xylazine, 20 mg ml$^{-1}$, 0.5 ml kg$^{-1}$)[67].

All mice were examined for mechanical allodynia at days 3, 5, 7, 10, 12, 14, 17, 19, 21,31 post-CCI. The behavioral test was conducted during the morning and in blind conditions.

For ex vivo and in vitro experiments, mice were sacrificed at different time points after CCI and the assessment of allodynia development. For adults, we used the following time table: three days, one or four weeks after CCI, while young mice were sacrificed at three days, one or two weeks after CCI. In particular, for adults the mechanical threshold (force-grams) was measured in naive and CCI at day 3, 7 and 21 after neuropathy induction (Supplementary Fig. 16).

As control, we used mice that underwent surgery without the sciatic nerve ligature (sham animals) and naïve animals, which did not undergo to any surgery. Both in vivo and in vitro data obtained from naïve and shams were pooled since no differences have been observed. Mechanical allodynia was assessed by measuring withdrawal threshold of both hindpaws to normally non-noxious punctuate mechanical stimuli by using an automatic von Frey apparatus (Dynamic Plantar Aesthesiometer, Ugo Basile, Comerio, Italy). After a 5 min of adaptation to the apparatus, the mechanical stimulus was applied to the midplantar surface of both hindpaws. The mechanical threshold was measured as the maximal force (expressed in grams) at which mice withdrew its hindpaw. At each testing day, withdrawal threshold were taken as mean of three consecutive measurements with 10-s interval between each measurement.

**Electrophysiological recordings from neurons.** Both naïve/sham and CCI adult C57BL6 and TRPV1 KO mice were deeply anesthetized with isofluorane inhalation, decapitated, and brains removed and immersed in cold 'cutting' solution (4 °C) containing (in mM): 126 choline, 11 glucose, 26 NaHCO$_3$, 2.5 KCl, 1.25 NaH$_2$PO$_4$, 10 MgSO$_4$, 0.5 CaCl$_2$ equilibrate with 95% O$_2$ and 5% CO$_2$. Coronal slices (300 μm) were cut with a vibratome (Leica) and then incubated in oxygenated artificial cerebrospinal fluid (ACSF) containing (in mM): 126 NaCl, 26 NaHCO$_3$, 2.5 KCl, 1.25 NaH$_2$PO$_4$, 2 MgSO$_4$, 2 CaCl$_2$ and 10 glucose; pH 7.4, initially at 32 °C for one hour, and subsequently at room temperature, before being transferred to the recording chamber and maintained at 32 °C. Recordings were obtained from visually identified pyramidal neurons in layer 2/3, easily distinguished by the presence of an emerging apical dendrite. Experiments were performed in the whole-cell configuration of the patch-clamp technique. Electrodes (tip resistance = 3–4 MΩ) were filled with an intracellular solution containing (in mM): Kgluconate 135, KCl 4, NaCl 2, HEPES 10, EGTA 4, MgATP 4 NaGTP 2; pH adjusted to 7.3 with KOH; 290 mOsm. Whole-cell voltage-clamp recordings (−70 mV holding potential) were obtained using a Muticlamp 700B (Axon CNS, Molecular Device). Spontaneous EPSCs recorded in the presence of tetrodotoxin (TTX; 1 μM) (mEPSCs) were filtered at 1 kHz, digitized at 10 kHz, and recorded on computer using Digidata1440A and pClamp10 software (Molecular Device). Series resistances were not compensated to maintain the highest possible signal-to noise and were monitored throughout the experiment. Recordings were discarded if Rs changed 25% of its initial value. Experiments in voltage-clamp recording were carried out in the presence of GABA$_A$ receptor antagonist picrotoxin (100 μM) and TTX (1 μM). Spontaneous events were detected and analysed with Clampfit 10.4 using amplitude and area thresholds set as a multiple (3–4X) of the SD of the noise. Each event was also visually inspected to prevent noise disturbance of the analysis. The cumulative amplitude and interevent plots obtained for each cell in controls and after drug application were compared using the Kolmogorov–Smirnov (KS) test.

Each slice received only a single exposure to capsaicin or to other agonists. In experiments performed to investigate a postsynaptic TRPV1-mediated response, the ionotropic glutamate receptor blockers 6,7-dinitroquinoxaline-2,3,dione (DNQX, 10 μM) and DL-2-amino-5-posphonovaleric acid (DL-APV, 50 μM) and the GABA$_A$ receptor antagonist picrotoxin (100 μM) were included in the bath solutions. For voltage-clamp experiments TTX were also added.

For current-clamp experiments membrane resistance was measured from responses to current injections (−60 pA, 250 ms, 0.2 Hz) and were performed in the presence of ionotropic glutamatergic and GABA$_A$ receptor antagonists.

**Electrophysiological recordings from microglia cells.** Acute slice of cingulate cortex (cG1 and cG2) was prepared from CX3CR1$^{+/GFP}$ male mice (P18-30). Animals were decapitated under halothane anesthesia, and whole brains were rapidly immersed for 10 min in chilled standard artificial cerebrospinal fluid (ACSF) containing (in mM): NaCl 125, KCl 2.3, CaCl$_2$ 2, MgCl$_2$ 1, NaHPO$_4$ 1, NaHCO$_3$ 26 and glucose 10 (Sigma Aldrich). The ACSF was continuously oxygenated with 95% O$_2$, 5% CO$_2$ to maintain physiological pH. Coronal 250 mm cingulate cortex slices were cut at 4 °C with a vibratome (DSK, Kyoto, Japan) and placed in a chamber containing oxygenated ACSF to recover for at least 1 h at room temperature.

All recordings were performed on slices submerged in warmed ACSF (30–32 °C) and perfused (1 ml min$^{-1}$) with the same solution in the recording chamber under the microscope.

Visually identified GFP-expressing microglial cells were patched in whole-cell configuration in the cingulate cortex. Micropipettes (4–5 MΩ) were usually filled with solution containing the following composition (in mM): KCl 140, EGTA 0.5, MgCl2 2, HEPES 10, and Mg-ATP 2 (pH 7.3 adjusted with KOH, osmolarity 290 mOsm; Sigma Aldrich). Voltage-clamp recordings were performed using an Axopatch 200A amplifier (Molecular Devices). Currents were filtered at 2 kHz, digitized (10 kHz) and collected using Clampex 10 (Molecular Devices); the analysis was performed off-line using Clampfit 10 (Molecular Devices). Slicing procedure might activate microglial cells especially near the surface of the slice, whereby recordings were performed on deep cells. The current/voltage (I/V) relationship of each cell was determined applying ramp protocol from -120 to +50 mV in 500 msec every 10–30 s after whole-cell configuration was achieved (HP = − 70 mV between steps). Resting membrane potential and membrane capacitance were measured at start of recording.

Capsaicin was first dissolved in DMSO and then in ACSF solution at a final concentration of 1 µM and applied in bath for 3–10 min.

One to four cells per mice were recorded. At least four animals per group were used.

**Immunofluorescence from fixed cortex, spinal cord and dorsal root ganglion.** Both young (P21-24) and adult (P80-90) naïve and CCI C57BL6J/$Trpv1^{-/-}$ mice were sacrificed with lethal dose of carbon dioxide and immediately underwent perfusion procedure. Blood was firstly transcardially washed out with cold phosphate buffer 0.9% saline solution (PBS), and tissues were fixed by cold 4% paraphormaldehyde in 0.1 M pH 7.4 phosphate buffer (PB) with a peristaltic pump. Brains and spinal cords were dissected, postfixed for 18–22 h at 4 °C, washed from paraphormaldehyde with PB and cryoprotected in 30% sucrose/PB at 4 °C.

Dry ice frozen brains were cut into 40 µm coronal sections with a cryostat microtome (Leica Microsystems) at − 20 °C, including anterior cingulated cortex (ACC, Figs 22–28 of mouse brain atlas, Franklin e Paxinos, 2001). Frozen lumbar spinal cord (L4-L5) was cut into 40 µm sections. Dorsal root ganglia were dissected from not perfused animals, fixed by 4% paraphormaldehyde solution, cryoprotected in sucrose solution and cut in 30 µm sections.

Sections were rinsed for tree time in PB and incubated with a mix of primary antibodies in PB 0.3% Triton X-100 (Applichem, BioChemica, Darmstadt, Germany) overnight at room temperature. Primary antibody incubation step was followed by three 10-min rinses in PB at RT. Afterwards, sections were incubated for 2 h at RT in a mix of secondary antibodies, followed by three 10-min rinses in PB. DAPI was applied for 5 min in the second rinse, dissolved in PB solution.

To exclude non-specific signals of secondary antibodies, sections from each group of animals have also been stained with secondary antibody alone, following the same experimental procedure but omitting the primary antibodies.

Two or three cortical sections per mice were analysed and 'n' represents the number of mice used for each experiment.

**Histology and immunofluorescence from fresh cortical slices.** Adult (P80-90) C57BL6 and B6.129S4-$Trpv1^{tm1Jul}$/J (TRPV1$^{-/-}$) mice were sacrificed after anesthesia with isoflurane and the brain was immediately dissected. 300 µm thickness coronal slices containing ACC were cut with a vibratome (Leica Microsystems) in oxygenated ACSF. Slices were then kept at 32 °C in a submerged chamber containing ACSF equilibrated with a mixture of 5% CO$_2$ and 95% O$_2$. For the pharmacological treatments, slices were transferred in a submerged chamber containing ACSF (control), ACSF plus DMSO (vehicle), capsaicin (1 µM) or LPS (500 ng ml$^{-1}$) for 10 min (same time as used for electrophysiological recordings) and immediately after, fixed in 4% paraphormaldehyde,0.1 M phosphate buffer (PB) for 2 h at room temperature. After the fixing, slices were cryoprotected in 30% sucrose/PB and resectioned into 40 µm coronal sections with a cryostat microtome, following the same procedure as for fixed slices.

**Antibodies.** The following primary antibodies were used: rabbit anti-TRPV1 (1:100, AB800-85 Immunological Sciences Rome, Italy), mouse anti-TRPV1 (1:100, 1:1,000 in western blot assays, MAB5568 Millipore Bioscience Research Reagents, USA), rabbit anti-TRPV1 (1:500, RA14113 Neuromics, Edina, MN), rabbit or mouse anti-NeuN (1:1,000, 1:500 in WB assays ABN78/MAB377 Millipore Bioscience Research Reagents, USA), rabbit anti-Neurofilament 200 (NF200 1:300, N4142 Sigma Aldrich, St Louis, USA), rabbit anti-Calcitonin Gene Related Peptide (CGRP 1:300, PC205L Millipore Bioscience Research Reagent, USA), guinea pig anti-glial glutamate transporter-1 (EAAT2, GLT1 1:300, AB1783 Millipore Bioscience Research Reagents, USA), rabbit anti-Glast (EAAT1 1:1,200, AB85863 Abcam, Cambridge, UK ), goat anti-neuronal Glutamate Transporter (EAAC1 1:100, AB1520 Millipore Bioscience Research Reagents, USA), rabbit anti-Glutamate Decarboxylase 65&67 (1:300 AB1511 Millipore Bioscience Research Reagents, USA), rabbit anti-CAMKII (1:300 AB52476 Abcam, Cambridge, UK), Isolectin IB4 Biotin conjugates (IB4 1:200, I21414 ThermoFisher Scientific, MA, USA), rabbit anti-Iba1 (1:500 in both IF and WB assays, 019-19741 Wako, Osaka, Japan), rat anti-CD11b (1:200, MCA711 Serotec; Kidlington, UK),

goat anti-Glutamate Transporter (EAAC1, 1:100, AB1520 Millipore Bioscience Research Reagents, USA). For nuclear staining, 4,6′ diamidino2-phenylindole dihydrochloride (DAPI,1:2,000, D1306 Life Technologies, Carlsbad, California) was used.

Secondary Cy3, DyLight488 and DyLight 649-conjugated antibodies were used (1:200, Jackson ImmunoResearch, West Grove, PA, USA).

**Acquisition and analysis of images.** Double-immunofluorescence and triple-fluorescence labelling was examined with a confocal laser scanning microscope (Leica SP5, Leica Microsystems, Wetzlar, Germany) equipped with four laser lines: violet diode emitting at 405 nm, argon emitting at 488 nm, and helium/neon emitting at 543 nm and 633. Confocal images were acquired through the 40 × /63 × objective at the 1 zoom factor. In the case of magnifications, zoom factor was raised to focus only on the area of interest.

Points of colocalization were supposed when a merging area in the same cell was evident, showing a yellow resulting colour from the overlap of two green-red signals, and they were verified by magnifications and analysis on the z axes with 1–2 microns stacks.

Quantitative data from images were obtained keeping the following image acquisition criteria: 40 × objective, 1024 × 1024 frame, 10 hz acquisition frequency, pinhole 1 airy unit, lasers at 50%.

For analysis of the area occupied by the signal, confocal acquisitions of ACC for each hemisphere were taken comprehending area covered by the 40x objective in both naïve and mice suffering from neuropathic pain (two for ipsilateral and two for contralateral ACC), to have an overview of the first cortical layers (layers 2–3), and a second acquisition including the more internal layers (layer five). Since there were no significant differences between medial and internal layers (measured for each couple of layers for any group, t Test $P > 0.05$), quantifications were cumulated so as to obtain an evaluation of the antibody expression in the whole ACC.

Each image has been analysed with Image J free software (National Institutes of Health, US; http://imagej.nih.gov/) with the following procedure: for each single image the background was subtracted in a same proportion for each group, in a range between 10–50% on respect to the specific antibody. Images were reduced at 8 bit from the original RGB acquisition and a threshold was established to obtain a digital image where neither original signal disappeared and nor new signal was created. These images were analysed with the command 'analysis of particles', setting a filter to the size of signal and measuring in micrometers. Data were expressed as Area Fraction, the percentage of the total image area covered by the measured object. To establish the percentage of neurons expressing TRPV1, or both TRPV1 and EAAC1 marker, images in double and triple staining were acquired. Cells immunopositive for NeuN, TRPV1 and EAAC1 were counted, in the merging images, by the use of non-automatic cell counting method (Leica Microsystems software). Data are presented as the percentage of means.

Colocalization was intended as two proteins that occupy the same volume of interest.

The amount of colocalization was evaluated by using the Pearson statistical index of correlation (r) between two variables. This analysis describes the effect size as the strength of the linear correlation between the two variables (Evans,[34]: 0–0.19 'very weak', 0.20–0.39 'weak', 0.40–0.59 'moderate', 0.60–0.79 'strong', 0.80–1.0 'very strong').

For colocalization analysis, a double fluorescence image (red and green dyes) was splitted in RedGreenBlu channels. Background was subtracted from the red and the green channels (8 bit each) and they were analysed for correlation using the colocalization plugin with Pearson and Manders index. Pearson index was chosen for its consistency and lack of sensibility to the picture background.

For each colocalization analysis, 256 × 256 Red-green scatterplots were generated by the Image J program: the intensity of red pixels is used as the x-coordinate whereas the intensity of the green pixels as the y-coordinate.

**Morphometric analyses.** Morphometric analysis were performed on a high resolution image acquisition criteria: 40 × objective, 2048 × 2048 frame, 200 hz acquisition frequency 3X average scan, pinhole 1 airy unit, lasers at 50%.

Although two 40x objective confocal acquisitions were acquired to obtain an overview of ACC (one for cortex layers 2/3 and one for layer five) only the first layers acquisition (L2/3) was chosen for morphometrical analysis.

Each image has been analysed with plugin 'Shape descriptor 1u' of Image J software (http://imagej.nih.gov/plugins) with the following procedure: images were reduced at 8 bit from the original RGB acquisition and a threshold was established to obtain a digital image where neither the original signal disappeared and nor new signal was created. The plugin setting was fixed at a minimum of 200 µm of size and analysis was performed automatically on cell silhouettes.

Transformation index (TI) was used as an index of microglia morphology[86,87] and calculated by the equation: [perimeter of cell (µm)]$^2$/4П [cell area (µm$^2$)]. Since TI is dependent on cell shape but independent of cell size, this latter crucial for the identification of the three different microglia activation states (bushy, hypertrophied and ameboid), we exploit this parameter to obtain an unbiased analyses and avoid negative falses. The area/TI ratio was calculated for each cell and a scale value was set to obtain 4 group of cells (Supplementary Fig. 1a): RESTING cells (values of area:TI ranging from 1 to 25); HYPERTROPHIED cells

(values of area/TI ranging from 25 to 35); BUSHY cells (values of area/TI ranging from 35 to 60); AMEBOID cells (values of area/TI up to 60). The percentage of cell ratio was calculated by dividing the number of a specific cell group (i.e., resting, bushy, hypertrophied and amoeboid) for the total cell number of each experimental condition (control, vehicle, capsaicin/lps, in both wt and TRPV1$^{-/-}$ tissue).

The frequency of cells in each category was statistically quantified by the using of Fisher's exact test and performed on raw data.

**Western blotting.** Western blotting was performed according to standard procedures. Briefly, isolated tissues (10 mg) and cells ($3 \times 10^5$) were resuspended in lysis buffer (50 mM Tris-HCl, pH 7.4, 150 mM NaCl, 5 mM MgCl$_2$, 1 mM EGTA, 1% (v/v) Nonidet P-40, 10 µg ml$^{-1}$ aprotinin, 10 µg ml$^{-1}$ leupeptin, 1 mM PMSF), homogenized with a glass-Teflon homogenizer and/or sonicated before centrifugation. Proteins in the lysates were quantified by the Bradford assay kit (Bio-Rad, Hercules, CA, USA), separated by SDS-PAGE (10%) and transferred to nitrocellulose membranes. Blots were then incubated with primary antibodies recognizing TRPV1 (diluted 1:1,000), NeuN (diluted 1:500), Iba1 (1:500), or actin (diluted 1:100,000). After incubation with the appropriate horseradish peroxidase-conjugated antibody, blots were developed using an enhanced chemiluminescence detection system (Luminata Crescendo Western HRP substrate, Merck Millipore, Darmstadt, Germany). Scanning densitometry on the developed film was conducted at 600 dpi using a CanoScan LiDE 210 (Canon, Tokyo, Japan) and analysed using Image J 1.49v program (http://imagej.nih.gov/ij).

Images have been cropped for presentation. Full-size images are presents in Supplementary Fig. 18.

**Isolation of neuronal and microglial cells and flow cytometry analysis.** Cerebral cortex from 26 C57Bl6J and 20 TRPV1$^{-/-}$ mice (60–70 days old) were dissociated to single-cell suspension by enzymatic degradation using neural tissue dissociation kit from MACS Technology (Miltenyi Biotec) according to the manufacturer's protocol. Briefly, cortical tissues were weighed before mincing, a pre-warmed enzyme (Papain) mix added to the tissue pieces, and incubated under slow, continuous rotation at 37 °C on a MACSMix Tube Rotator. The tissue was further mechanically dissociated by trituration and the suspension was applied to a 70-micron cell strainer. Myelin was removed using Myelin Removal Beads II kit on autoMACS Pro Separator. Cells were processed immediately for MACS MicroBead separations. To separate primary microglia, The CD11b-positive cells were magnetically labelled with CD11b (Microglia) MicroBeads and isolated on autoMACS Pro Separator. The negative fraction was further processed with Neuron Isolation kit by depletion of non-neuronal cells and neurons were obtained from the unlabelled cells running through autoMACS Pro Separator. Both cells populations were fixed with 4% paraformaldehyde (PFA) for 10 min and then stained first with mouse primary TRPV1 antibody and then with anti-mouse Alexa633 secondary antibody. Subsequently neuronal cells were stained with mouse primary antibody NeuN and then with anti-mouse Alexa488 secondary antibody, while microglial cells with PE-Cy7-conjugated CD11b. For microglial activation experiments, $2 \times 10^5$ cells were left untreated or stimulated with Capsaicin (1 µM) for 10 min, washed and then left in complete medium in presence of Brefeldin A (10 µg ml$^{-1}$) to block cytokine exocytosis. At the end of the incubation, cells were stained at the cell surface with CD11b, fixed with 4% PFA for 10 min, and then stained intracellularly with APC-conjugated TNF-α and PE-conjugated IL-10 antibodies in 0.5% saponin for 20 min. Cells were acquired using FACSCyan ADP (Beckman Coulter) flow cytometer and analysed using Flowjo software (TreeStar, Ashland, OR, USA).

**qRT-PCR.** Total RNA was extracted by previously isolated mouse neurons and microglial cells as well as from spinal cord, using ReliaprepTM RNA Cell Miniprep System (Promega). cDNA reverse transcription was performed by means of Superscript VILO cDNA Synthesis Kit (ThermoFisher Scientific), according to manufacturer instructions. The following program was used for the quantitative RT-PCR: 25 °C for 10 min, 42 °C for 50 min, 85 °C for 5 min, then after addition of 0.1 unit/ml of Escherichia coli RNase H, the product was incubated at 37 °C for 20 min. The target transcripts were amplified by means of an ABI PRISM 7700 sequence detector system (Applied Biosystems, Foster City, CA), using the following specific primers for TRPV1 and β-actin: mouse TRPV1 F1 (5'- CATCC TCCTGCTCAACATGC-3') and R1 (5'- GCCTTCCTCATGCACTTCAG-3') and mouse β-actin F1 (5'- TGTTACCAACTGGGACGA-3') and R1 (5'- GTCTCAAA CATGGATCTGGGTC-3'). All assays were performed in duplicate, and for each well we used 10 ng of sample template cDNA. Assays were run on Roche Lightcycler 480 Real-Time PCR System. 20 µl assays were prepared for each sample as it follows: 5 µl of cDNA product, 10 µl of SYBR Green master mix (Roche, cat # 04707516001), 2 µl of each primer (4 µM, Sigma) and 3 µl of PCR-grade water. The following PCR program was used: 5 min of pre-incubation at 95 °C, followed by 40 amplification cycles at 95 °C for 10 s, 61 °C for 20 s, and 72 °C for 7 s. All data were obtained using automatic detection of TRPV1 gene Ct normalized with β-actin; for each sample, TRPV1 relative expression was evaluated as fold increase of gene expression compared with β-actin ($2^{\Delta Ct}$, with $\Delta Ct = Ct$ (TPRV1) –Ct (β-actin)). The specificity of PCR reactions was evaluated at the end of each experiment by

analyzing melting curves for each sample and, later, by running the assays in a 2% agarose gel.

**Microglia cell cultures.** Primary cultures of cortical glial cells were obtained from 2-day-old C57BL6 mice and killed by decapitation according to Directive 2010/63/EU and as describe in Levi et al. Cells were plated at low density ($3 \times 10^6$ cells per 90-mm dish) and cultured in basal Eagle's medium (BME), supplemented with 10% heat-inactivated fetal calf serum (FCS) and 5 mM KCl for 20 days. Microglial cells were obtained by dish gentle shaking, collected by centrifugation and reseeded. This protocol produced cultures with >90% microglial cells, as verified by immunofluorescence with monoclonal antibodies for the specific markers ED1 and iba1.

**Quantification of EVs.** Nanosight measurement. Microglia were exposed to serum free culture medium for 10 min, to quantify constitutive EV production. The medium was collected, the cells kept for about 5 h in complete medium, and then re-exposed to serum free medium added with 300 nM capsaicin or 1 mM ATP (positive control) for 10 min, to measure EV production upon TRPV1 stimulation. Conditioned media were pre-cleared from cells and debris at 300 g for 10 min (twice) and the number and dimension of MVs and exosomes were analysed with Nanosight LM10-HS system configured with a 405 nm laser and EMCDD camera (Hamamatsu Photonics). Videos were collected and analysed using the NTA-software (version 2.3), with the minimal expected particle size, minimum track length, and blur setting, all set to automatic. Camera shutter speed was fixed at 20.01 ms and camera gain was set to 350. Ambient temperature was ranging from 25 to 28 °C. Five recordings of 30 s were performed for each sample. EV production under capsaicin exposure was normalized to constitutive EV release from the same Petri dishes, to minimize possible variability in the density of donor microglia.

Spectrophotometric quantification of shed MVs. Microglia were incubated with 5 µM NBD-C6_HPC, washed with Kreb's Ringer solution, and stimulated with ATP or capsaicin. Supernatants were collected, centrifuged 10 min 300 g 4 °C to remove cells and debris, and then total fluorescence was assayed at 463/536 nm with a Tecan Infinite500 spectrophotometric system (Tecan, Group Ltd, Switzerland). Constitutive or evoked MV shedding was quantified from at least three distinct Petri/condition in each experiment.

**Statistics.** All data were expressed as means ± s.e.m. For immunofluorescence experiments, significance was tested using one-way ANOVA followed by a Tukey *post hoc* test for internal significance or student *T* test in the case of two groups comparisons. For electrophysiological data, Shapiro–Wilk test was exploited to verify a normally distributed data population. We then applied parametric or non-parametric tests. For the former, we used ANOVA and Student *t* test. For non-parametric data we used Wilcoxon Signed Rank test (paired samples) and Mann–Whitney test and Kolmogorov Smirnov test (unpaired samples).

Statistical differences were considered as significant at $P < 0.05$. Statistical analysis was performed by using Origin 8.1 software (OriginLab Corporation).

**Data availability.** The data that support the findings of this study are available from the corresponding author upon request.

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

## Acknowledgements

The authors are grateful to N. Berretta and G. Marsicano for critical reading the manuscript. M. Matteoli for helpful discussion and G. Martina for support in electrophysiological recordings. M.T. Ciotti for making microglia cell cultures, D. Pizzirani for providing ARN14988 and P.A. Heppenstall for IB4. R Brandi for helping in DRG dissections, C. Giorgi for oligo selections. We are grateful to A. Cattaneo for his insights and precious contribution on the revised version of this manuscript. This study was supported by grants from the Italian Minister of Health (Young Investigator grant) and the European Community's Seventh Framework Programme (FP7) under grant agreement number 603191, the PAINCAGE project to S.M. The work performed in M. Maccarrone's laboratory was supported by the Italian Ministry of Education, University and Research (MIUR) under PRIN 2010-2011 grant. Italian Research Council (Framework Agreement EBRI-CNR, 2015-2017).

## Author contributions

M.C.M. and M.M. performed ISH experiments and analysed the data; M.G. and A.M. performed electrophysiological experiments and analysed the data; S.M. and M.C.M. made the murine model of neuropathic pain; L.R. and M.L. performed EV isolation and measurements; V.C. and A.L. isolated neurons and microglia from cerebral cortexes, performed qRT-PCR and flow cytometry and analysed data; A.T. and S.O. performed immunochemical analysis; E.M. performed CX3CR1$^{+/GFP}$ experiments and analysed the data; D.R. supervised CX3CR1$^{+/GFP}$ experiments and revised the manuscript; D.P. provided reagents; M.M. supervised immunochemical experiments and revised the manuscript; C.V. contributed to design the study. S.M conceived, designed and supervised the whole study. S.M. wrote the manuscript.

## Additional information

**Competing interests:** The authors declare no competing financial interests.

