## [Peer Review File · Nature Communications]

Reviewers' comments:

Reviewer #1 (Remarks to the Author):

In this article, Marrone et al. provide novel evidence that TRPV1 is expressed in the brain, in the ACC and other brain regions, by microglial cells in basal conditions, and by neurons following peripheral nerve injury. The authors' initial observation of TRPV1 expression in the ACC relies on the use of a monoclonal antibody, which immunoreactivity is lost in TRPV1 knockout mice. The authors additionally provide functional evidence of TRPV1 expression as they show that capsaicin increases mEPSC frequency in ACC pyramidal neurons in slices from wildtype mice, an effect that is lost in the presence of TRPV1 antagonist IRTX or in TRPV1 knockout mice. The authors found that minocycline prevents capsaicin-induced increases in mEPSC frequency, suggesting that microglia activation is required for TRPV1 modulation of synaptic transmission. Mechanistically, they further propose that TRPV1 stimulation alters the morphology of microglia, which switches from resting and bushy morphologies to hypertrophied morphology, and promotes the production of microglial ectosomes. Finally the authors report that TRPV1 expression in ACC neurons following CCI modestly increased capsaicin - evoked mEPSC amplitude and charge transfer.

The results are novel, provocative, and of great interest to the neuroscience community. But TRPV1 expression and function in brain is highly controversial as several studies on the contrary reported very limited expression in brain (in this case expression is generally observed in neurons, rather than glial cells) using other antibodies or reporter mice (for example Cavanaugh et al., 2011). Thus the results need to be watertight and the controversy regarding TRPV1 expression addressed directly, with an explanation reconciling contradicting results in the literature and in this manuscript. Additionally, the molecular mechanisms by which TRPV1 modulate microglia function, and the functional consequence on somatosensation and pain remain obscure.

Major comments

-What is the staining pattern obtained in DRG and spinal cord in wildtype and TRPV1 knockout mice with the monoclonal antibody used? If DRG neurons are labelled, do immunoreactive DRG neurons respond to capsaicin?

-Which TRPV1 splice variants are recognized by the different antibody used (mono - and polyclonal in table 1a). What mechanisms underlie TRPV1 expression in the ACC, i.e. what transcription factors, are these mechanisms absent in TRPV1-negative microglia or neurons in brain regions where the channel is not expressed?

-Additional approaches should be used to confirm TRPV1 expression in ACC microglia and neurons, in particular at the mRNA level, such as RNA -seq, in situ hybridization or qPCR.

-It is well established that culturing microglia alters gene expression. Consequently in vitro studies on TRPV1 expression microglia expression should use acutely isolated purified microglia, not cultured microglia.

-The mechanisms by which TRPV1 modulates microglia activation and functions in these cells remain obscure. What signaling mechanisms are involved? Beside a change in morphology, and ectosome production, what are the consequences of TRPV1 activation on microglia function i.e. proliferation, migration, cytokine release? What do ectosomes contain? What explains the activation of microglia in basal conditions in TRPV1 knockout mice? How does this novel observation impact previous conclusions drawn on TRPV1 function in pain from the phenotypic analysis of TRPV1 knockout mice? What is the impact of microglial TRPV1 function on somatosensation and pain? How can TRPV1 KO and

capsaicin both result in microglia activation? Specific deletion or rescue of TRPV1 in microglia in adult mice would greatly help addressing these questions.

-The authors suggest that LPA is endogenous activator of TRPV1: what is the source of LPA, how/when is it produced/released to modulate TRPV1 function in glia?

- The authors also report TRPV1 expression in astrocytes: what is the function of TRPV1 in these cells? Do astrocytic TRPV1 also regulate neuronal function?

-How does peripheral nerve injury result in TRPV1 change in expression in ACC? Does peripheral inflammation also alter TRPV1-expression pattern?

-What is the physiological consequence of TRPV1 expression in ACC neurons in the absence of capsaicin?

-What is the identity of ACC neurons that express TRPV1, and in what proportion? Histological studies should clarify this point (i.e. using NeuN and markers of cortical neurons). Based on the images it seems that the majority of ACC neurons express the channel. If they are pyramidal neurons as proposed by the authors, where do these cells project and what is their function in pain processing?

Minor comments.

- There are many typos throughout the manuscript, careful proofreading is necessary. e.g. Fig 6. Hypertrophied, not ipertrophied. Fig 4d is referred to as figure 5d in the results.

Reviewer #2 (Remarks to the Author):

Summary of key results:

In this manuscript, Drs Marrone and colleagues first use immunohistochemistry to show that TRPV1 is expressed in microglia and electrophysiological recording to show that it is functional. In a second part of the manuscript, they show that upon CCI, TRPV1 is also expressed by neurons. Then the authors show that application of Capsaicin on acute slices of ACC increases the frequency of mEPSC. In another section of the manuscript, the authors show that capsaicin stimulation of cultured microglia promotes the production of micro vesicles. They also show that blocking the production of neuronal sphingosine prevents the effect of capsaicin on minis. The authors then show that microglia undergo similar morphological changes upon chronic TRPV1 loss-of-function or acute Capsaicin stimulation. Finally, the authors examine the neuronal excitability of neurons and show that upon CCI, the electrophysiological responses of neurons to capsaicin is greatly modified as compared to control.

Originality and interest :

It is of great interest to understand how microglia and inflammation modulate the brain physiology. Here, the authors demonstrate that TRPV1 is specifically expressed by microglia in the healthy brain and therefore that the effects of capsaicin are mediated by the cells (see below, the remark on the astrocytic expression). They further characterize some neuronal consequences of TRPV1 stimulation by capsaicin. Such experiments could be the first steps to demonstrate the involvement of microglial TRPV1 in the physio- pathology of the brain. However, at this stage, the manuscript is more a juxtaposition of related stories than an homogenous study. Of note, each of these stories, if properly conducted and demonstrated would certainly be of the highest interest.

Data & methodology:

The methodology is appropriate and the data presented are of sufficient quality. However, the

presentation is sometime confusing, mostly because the different stories are intermingled, but also because of the number of supplementary figures is too big and most of them show the same kind of information. Supp fig 5-7 might be fused. This is minor, but the supplementary figures might be easier to read if they share the organization of the principal figures (TRPV immunoreactivity consistently shown in the green channel and displayed in the upper rows)

Appropriate use of statistics:

It is not clear from figure 4d if LPA has an effect on its own or if the effect is only significant when compared with the BrPLPA condition. In that case, it seems difficult to conclude that "LPA is a potential endogenous activator of TRPV1 in the brain" (page 9 last sentence).

Conclusions:

One of the conclusion of this manuscript is that "Capsaicin increases (...) synaptic activity by promoting shedding of microglial MVs which in turn fosters sphingosine metabolism in neurons and enhances presynaptic probability releases". However, the data do not support such hypothesis. The authors convincingly show that capsaicin increases the frequency of EPSC in acute ACC slices and that this increase is prevented by inhibiting the acid ceramidase. In another series of experiments, the authors show that TRPV1 stimulation of cultured microglia promotes the production of MV, which are known from a paper previously published by some of the authors, to increase the EPSC frequency of cultured neurons by a sphingosine dependent mechanism. These correlations however do not support the conclusion raised by the authors.

Suggested improvements:

Each of these stories, if properly demonstrated, could be a very interesting piece of data on its own. The authors could focus the manuscript either on the consequences of TRPV1 expression in neurons upon CCI or on the involvement of microglial TRPV1 in the control of mEPSC frequency.

The authors convincingly characterize the morphological consequences of TRPV1 stimulation by capsaicin (figure 6). Therefore, they should stick to morphological description of microglia (ramified, bushy etc) and avoid describing microglia as being "activated", "resting" or in a "surveillance status" (which correspond to functional description).

The authors could be more precise on the TRPV1 expression in astrocytes. Visual analysis of the merged panel of supplementary figure 5, suggests most of the red -labelled cells (astrocytes) of the hippocampus parenchyma also display green labelling (TRPV immunoreactivity). The same conclusion can be drawn from sup figure 7a and b. Yet, the authors write that "TRPV1 is expressed at lower levels in astrocytes than in microglial cells", which is unclear. Do all astrocytes express low level of TRPV1 or do few astrocytes express strong level of TRPV1 ? The interpretation of the effect of capsaicin application would be very different depending on the expression by astrocytes.

According to the experimental procedures (page 25 and 30), Capsaicin treatment is applied for 10 min. However, according to supp figure 4d, the effect of capsaicin application on microglia is very transient (2 min) and the electrical properties of microglia returns to basal level after 10 min. The authors should at least propose an explanation to this apparent paradox

The labeling pattern of the spinal cord shown in supp fig 7 is puzzling. According to panel h, a large proportion of neurons are labelled. Yet, almost no labelling is detected in panel b and e even though dapi staining suggests that neurons are most present in the section. The size of the neurons shown in supp 7h suggests that these are motoneurons. Yet, the authors indicate that images have been taken in the dorsal horn, in which there is no motoneurons. Authors might clarify this issue.

Minor :

The authors should use the same scale for the y axis when describing the mEPSC frequencies and amplitudes. This could help the reader to compare the panels within a figure

In figure 1, the authors should add yellow arrows in the "NeuN" panel (and not only in the TrpV1 channel).

Legend of figure 5d : the capsaicin (not capsin) line is in red not dotted

Typos:

Iperrophied is not english.

p9 para3 "(, Fig.4b)"

p16 "...mice do not undergo to further activation".

Legend of supp figure 4: the concentration of capsaicin cannot be read.

Reviewer #3 (Remarks to the Author):

The role of TRPV1 in the brain is intriguing. In the study by Marrone and colleagues, it is concluded that TRPV1 is mainly expressed in microglial cells under normal conditions, and that stimulation of TRPV1 indirectly enhances glutamatergic transmission. However, in the case of chronic pain, neuronal TRPV1 is proposed to play a more important role in the regulation of neuronal signalling. The immunohistochemical characterization is convincing but not the link between microglia to neuron communication.

Several studies, not quoted, have raised the possibility that TRPV1 in PAG is of importance in pain processing. Why was this region not included in the study?

Is the capsaicin-induced increase in mEPSC blocked by glutamate receptor antagonists?

What is the effect of minocycline on heterologously expressed TRPV1?

What is the effect of LPS on TRPV1?

p. 17, 1st sentence: "...ones pain became chronic." It would be good to show corresponding behavioural data as graphs.

Most of the electrophysiology data from slices display tiny differences with overlapping SEM values. Even though significant, it is difficult to appreciate what the physiological impact would be caused by such small changes in mEPSC. For example, the effect of LPA is not convincing in Figure 4c (is BrPLPA different from ctrl? +LPA is unlikely different from ctrl?, what statistical method was used?). Also, with regard to the mentioned study on FAAH and microglia function (p. 16), other endovanilloids would have been more relevant to study. Taken together, statistics are not clearly described. Each statistical test should be clearly stated in the text including figure legends. Statistical significance is normally $*P < 0.05$, $**P < 0.01$ and $***P < 0.001$, but not in legend to Figure 4 and 7 where $**p < 0.001$. Also $*p \leq 0.05$ in figure legend 8 is not informative.

Ref 60 is incomplete.

Supplementary Figure 12. This figure should be moved to the main text as it summarizes very well the authors' thoughts.

Summary, last sentence: TRPV1 is already identified as a detector of harmful stimuli.

Discussion, last sentence: How could TRPV1 be a potential biomarker? By removing brain tissues from humans?

Reviewer #1 (Remarks to the Author):

In this article, Marrone et al. provide novel evidence that TRPV1 is expressed in the brain, in the ACC and other brain regions, by microglial cells in basal conditions, and by neurons following peripheral nerve injury. The authors' initial observation of TRPV1 expression in the ACC relies on the use of a monoclonal antibody, which immunoreactivity is lost in TRPV1 knockout mice. The authors additionally provide functional evidence of TRPV1 expression as they show that capsaicin increases mEPSC frequency in ACC pyramidal neurons in slices from wildtype mice, an effect that is lost in the presence of TRPV1 antagonist IRTX or in TRPV1 knockout mice. The authors found that minocycline prevents capsaicin-induced increases in mEPSC frequency, suggesting that microglia activation is required for TRPV1 modulation of synaptic transmission. Mechanistically, they further propose that TRPV1 stimulation alters the morphology of microglia, which switches from resting and bushy morphologies to hypertrophied morphology, and promotes the production of microglial ectosomes. Finally the authors report that TRPV1 expression in ACC neurons following CCI modestly increased capsaicin-evoked mEPSC amplitude and charge transfer.

The results are novel, provocative, and of great interest to the neuroscience community. But TRPV1 expression and function in brain is highly controversial as several studies on the contrary reported very limited expression in brain (in this case expression is generally observed in neurons, rather than glial cells) using other antibodies or reporter mice (for example Cavanaugh et al., 2011). Thus the results need to be watertight and the controversy regarding TRPV1 expression addressed directly, with an explanation reconciling contradicting results in the literature and in this manuscript. Additionally, the molecular mechanisms by which TRPV1 modulate microglia function, and the functional consequence on somatosensation and pain remain obscure.

We are really grateful with the Reviewer for his/her comments that allowed increasing the impact of our study besides enhancing our own critical understanding of the data. As reported in “major comments”, we have performed new sets of experiments to fully address Reviewer concerns. We are enthusiastic for the new data that have been added in Results and Discussion, and illustrated in the Figures.

Concerning the contradicting results in literature and in our manuscript, **a new paragraph has been added in the Discussion**

Here below we report a more detailed explanation on this discrepancy, with particular focus on TRPV1 reporter mice.

The reporter mouse in Cavanaugh et al was generated by a gene targeting approach in which PLAP and LacZ reporter genes are expressed under the control of the endogenous TRPV1 promoter, at the TRPV1 genomic locus. In that paper, the authors use the “authority” of a purportedly genetically clean method to draw the apparently definitive conclusion that the expression of TRPV1 in the brain is very low and is restricted to a very limited set of brain regions. However, as we argue below, even that method is flawless and the conclusions drawn must be qualified.

Indeed, while that approach is genetically “clean” , in terms of the genomic targeting locus, it suffers from pitfalls deriving the many levels of regulation of the mRNA transcripts, including the translational control, that may and will affect the expression

level of proteins deriving from bicistronic mRNAs. Actually, it is well known that the expression levels of proteins translated downstream of an IRES sequence (the two reporters PLAP and lacZ, in Cavanaugh et al) is much lower than the expression level of the first protein (i.e. TRPV1), translated by cap-dependent translation (for example, see Houdebine LM et al., Transgenic Res 1999; Dirks W et al., Gene 1993; Mizuguchi H et al., Mol Ther 2000).

Even more variability and reduced expression can be observed in multicistronic vectors, such as the tri-cistronic targeting vector used by Cavanaugh et al.

In line with this caveat, Cavanaugh et al observed that no PLAP reporter expression could be detected in the brain of the transgenic mice, but only lacZ expression. So, one reporter, in the same construct, did not work (PLAP), on the basis of which one would have concluded that no TRPV1 is expressed.

Thus, the expression pattern of the reporter genes does not precisely mimic the expression pattern of the TRPV1 but reflects the limits of the seemingly precise and sensitive method used. Alternative methods, such as the use of well-validated monoclonal antibodies (as performed in our study), would provide complementary information that integrates the information on the TRPV1 expression pattern.

In the second line of reporter mice described by Cavanaugh et al, the TRPV1 *Cre/R26R-lacZ* mice, the staining by the reporter was more widespread than that observed in the previous TRPV1*PLAP-nlacZ* mice and included several regions that were not observed in TRPV1*PLAP-nlacZ* mice. Indeed, the crossing of this line with Cre-dependent reporter lines provided a fate map of TRPV1 expression, which revealed all loci of TRPV1 expression, over time, no matter how transient. Because only a few molecules of Cre are necessary to induce recombination, it is possible that some of these areas represent regions where TRPV1 was expressed at low levels, below the detection threshold in the TRPV1*PLAP-nlacZ* mice.

Moreover, in Cavanaugh et al the identity of the cells stained in the brain by lacZ reporter was not investigated. Therefore their reported “neuronal staining” was not formally demonstrated and could also very well include non neuronal cells such as microglia cells. Altogether, the results described by Cavanaugh et al, mitigated by the above mentioned caveats, can be considered to be consistent with our results, that demonstrate in a direct way the expression of TRPV1 in microglia, with a well validated anti-TRPV1 monoclonal antibody, that does not react with TRPV1 knock-out sections. So, we do not see any contradiction and we believe that we have seen, with a direct approach, an expression pattern that a reporter mouse-based approach might very well have missed, for reasons explained above.

As a final note, the formal possibility remains that microglia express a splicing form of TRPV1, still to be discovered, that is not detected by the reporters described in Cavanaugh et al.

Major comments

1) What is the staining pattern obtained in DRG and spinal cord in wildtype and TRPV1 knockout mice with the monoclonal antibody used? If DRG neurons are labelled, do immunoreactive DRG neurons respond to capsaicin?

As shown in the Suppl. Fig 7 (panels a-c), the anti-TRPV1 MAb labels neurons,

astrocytes and microglia of WT spinal cord. In this revised version **we have included a fourth panel (Suppl Fig 7d)** illustrating the specificity of the MAb in spinal cord sections of TRPV1^{-/-} animals. As in brain areas, also in spinal cord sections the anti-TRPV1 MAb staining is absent in tissues from TRPV1^{-/-} mice.

As requested by the Reviewer we have carried out new immunofluorescence experiments on DRG neurons from both WT and KO mice that **are now integrated in the manuscript as Suppl. Fig.8**. Similarly to previous studies (Ji R et al., Neuron 2002) 53.8% of DRG neurons were anti-TRPV1 positive. Conversely, in ^{-/-} sections no significant staining by the anti-TRPV1 Mab was detected.

Unfortunately, we were not able to perform electrophysiological recordings from TRPV1 positive DRG neurons, due to our lack of expertise with ex-vivo preparations and recording of sensory neurons. In addition, in the period of this revision, we could not find colleagues confident with DRG neurons recordings that could help us addressing this point. We hope the reviewer will understand our difficulties in fully answering his/her request. Finally, we respectfully believe that the lack of this evidence will not weaken the main findings of our study.

2) Which TRPV1 splice variants are recognized by the different antibody used (mono- and polyclonal in table 1a).

Western blot analysis of TRPV1 expression was performed by using two antibodies against the C-terminal domain of the channel. In the lysates of ACC, the monoclonal antibody (mouse monoclonal anti-TRPV1, cod. MAB5568 from Millipore Bioscience Research) recognized three major bands: (i) one band at about 100 kDa, absent in KO, attributable to full-length channel; (ii) a second one at about 70 kDa, and (iii) a third band at about 50 kDa (Fig. SS1). Since the latter two bands were equally expressed in both wild type and TRPV1-KO mice, we considered them as non-specific bands. In the same tissue extract, the polyclonal antibody (rabbit polyclonal anti-TRPV1, cod. RA10110 from Neuromics) was not able to detect any band at 100 kDa, as well as any other specific band at lower molecular weights (Fig. SS2) and therefore it was not used in the subsequent immunoblotting analysis. Previous evidence shows that neurons were marked by this anti-TRPV1 pab (Sharif Naeini et al., Nature Neurosci 2006). To the best of our knowledge, we can conclude that, ACC tissue, microglial cells and, to a lesser extent, neurons, certainly express the full-length isoform of TRPV1. However, a more dedicated study is needed for assessing the possible co-presence of the other different splice variants of TRPV1, as well as their specific contribute in TRPV1-mediated mechanisms described in the present paper.

Fig. SS1

Fig.SS2

What mechanisms underlie TRPV1 expression in the ACC, i.e. what transcription factors, are these mechanisms absent in TRPV1-negative microglia or neurons in brain regions where the channel is not expressed?

We thank the Reviewer for this question concerning the possible mechanisms underlying the differential expression of TRPV1 in specific subsets of neuronal and non-neuronal cells in the brain.

There are two issues to be considered, to address this question: one is the mechanism behind the regional expression of TRPV1 (ACC versus other brain areas), and the second is the mechanism responsible for the differential expression of TRPV1 in microglia and neurons in the ACC, under physiological or chronic pain conditions. The first question, while interesting, is not relevant for our paper. As for the second question, that is central for our paper, our new data, included in the revised manuscript, indicate that the TRPV1 mRNA levels are similar in both neurons and microglia cells while the protein is much higher expressed in microglia than neuronal cells. Therefore, the differences in the amount expression of the protein could be ascribed to alterations in post-transcriptional regulation of gene expression. In particular to decreased translational **efficiency** or to increased protein degradation in neurons.

We think that this attractive but complex issue would require a more dedicated study,

which unfortunately is beyond the scope of this paper.

However, we would like briefly discuss this aspect, highlighting that different regulatory mechanisms may be involved, including transcriptional and post-transcriptional ones. To our knowledge, there is only one study that deals with transcription-dependent mechanism regulating the expression of TRPV1 mRNA in a specialized cell type (Chu et al., Mol Pain 2011). This study showed that, in sensory neurons, TRPV1 gene expression is dependent on Specificity protein 1 (Sp1) and Sp4 transcription factors, with Sp4 playing a critical role in activating TRPV1 transcription (Chu et al., Mol Pain 2011). Although there are no published reports focused on non-neuronal cells, it is possible that microglial cells resident in ACC possess a unique set of transcription factors, possibly including Sp1 and Sp4, for driving TRPV1 gene transcription. Thus, the availability of this specific set of transcription factors could be the molecular underpinning by which these cells dynamically regulate their levels of TRPV1 in response to cellular and environmental condition changes. Noteworthy, the similar expression of TRPV1 mRNA in neurons and microglial cells, acutely isolated from adult mouse brain (suppl Fig. 4a), strongly supports the presence of post-transcriptional mechanisms underlying the differential expression of TRPV1 that we documented in the two cell populations of ACC. The possibility of translational control of TRPV1 mRNA would call naturally into play TRPV1-selective miRNAs in neurons, that might release their translational repression in an activity dependent manner, under the chronic pain conditions that we demonstrated to induce ACC neuronal expression of TRPV1.

We have added paragraphs discussing these issues in the revised Discussion

3) Additional approaches should be used to confirm TRPV1 expression in ACC microglia and neurons, in particular at the mRNA level, such as RNA-seq, in situ hybridization or qPCR.

Along with the Reviewer suggestion, we purchased the dissociation kit from MACS® Technology (Miltenyi Biotec) to acutely isolated microglial and neuronal cells from the cortex of adult mice. Subsequently, we performed RT-PCR of TRPV1 gene expression in these samples and **the new obtained results have been added in the first paragraph of the Result section and illustrated in Suppl. Fig.4a**. Both microglia and neurons express comparable mRNA transcript levels of TRPV1. As discussed above, this result opens the exciting possibility that the expression of TRPV1 in neurons is under activity-dependent (or pain-dependent) translational control, which might open a new avenue of investigation. These new experiments, together with the Flow Cytometric analyses (points 4 -5-6) were carried out by two experts on this technique so that they are now included as coauthors of this study.

4) It is well established that culturing microglia alters gene expression. Consequently in vitro studies on TRPV1 expression microglia expression should use acutely isolated purified microglia, not cultured microglia.

Accordingly with the Reviewer, we carried out flow cytometric experiments in acutely isolated microglia and neurons. The results revealed low mean fluorescence intensity levels of surface TRPV1 expression in NeuN positive cells and intense amount of this protein in CD11b positive microglial cells. **These new results are now present in the main Fig.6 and in the first paragraph of the Result section.**

5) The mechanisms by which TRPV1 modulates microglia activation and functions in these cells remain obscure. What signaling mechanisms are involved?

How can TRPV1 KO and capsaicin both result in microglia activation?

Based on our newly obtained results on cytokine production upon TRPV1 stimulation in WT and TRPV1^{-/-} mice, shown in the Fig. 6g-l, we hypothesize the following signaling mechanism:

Upon any type of injury, the increased levels of endovanilloids activate TRPV1 on microglia cells. Following stimulation of these non-selective cationic channels, calcium influx increases in microglial cells (Hassan S et al., Br J Pharmacol 2014) and activates protein kinases which, in turn, trigger transcription-factor dependent cytokine production and release (Farber K and Kettenmann H Glia 2006). Based on previous evidence and on our own data (Fig.6g-l), we postulate that TRPV1 activation stimulates Cdk5 (Rozas P et al., Pain 2016) that in turn would activate NF- κ B (Sappington RM and Calkins, DJ Invest Ophthalmol Vis Sci 2008), finally elevating TNF α (Schow SR & Joy A Cellular Immunol 1997). Interestingly, we found that microglial cells lacking of TRPV1 shift into an anti-inflammatory phenotype by expressing in basal conditions higher level of IL-10 compared to their WT counterpart (Fig.6). IL-10 production is regulated by the transcription factor IRF4 and mediates inhibition of NF- κ B (Ahyi AN et al., J Immunol 2009; Driessler F et al., Clin Exp Immunol. 2004). To note, in antiviral responses, the two transcription factors IRF and NF κ B mutually regulate each other i.e. reducing IRF expression upregulates NF- κ B transcription and *viceversa* (Rollenhagen C et al., Plos One 2015). This suggests that TRPV1 reduces IRF4 by upregulating NF- κ B, whereas its absence unblocks IRF-dependent IL-10 production (Fig.6). In this last condition, microglia is in a more protective status, thus decreasing the immunological response to pathogens. This hypothesis is supported by our evidence that the LPS-mediated effects are significantly reduced in TRPV1^{-/-} mice (Fig. 7 and Suppl Fig. 11c). Given also TRPV1 role as regulator of CD4⁺ cell functions (Bertin S et al., Nat Immunol 2014), we could predict that this channel maybe key in controlling the innate immune responses, but this remains to be seen.

This postulated mechanism, which is based on the robust experimental observation of the opposing TNF α and IL-10 production in WT and TRPV1^{-/-} microglia cells (Fig 6), leads to clear predictions that can be readily testable in future experiments.

Beside a change in morphology, and ectosome production, what are the consequences of TRPV1 activation on microglia function i.e. proliferation, migration, and cytokine release?

As reported in the previous manuscript, the activation of TRPV1 expressed by microglial cells causes:

- a) an increase of mitochondrial calcium microglia/mtROS production/MAPK activation/enhancement of microglia chemotaxis (Miyake T et al., Glia 2015);
- b) NADPH oxidase-mediated ROS generation (Schilling T & Eder C J Neuroimmunol 2009);
- c) IL-6 release and NF- κ B translocation upon elevated hydrostatic pressure in the retinal microglia (Sappington RM & Calkins DJ, Invest Ophthalmol Vis Sci 2008).

We have performed a new set of experiments to investigate microglia phenotype induced by TRPV1 activation (as already reported above). Flow Cytometry analyses shows that

acutely isolated microglia cells incubated with capsaicin for 10 minutes, and after a recovery of 4 hours, produced and released significant high level of TNF α whereas IL-10 levels remained to baseline values. **These new data are illustrated in Fig.6.**

What do ectosomes contain?

MVs secreted from inflammatory microglia were first described by our group to act as a vehicle for the secretion of the leaderless protein IL-1 β and to contain also the machinery for IL-1 β maturation (inflammasome components, Bianco F et al., 2005; 2009). More recently, we demonstrated that EVs secreted from reactive microglia also contain and transfer to recipient glial cells IL-1 β transcript (Verderio et al, 2012). In subsequent work proteomic analysis revealed that EVs secreted from microglia contain proteins involved in cellular architecture, metabolism, protein synthesis and degradation, including the hydrophobic lipid-modified protein wnt3a, which acts as morphogen during development (Hooper C et al., 2012). Among bioactive lipids, we recently found that microglial EVs contain a strike high concentration of the endocannabinoid anandamide (Gabrielli M et al., 2015) and lipid components, which promote sphingolipid metabolism in neurons (Antonucci F et al., 2012). Thus, ectosomes produced by microglia are expected to contain and deliver complex "signal" to neurons, including lipids and RNA, as described for EVs secreted by other cell types. However, to the best of our knowledge, only two proteomic studies (Potolicchio et al., 2005; Hooper C et al., 2012) have been carried out on microglia-derived EVs, and in particular on exosomes, and no lipidomic and transcriptomic data have been generated yet on exosomes/ectosomes thus limiting current knowledge of microglia-derived EV composition and complexity (for a review see Prada et al., 2015). It remains to be seen if and how the absence of TRPV1 receptor expression changes the composition of microglial EV, another lead that stems from the present work.

So far we have not added any sentences in this regard in the revised version. However if the Reviewer considers it worthwhile, we will certainly add it.

What explains the activation of microglia in basal conditions in TRPV1 knockout mice?

The revised version of the manuscript contains new data on cytokine production and release, measured by Flow cytometry and performed on fresh isolated microglia cells from WT and TRPV1 $-/-$ adult cortical tissues (Fig.6). Given the known inflammatory role of TRPV1, we hypothesized that the lack of this protein could shift microglia phenotype from M1 to M2. Among anti-inflammatory cytokines, we measured IL-10 in TRPV1 deficient microglia cells and found a threefold increase compared to the WT counterpart, while the amount of the pro-inflammatory cytokine TNF α did not differ from microglia WT. The data suggest that the activated state of TRPV1 deficient microglia in basal condition displays an anti-inflammatory phenotype. This protective or, better, refractory microglial state could, for instance, result from a less functioning capsaicin-NF- κ B-TNF α signalling. In normal conditions, this signalling would cause the opposite upregulation of the IRF4-IL-10 pathway.

What is the impact of microglial TRPV1 function on somatosensation and pain?

So far, the role of cortical microglia cells in the elaboration and perception of a nociceptive stimulus as painful has not been investigated. In addition, there are no studies using brain specific TRPV1 conditional knock out mice and therefore no direct genetic evidence of brain TRPV1 role on somatosensation signaling/acute and chronic pain. Therefore our data provide a strong motivation to derive brain microglia-specific or

central neuron-specific TRPV1 KO mice to directly address this issue in the next our work.

However, there is evidence on the role of microglia in the visual somatosensory perception. In particular, two-photon *in vivo* imaging of cortical microglia motility show that, in resting conditions, microglia dynamically interacts with dendritic spines during visual sensory experience facilitating the elimination of small and structurally dynamic spines and finally regulating neuronal spontaneous activity (for review see Tremblay ME, Neuronal Glia Biol. 2011). Although our *ex vivo* experiments on the effect of IRTX or minocycline on baseline glutamatergic transmission did not reveal a tonic modulation of neurotransmission by TRPV1s, it could be possible instead that in *in vivo* healthy brain, there may be sufficient endovanilloid tone controlling microglia surveillance behavior and hence elimination of synaptic structures.

In pain mechanism, the scenario may be different, especially in those forms of pain in which both neural and immune changes coexist, such as in nerve injured derived- and chronic inflammatory pain. The release of inflammatory factors (chemokines, cytokines, prostaglandins, glutamate, ATP, neurotrophic factors, LPA and so on), in addition to facilitate the development and the persistence of peripheral and central sensitization, signal directly to the brain via both neural and blood-borne routes (Konsman, JP et al., Trends Neurosci 2002; Barrientos RM et al., Neurobiol Aging 2006). It is possible that once in the brain, cytokines and co-partners, whose levels are sustained by local glial cells, trigger local inflammation with production of endovanilloids, activation of microglia TRPV1, release of MVs and finally modulation of neuronal transmission.

In conclusion, at the first stage of chronic pain as well as in acute pain, when TRPV1s are preferentially expressed by microglia cells (Fig 2), their function is limited to an inflammatory detection (and to an indirect modulation of neurotransmission). On the other hand, when the chronic pain became established, and the TRPV1 is also expressed by neurons (Fig 2), it directly causes cortical hyperexcitability and the increase of neuronal synaptic strength, both hallmarks of this pathology.

How does this novel observation impact previous conclusions drawn on TRPV1 function in pain from the phenotypic analysis of TRPV1 knockout mice?

Another finding of our study reveals that in physiological conditions, cortical TRPV1^{-/-} microglia is activated *per se* and results in anti-inflammatory state. Accordingly, another study reports higher immunostaining for Iba1 in the spinal cord of naïve TRPV1^{-/-} mice (Chen Y et al., Exp Neurol 2009), suggesting a microgliosis similar to our anti-inflammatory phenotype. In this condition, microglia should not be able to generate an inflammatory response or the so-called “primary innate immune response”.

Given a) show the impaired sensitivity to thermal noxious and inflammatory pain stimuli in mice lacking this channel (Caterina M et al., Science 2009; Davis JB et al., Nature 2000); b) the pivotal role of glial cell in the induction and maintenance of peripheral and central sensitization and hence in pain hypersensitivity and chronic pain (Tanga FY et al Neurochem Int 2004) (Milligan, E.D. and Watkins, L.R., Nat. Rev. Neurosci. 2009), c) the enhanced production of TNF α and IL-6 by microglia cells upon TRPV1 stimulation [Fig. 6d and (Sappington RM & Calkins DJ, Invest Ophthalmol Vis Sci 2008), respectively], we suggest that the lack of sensitivity to certain type of pain and of inflammatory response in mice lacking TRPV1 could be due to the persistent M2 anti-inflammatory phenotype of these animals. Perhaps these mice are able neither to generate

the inflammatory cascade responsible for peripheral and central sensitization, nor the primary defensive inflammatory response. In keeping with this, neuropathic pain is constitutively suppressed in early life by anti-inflammatory neuroimmune regulation (McKelvey R et al., J Neurosci 2015).

Indeed, our behavioral experiments (not reported in this study) showed that a) naïve TRPV1^{-/-} mice did not exhibit different mechanical threshold compare to naïve WT animals but had higher thermal threshold than WT mice; b) CCI WT mice developed allodynia whereas CCI TRPV1^{-/-} mice did not in the first three weeks from the sciatic nerve lesion; c) TRPV1 deficient mice showed a great and long lasting inflammation (edema) associated to autotomy within the third week and in most serious cases this phenomenon lead to death (see table and pictures below).

TOT OP	DEATH POST-CCI	HEAVY OEDEMA	AUTOTOMY	NOTHING
N 16	N 7 D3	N6 D3-D14	N6 D14-D21	N3

This last event could be ascribed to the persistent lack of M1 phenotype of these immunocompetent cells chronically lacking of TRPV1.

Specific deletion or rescue of TRPV1 in microglia in adult mice would greatly help addressing these questions

We agree with the reviewer that performing the rescue experiments or selective deletion of the protein in microglia of adult mice would greatly help discussing and give more impact to our finding. Indeed, our new results provide a strong motivation to derive brain microglia-specific or central neuron-specific TRPV1 KO mice. However, they require longer time than that assigned by the editor and therefore they will be included in our next study.

6) The authors suggest that LPA is endogenous activator of TRPV1: what is the source of LPA, how/when is it produced/released to modulate TRPV1 function in glia?

LPA is the major member of a family of lipid signaling molecules, the lysophospholipids, which exert its effects trough the interaction with the GPCRs, LPA receptors. LPA is generated from phosphatidylcholine through different enzymatic pathways, the major one involving the enzyme autotaxin (Yun C. Yung et al., Neuron 2015). LPARs are present in various organ systems, at both neuronal and glial levels, accountings for their diverse biological roles (for review see Mutoh T et al., Br. J. Pharmacol., 2012). The CNS is one

of the biological systems markedly affected by LPA signaling, during both embryonic and adult stages. Notably, beyond being involved in diverse CNS pathologies i.e. ischemic stroke, seizures, neuropsychiatric and development disorders, LPA, via activation of its receptors, play a central role in triggering and maintaining neuropathic pain (Velasco M et al., *Neuropharmacology* 2016). LPA has been shown to increase inflammation and glial cell proliferation (Goldshmit Y et al., *Am. J. Pathol.* 2012) LPA-mediated stimulation of macrophages/microglia causes a self-sustaining feedforward loop of LPA production within macrophages/ microglia, which can be inhibited with minocycline (Ma L. et al., *Mol. Pain*, 2013; Uchida H et al., 2014). Moreover, intrathecal injection of mice with LPA increases the transcription of genes such as CD11b, leading to activation of microglia and morphological changes from ramified to amoeboid phenotypes. Like the endocannabinoids, LPA exerts its effects not only interacting with its metabotropic receptors LPAR₁₋₆, but also acting as agonist of TRPV1 channels (Nieto-Posadas A et al., *Nature Chemical Biology* 2011).

Altogether this evidence prompted us to investigate if, also in the brain, LPA may behave as an endogenous agonist for TRPV1. In this revised version of the manuscript, we performed experiments in which LPA were bath applied without previous blocking LPARs. In this experimental condition, in which LPA can bind to both metabotropic receptors and TRPV1, LPA induced a consistent increase of mEPSC frequency. This enhancement was much less effective when TRPV1 was block (Fig 6e). Therefore, in a more physiological/pathological context, LPA activates TRPV1 despite having available their high affinity receptors LPARs.

Interestingly, similarly to capsaicin, LPA activates potassium outward current in microglia cells (Schilling T et al., *Eur.J. of Neurosci* 2004) and induces microglial microvesicles shedding (Duc Bach Nguyen et al., *Cell Physiol Biochem* 2016).

A sentence on this point has been added in the Results and Discussion

7) The authors also report TRPV1 expression in astrocytes: what is the function of TRPV1 in these cells? Do astrocytic TRPV1 also regulate neuronal function?

As proven by Alain Bessis group, microglia stimulation by LPS tunes neurotransmission by the binding of ATP to purinergic receptors expressed by astrocytes (Pascual O et al., *PNAS* 2011). Accordingly, we also tested if stimulation of microglial TRPV1 modulates neurotransmission by recruiting astrocytes. To this aim we performed a set of experiments by testing the capsaicin response in the presence of the glial metabolic poison fluoroacetate (FAc). FAc is exclusively taken up by astrocytes and is converted in the astrocyte to fluorocitrate, which is an inhibiting substrate of the Krebs cycle enzyme aconitase and has been shown to specifically depress astrocytic function (Fonnum *et al.* 1997). It has been employed successfully in different studies to assess astrocyte functionality (for example in Henneberger C et al., *Nature* 2010 and in Andersson M et al., *J Physiol* 2007). To explore if astrocytes are also mediators of microglia-neuron communication upon TRPV1 activation, a set of experiments were carried-out by testing the effect of capsaicin in the presence of 1 mM FAc. In this experimental condition, FAc did not prevent the increase of mEPSC frequency by capsaicin suggesting that, at least in the resting conditions, astrocytes are not implicated in microglia TRPV1 modulation of neurotransmission.

These results are now included in the revised manuscript in Results, Discussion and

as Fig.5c.

Nevertheless a role of astrocytic TRPV1 in conditions of astrocytosis is well documented i.e. in Parkinson disease, ischemic injury and stress (Nam JHet al., Brain 2015; Miyanohara J et al., Biochem Biophys Res Commun 2015; Ho KV et al., Glia 2014). Therefore it may be possible that the same experiment performed under astrogliosis condition would have the opposite result.

8) How does peripheral nerve injury result in TRPV1 change in expression in ACC?

We suggest that during the chronicization of neuropathic pain, due to a sustained central sensitization and continuous production of inflammatory molecules, a series of cellular and molecular event cascade takes place, that lastly results in i) reorganization of supraspinal structures and maladaptive plasticity, ii) structural brain changes, iii) alteration in neurochemistry iv) alteration of neuronal excitation (Seifert and Maihofner Cell Mol Life Sci 2009). And it is in this phase that some neuronal and/or inflammatory factors may regulate the post-transcriptional mechanisms of the TRPV1 gene expression.

This hypothesis has been postulated on the following evidence:

- a) TRPV1s start being expressed by neurons of 1 week CCI mice and become more widespread by 4 weeks after nerve surgery (Fig.2).
- b) This expression pattern is absent in young mice (Suppl.Fig.1).
- c) Neuropathic pain is constitutively suppressed in early life by anti-inflammatory neuroimmune regulation (McKelvey R et al., J Neurosci 2015).
- d) TRPV1 mRNA is already present in neurons in physiological conditions (Suppl. Fig 4a).

We believed that the persistent immune/inflammatory component in pain chronicization is the upstream crucial step for the regulation of TRPV1 pattern expression. Notably, cytokines and inflammatory molecules peripherally produced can directly signal to brain areas via humoral and blood-borne ways thus probably directly affecting neurochemistry brain alteration (Konsman JP et al., Trend Neurosci 2002). In addition, alterations of the blood spinal cord barrier (and presumably of the blood brain barrier) following nerve injury, facilitate the influx of inflammatory mediators and the recruitment of blood borne macrophage in the CNS.

To test this hypothesis, our lab is currently testing the effect of in vivo minocycline treatment on TRPV1 cellular pattern distribution in CCI mice (Li WW et al., J Neuroimmunol 2005; Riazi K et al., J Neuroscie 2015). **A sentence on this point has been added in the discussion**

Does peripheral inflammation also alter TRPV1-expression pattern?

Based on our above reported observations, and on the similar mechanisms shared by neuropathic and inflammatory pain (Xu Q & Yaksh T Curr Opin Anaesthesiol 2011), we hypothesize that only chronic and not acute inflammatory pain may affect TRPV1-expression pattern. For example, in an acute inflammatory model that peaks at 4 days and resolve in few days (Riazi K et al., J Neuroscie 2015), we do not expect to find any change on TRPV1 cortical expression, as we start seeing it from the first week after peripheral injury. Anyway, we cannot truly answer to this question without having before performed any experiments on it.

9) What is the physiological consequence of TRPV1 expression in ACC neurons in the

absence of capsaicin?

To address this question we have analyzed both the spontaneous glutamatergic currents and the number of action potential of cortical pyramidal neurons in baseline conditions from Sham and CCI mice (after 4 weeks from the ligation of sciatic nerve). According to previous studies, the frequency of cortical spontaneous glutamatergic currents action potential independent (mEPSCs) are significantly higher in mice suffering from chronic pain (Matos SC et al., J of Neurosci 2015; Zhao M et al., J of Neurosci 2006). Actually, the increase of excitatory neurotransmission, as well as a decrease of the inhibitory one, is a relevant aspect of pain central sensitization. In contrast, we found a great reduction of mEPSC amplitude in recordings from CCI mice. This discrepancy could rise by the different time window from pain onset employed in our study. While our experimental mice have been tested at 4 weeks after the pain onset, previous studies have used animals at 1-2 weeks from pain induction (Zhao M et al., J Neurosci 2006; Xu H et al., J Neurosci 2015; Blom SM et al., J Neurosci 2015). The amplitude drop of spontaneous AMPA-mediated current could be attributed to AMPA receptor trafficking as a mechanism to maintain neuronal synaptic homeostasis following the persistent increase of synaptic activity (Turrigiano GG, Cell 2008). This compensatory mechanism can be executed by accumulation of Arc protein at synapses (Shepherd JD et al., Neuron 2006). Interestingly, Arc/Arg3 is preferentially expressed following nociception stimulation (Hossaini M et al., Mol. Pain 2010). Other event that might underlie the amplitude reduction is the alteration of VGLUT-dependent glutamate content of synaptic vesicles, which in turn modify the strength of excitatory synaptic transmission (Daniels RW et al., J Neurosci 2004). Intriguingly, reduction of VGLUT level results in a nearly abolishment of neuropathic pain (Moechars D et al., J Neurosci 2006)

The level of intrinsic neuronal excitability has been measured as the number of spikes in response to the indicated amount of current injected into PNs. As shown in the input-output curves of the new Suppl Fig 11a, the CCI caused a left shift of the input current to number of action potential curves with a significantly increase of the spike number at different range of current steps (* $p < 0.05$, Mann-Whitney test).

This data confirms previous evidence in which cortical PNs displayed higher firing frequency after 2-3 weeks from the nerve injury (Blom SM et al 2014 and Matos SC et al., J Neurosci 2015).

In order to demonstrate that TRPV1 may be one of the players responsible of this cortical hyperexcitability, current clamp recordings of PNs from CCI mice were performed in the presence of the TRPV1 antagonist IRTX (300nM). As shown in suppl. Fig 11b, we found that this toxin significantly reduced the number of spikes elicited with different amount of current (* $p < 0.05$, ** $p < 0.005$, Paired Sample Wilcoxon Signed Rank test).

These new results have been included in the Results, Discussion and as Suppl. Fig.11a-d

10) What is the identity of ACC neurons that express TRPV1, and in what proportion? Histological studies should clarify this point (i.e. using NeuN and markers of cortical neurons). Based on the images it seems that the majority of ACC neurons express the channel.

Before identifying the neuronal phenotype immunopositive for TRPV1 we evaluated the proportion of neurons expressing this protein.

Methodologically, experiments in double and triple staining were acquired with the use

of a 4-lasers confocal microscope (Leica) keeping fixed the following parameters: frame 1024x1024, 10 hz, pinhole 2 airy unit. NeuN positive cells were reckoned by the non automatic cell counting tool of Leica microscope software (LAS Lite) in order to establish the total amount of neurons in each acquisition. Neurons positive for TRPV1 and/or for TRPV1 and EAAC1 were counted in the merging image. Data are presented as the percentage of means \pm SE.

Experiments performed in double staining (anti-TRPV1 MAb and NeuN on 1 week CCI mice) showed TRPV1 neuronal expression almost in the superficial layers (L2-3) of contralateral hemisphere. Neurons immunopositive for TRPV1 were 43.84% of the total amount of neurons present in this area (n=3 from 3 mice). In 2 out of 3 experiments also ipsilateral cortex showed TRPV1 immunoreactivity in neurons and with lesser extent than the contralateral one (18.4%). Cortical neurons of contralateral deeper layers (L5-L6) were TRPV1 immunopositive (17.59%) just in 1 out of 3 experiments while ipsilateral inner layers never showed neuronal colocalization.

The analyses carried out on 4 weeks CCI mice revealed that 34.89 % of total NeuN positive cells of contralateral layer 2/3 and 28.17 % of deeper layers of ACC were also immunoreactive for TRPV1 Mab (n=7/7 and n=5/7 from 3 mice, respectively). In the ipsilateral emisphere instead, only in 1 out of 7 experiments showed TRPV1 neuronal expression in the superficial layers.

In order to characterize the phenotype of TRPV1+ neurons, we carried out triple immunostaining with NeuN (rabbit) anti-TRPV1 MAb (mouse) and the neuronal glutamate transporter EAAC1 pAb (goat). This latter was employed for two main reasons: it selectively stains the cytoplasm of principal neurons (Wernig B et al., 2004 J Neurosci) and is hosted in goat therefore was suitable for crossing with the NeuN (rb) and anti-TRPV1 (mo) abs.

In the contralateral cortex (n=3/3 mice at 4 weeks after CCI) the percentage of EAAC1 negative neurons expressing TRPV1 was 6.7%, whereas the remnant neurons were identified as glutamatergic (EAAC1 positive neurons).

The EAAC1 immunonegative neurons expressing TRPV1 could be interneurons and/or principal neurons with low affinity for this glutamatergic marker.

This hypothesis is supported by our electrophysiological results. Although the aim of our study was to characterize the response of pyramidal neurons to TRPV1 stimulation, and therefore we recorded almost PNs, we stumbled upon GABAergic cells (parvalbumine positive interneurons n=4) and found that 1 out of 4 interneuron respond to capsaicin with a change of both membrane potential and resistance.

These results have been added in Figure 3 of the revised manuscript, in the Results and Methods.

If they are pyramidal neurons as proposed by the authors, where do these cells project and what is their function in pain processing?

In a very recent Review by Tim Bliss et al., (Nature Neuroscience Review, 2016) the input to and output from the ACC and its functions are clearly described. Layer 2/3 is mainly formed by principal neurons and is the cortical layer that received inputs from the other brain areas. It in turn sends projection both to deeper layers of the cortex (L5-L6) and the contralateral ACC. Layers 5 and 6 comprise interneurons and pyramidal cells, which send projections to the superficial layers as well as to cortical output areas (amygdala, prefrontal cortex, hypothalamus, locus coeruleus, periaqueductal gray and spinal cord dorsal horn). Sensory/nociceptive information spread to ACC layer 2/3 from

three different pathways: spino-thalamic, parabrachial–amygdala and insular cortex-somatosensory cortex tracts.

Interestingly, contralateral layer 2/3 pyramidal cells, the input cortical neurons of the spino-thalamus-cortical tract, are those neurons that in our study functionally expressed TRPV1 in the first week after the sciatic nerve injury and undergo to both morphological and synaptic transmission properties changes (Metz AE et al., PNAS 2009). In turn, they send afferents to layer 5/6 neurons, the output layer of the cortex. Our data show that also layer 5/6 pyramidal cells expressed TRPV1 after four weeks of the neuropathic pain onset. Recent evidence report that after nerve injury cortico(layer 5/6)-spinally projecting neurons undergo to post-synaptic potentiation (Chen T et al., Mol Pain 2014). Probably, these cortical spinally projected neurons may be our TRPV1 immunopositive neurons. Since our data suggest a strategic role of TRPV1 in cortical hyperexcitability, it is tempting to say that this channel may enhance neuronal firing response to incoming sensory thalamic input and trigger spinal facilitation of synaptic transmission.

A sentence on this point has been added in the discussion

Minor comments.

1) There are many typos throughout the manuscript, careful proofreading is necessary. e.g. Fig 6. Hypertrophied, not ipertrophied. Fig 4d is referred to as figure 5d in the results.

The manuscript has been revised by an *English mother-tongue* colleague prior to re-submission and the above indicated corrections has been done.

Reviewer #2 (Remarks to the Author):

Summary of key results:

In this manuscript, Drs Marrone and colleagues first use immunohistochemistry to show that TRPV1 is expressed in microglia and electrophysiological recording to show that it is functional. In a second part of the manuscript, they show that upon CCI, TRPV1 is also expressed by neurons. Then the authors show that application of Capsaicin on acute slices of ACC increases the frequency of mEPSC. In another section of the manuscript, the authors show that capsaicin stimulation of cultured microglia promotes the production of micro vesicles. They also show that blocking the production of neuronal sphingosine prevents the effect of capsaicin on minis. The authors then show that microglia undergo similar morphological changes upon chronic TRPV1 loss-of-function or acute Capsaicin stimulation. Finally, the authors examine the neuronal excitability of neurons and show that upon CCI, the electrophysiological responses of neurons to capsaicin is greatly modified as compared to control.

Originality and interest :

It is of great interest to understand how microglia and inflammation modulate the brain physiology. Here, the authors demonstrate that TRPV1 is specifically expressed by microglia in the healthy brain and therefore that the effects of capsaicin are mediated by

the cells (see below, the remark on the astrocytic expression). They further characterize some neuronal consequences of TRPV1 stimulation by capsaicin. Such experiments could be the first steps to demonstrate of the involvement of microglial TRPV1 in the physio- pathology of the brain. However, at this stage, the manuscript is more a juxtaposition of related stories than an homogenous study. Of note, each of these stories, if properly conducted and demonstrated would certainly be of the highest interest.

We thank the Reviewer for his/her appreciation of our work and for his suggestion in splitting the study in two different stories. We think, however, that the novelty of TRPV1 functional expression in microglia cells under physiological conditions will be better accepted by the neuroscience community and will be more solid if presented together with the evidence on changes of TRPV1 expression pattern following an injury. In addition, showing together the two different mechanisms by which TRPV1 interact with neurons (indirectly when expressed in microglia and directly when distributed on neurons) give a more complete vision on the TRPV1 function in the healthy and pathological brain.

However we are open to any changes if the Editor along with Reviewers consider worthwhile to split the story in two separate papers.

Data & methodology:

The methodology is appropriate and the data presented are of sufficient quality. However, the presentation is sometime confusing, mostly because the different stories are intermingled, but also because of the number of supplementary figures is too big and most of them show the same kind of information. Supp fig 5-7 might be fused. This is minor, but the supplementary figures might be easier to read if they share the organization of the principal figures (TRPV immunoreactivity consistently shown in the green channel and displayed in the upper rows)

We thank the Reviewer for his/her suggestions and we agree with him/her opinion in reducing the number of supplementary figures. Unfortunately, we could not fuse Fig.6-7 by adding the spinal cord panel below the brain areas images. The reason for this is because spinal cord neurons have different acquisition focal planes respect to glial cells and therefore in the same panel it cannot be illustrated TRPV1 staining for all spinal cord cells. Regarding the fusing of Suppl. Fig 5 with Suppl. Fig 6, we think that is not correct to illustrate in a single figure two different messages: localization of TRPV1 in astrocytes (Suppl. Fig 5) and similar TRPV1 microglia staining between different brain areas.

Consistently with the other figures, we changed the color channel and the display order of TRPV1 immunoreactivity.

Appropriate use of statistics:

It is not clear from figure 4d if LPA has an effect on its own or if the effect is only significant when compared with the BrPLPA condition. In that case, it seems difficult to conclude that "LPA is a potential endogenous activator of TRPV1 in the brain" (page 9 last sentence).

The Reviewer rightly points on the difficulties in believing LPA as an endogenous activator if the experiments have been performed with the BrP-LPA (LPA1-4 receptor antagonist). Therefore, we carried out a new set of experiments without blocking LPA1-LPA4 receptor. **The new results are now included in the Fig 6 and in Results.** What

we observed is that LPA alone induced a huge increase of mEPSC frequency but not of amplitude. We then tested if this enhancement of glutamatergic currents was TRPV1 mediated. Although LPA still augmented the rate of mEPSCs when TRPV1s were pharmacologically blocked, this enhancement was significantly smaller compared to one induced by LPA alone ($p < 0.05$, Fig. 6e). Blocking both LPA receptors and TRPV1, no increase of glutamatergic transmission was detected (Fig. 6g).

Conclusions:

One of the conclusions of this manuscript is that "Capsaicin increases (...) synaptic activity by promoting shedding of microglial MVs which in turn fosters sphingosine metabolism in neurons and enhances presynaptic probability releases". However, the data do not support such hypothesis. The authors convincingly show that capsaicin increases the frequency of EPSC in acute ACC slices and that this increase is prevented by inhibiting the acid ceramidase. In another series of experiments, the authors show that TRPV1 stimulation of cultured microglia promotes the production of MV, which are known from a paper previously published by some of the authors, to increase the EPSC frequency of cultured neurons by a sphingosine dependent mechanism. These correlations however do not support the conclusion raised by the authors.

We thank the reviewer for this comment that prompted us to perform a set of new experiments to strengthen the involvement of microglial MVs in the increase in EPSC frequency caused by capsaicin.

We now provide evidence that block of MV shedding from microglia prevents the increase in EPSC frequency induced by capsaicin.

To pharmacologically block MV shedding from microglia we used the p38 MAPK inhibitor SB203580. P38 MAPK was described by our group to be activated downstream P2X7 ATP receptor and to be essential for MV shedding evoked by ATP, as its activation causes translocation into the PM of a key enzyme (acid sphingomyelinase) mediating MV budding (Bianco F et al., EMBO J 2009). Given that p38 MAPK is activated downstream TRPV1 receptors (Amantini C et al., J Neurochem 2007), we reasoned that inhibition of p38 MAPK could block capsaicin-induced MV shedding as well. To test this hypothesis we measured by nanotacking particle analysis MV production and found a significant decrease in capsaicin-induced MV shedding from cultured microglia treated with 400 nM SB-203580 for 15 min incubation (fig X). Having assessed the capability of SB-203580 to decrease capsaicin-induced MV production we incubated cortical slices with SB203580 2 mM for 30 min before being transferred in the recording chamber. We found that capsaicin failed to modulate mEPSC frequency and other kinetic parameters in slices in which MV shedding was impaired. These new data are reported in Fig 5g and suppl. Fig 9d. In addition we relocated Fig 5f as suppl. Fig 9a.

Concerning the inhibitory action of the ceramidase inhibitor ARN, we would like to point out that in the original version of the manuscript we have dissected out the site of action of the ceramidase inhibitor by performing experiments on cultured microglia and neurons. In case the reviewer has missed these experiments they are reported in suppl. fig 9. We showed that in the presence of ARN, capsaicin efficiently promotes MV shedding from microglia and that MVs secreted from ARN-treated microglia upon capsaicin stimulation

are able to increase mEPSC frequency in cultured neurons (suppl.fig 9). These results suggest that ARN strongly dampens TRPV1-mediated enhancement of glutamatergic neurotransmission, because it impairs EV action on target neurons: by inhibiting sphingomyelin metabolism to sphingosine in response to EVs the ceramidase inhibitor reduces the synaptic vesicle fusion with the neuronal plasma membrane.

Suggested improvements:

1) Each of these stories, if properly demonstrated, could be a very interesting piece of data on its own. The authors could focus the manuscript either on the consequences of TRPV1 expression in neurons upon CCI or on the involvement of microglial TRPV1 in the control of mEPSC frequency.

As stated above in the “Originality and interest”, we believed in a more complete vision on the cortical function of TRPV1. In addition, in the editorial decision it was reported “Referee #2 recommended refocusing the manuscript; editorially, we would expect you to keep all current data while expanding the manuscript to address the concerns of all referees”.

2) The authors convincingly characterize the morphological consequences of TRPV1 stimulation by capsaicin (figure 6). Therefore, they should stick to morphological description of microglia (ramified, bushy etc) and avoid describing microglia as being "activated", "resting" or in a "surveillance status" (which correspond to functional description).

Thank you for your correct advice. However, in the new version of the manuscript we have added new data on the microglia phenotype based on the cytokine produced and released upon TRPV1 stimulation in both WT and TRPV1^{-/-} mice.

3) The authors could be more precise on the TRPV1 expression in astrocytes. Visual analysis of the merged panel of supplementary figure 5, suggests most of the red-labelled cells (astrocytes) of the hippocampus parenchyma also display green labelling (TRPV immunoreactivity). The same conclusion can be drawn from sup figure 7a and b. Yet, the authors write that "TRPV1 is expressed at lower levels in astrocytes than in microglial cells", which is unclear. Do all astrocytes express low level of TRPV1 or do few astrocytes express strong level of TRPV1 ? The interpretation of the effect of capsaicin application would be very different depending on the expression by astrocytes.

As reported in the Methods, Results and Figures, the amount of colocalizations was evaluated by using the Pearson statistical index of correlation (r) between two variables. Based on this coefficient values we found that in anterior cingulate cortex the TRPV1 expression in astrocytes was significantly than in microglia, and ($r = 0.291 \pm 0.011$ $r = 0.720 \pm 0.010$, for astrocytes and microglia, respectively, $p < 0.01$; Two Sample t-Test). However since GFAP positive cells in the ACC were very sparse we have chosen the hippocampus as comparison area to avoid an underestimation of TRPV1 in astrocytes (Supplementary Figure 5b). Despite the wider population GFAP positive cells, a low degree of TRPV1-GFAP colocalization was also detected in hippocampus ($r = 0.344 \pm 0.016$; Supplementary Figure 5c,d). Hence, TRPV1 is expressed at much lower levels in astrocytes than in microglial cells. A visual inspection of all TRPV1/GFAP double positive cells reveals a heterogeneous expression of TRPV1 in astrocytes. Therefore the

Reviewer rightly suggest that capsaicin effects on astrocytes maybe dependent on the expression level of TRPV1 in these cells. However, in our proposed model astrocytes are not involved in microglia to neuron communication upon TRPV1 stimulation. **This result is now included in the revised manuscript as Fig. 5d-f.**

Certainly, these experiments have been carried-out in resting conditions. We cannot exclude that during gliosis astrocytes may mediate microglia-to-neuron communication upon TRPV1 stimulation or be themselves key players of neurotransmission regulation (Nam JHet al., Brain 2015; Miyanohara J et al., Biochem Biophys Res Commun 2015; Ho KV et al., Glia 2014)

4) According to the experimental procedures (page 25 and 30), Capsaicin treatment is applied for 10 min. However, according to supp figure 4d, the effect of capsaicin application on microglia is very transient (2 min) and the electrical properties of microglia returns to basal level after 10 min. The authors should at least propose an explanation to this apparent paradox

We are grateful with the Reviewer for raising this point that allow us to correct a mistake. **Experimental procedures of microglia cell recordings have been corrected (10 minutes has been replaced with 3-10 minutes capsaicin application). The bar application of capsaicin in Suppl. Fig 4e has been substituted with an arrow, and in the legend it has been specified the time of application.**

In patch-clamp recordings of microglia cells the change of microglial electrical properties was clearly transient only in 3/9 cells. The average time plot reported in Suppl Fig 4e shows long lasting current response over the time of capsaicin application (4 minutes). Actually, it is not unexpected that the activation of desensitizing TRPV1 (Caterina et al., 1997) can lead to calcium-dependent long lasting responses (see Gibson et al., 2008 *Neuron*).

Since the aim of these experiments was to demonstrate the existence of functional TRPV1 in microglia cells, in our future experiments will be deeply investigate the mechanistic aspects on the microglial /neuronal current relationship.

5) The labeling pattern of the spinal cord shown in supp fig 7 is puzzling. According to panel h, a large proportion of neurons are labelled. Yet, almost no labelling is detected in panel b and e even though dapi staining suggests that neurons are most present in the section.

This is due to the different acquisition focal planes for neurons/astrocytes/microglia that is typical for spinal cord tissue.

The size of the neurons shown in supp 7h suggests that these are motoneurons. Yet, the authors indicate that images have been taken in the dorsal horn, in which there is no motoneurons. Authors might clarify this issue.

As for point 4 we really thank the Reviewer for allowing us to note the terrible mistake. Both the GFAP and NeuN panels have been replaced with new ones. Actually, the previous version represented acquisitions from the ventral horn, while now are included images of the dorsal horn. Sorry again and thank you for pointing this out!

Minor :

-The authors should use the same scale for the y axis when describing the mEPSC frequencies and amplitudes. This could help the reader to compare the panels within a figure

Accordingly with the Reviewer we have changed the y axis except for those graphs in which this change would have caused a leveling of the grouped data and therefore a lost of information (i.e. the ongoing of a single experiment represented as “dot”).

-In figure 1, the authors should add yellow arrows in the "NeuN" panel (and not only in the TrpV1 channel).

Done! Thank you

-Legend of figure 5d : the capsaicin (not capsicin) line is in red not dotted

Ok we corrected it. Thanks!

Typos:

-Ipertrophied is not English.

☺

-p9 para3 "(, Fig.4b)" corrected!

-p16 "...mice do not undergo to further activation". Corrected!

-Legend of supp figure 4: the concentration of capsaicin cannot be read. Thank you!

Reviewer #3 (Remarks to the Author):

The role of TRPV1 in the brain is intriguing. In the study by Marrone and colleagues, it is concluded that TRPV1 is mainly expressed in microglial cells under normal conditions, and that stimulation of TRPV1 indirectly enhances glutamatergic transmission. However, in the case of chronic pain, neuronal TRPV1 is proposed to play a more important role in the regulation of neuronal signalling. The immunohistochemical characterization is convincing but not the link between microglia to neuron communication.

We thank the reviewer for his/her comments, which allowed improving our study and to better present the link between microglia-neuron communication. In addition to experiments reported below (minocycline/CCI cortical slices), further experiments were carried out to strengthen the MVs mediation of microglia to neuron communication upon TRPV1 activation (Suppl. Fig 9D, Fig5m). For more detail please see our response to Reviewer 2 – Conclusion point.

Several studies, not quoted, have raised the possibility that TRPV1 in PAG is of importance in pain processing. Why was this region not included in the study?

We designed this study focusing onto the spino-thalamus-cortical tract, and for this

reason we studied the ACC. However, **in this revised version we have added immunohistochemical characterization of TRPV1 in PAG from naïve and CCI mice in the Results and as Suppl. Fig6d.**

Is the capsaicin-induced increase in mEPSC blocked by glutamate receptor antagonists?

To address this point we performed a set of experiments in the presence of picrotoxin (100 μ M) and tetrodotoxin (300nM) to exclude the GABAergic component and the glutamate release from pre-synaptic terminals imposed by the action potentials. Bath application of the non NMDA receptors antagonist, 6,7-dinitroquinoxaline-2,3-dione (DNQX, 20 μ M) completely abolished both frequency and amplitude of mEPSCs (from 6.45 \pm 1.09 Hz to 0.38 \pm 0.11 Hz, n= 6,paired T-test p= <0.001; from 16.79 \pm 0.80 pA to 0.88 \pm 0.18 pA, n=6, paired t test p= <0.001). In these conditions, capsaicin increased neither frequency nor amplitude of mEPSCs, suggesting that downstream TRPV1 signalling is AMPA receptor mediated. The NMDA component was excluded in this type of experiments (resting membrane potential -70mV, [Mg²⁺] 2mM in the internal solution).

Glia-derived glutamate facilitates neurotransmitters release through presynaptic metabotropic receptors {Fiacco and McCarthy, 2004;Perea and Araque, 2007; Pascual et al. 2012); to investigate a possible involvement of these receptors in the presynaptic modulation of glutamatergic transmission upon TRPV1 activation, capsaicin was applied with the mGluR5 antagonist MPEP (50 μ M) or with the non-selective group I mGluR antagonist MCPG (100 μ M) (FigSS3 B-C). In both assays capsaicin significantly increased mEPSC frequency (from 7.96 \pm 1.11 to 9.27 \pm 0.89,n=7, p= 0.040 paired sample T-test; from 6.67 \pm 0.71 to 8.039 \pm 1.14385, n=8, p=0.042 paired sample T-test), suggesting that mGluR are not involved in the upstream TRPV1 signalling. Both MPEP and MCPG did not induce changes in basal synaptic properties. (Ctrl 7.62203 \pm 0.83 Hz, MPEP 7.95877 \pm 1.10 Hz; Ctrl 19.16708 \pm 0.98 pA MPEP 18.83072 \pm 0.69 pA, n=7,p=0.61 and p=0.62 respectively, Paired sample T-test; Ctrl 6.19 \pm 0.67606 MCPG 6.66 \pm 0.71Hz; Ctrl 19.46 \pm 0.97 MCPG 18.66 \pm 1.18,n=8, p=0.10 and p=0.13, Paired sample T-test.)

Figure SS3

What is the effect of minocycline on heterologously expressed TRPV1?

A new set of voltage clamp recordings from PNs of CCI mice in the presence of minocycline has been performed. **The obtained results have been reported in the text of the last paragraph of the Result section and as Suppl Fig 11e.**

Bearing in mind that not all recorded neurons from CCI mice expressed functional TRPV1, we found a similar proportion of TRPV1 responding neurons between experiments performed with and without minocycline (4/14 and 10 /18 neurons, respectively; $p= 0.61$ Chi-Squared of differences on two levels between 2 groups: 2x2), In 4 out of 14 neurons from slices treated with minocycline, capsaicin increased the frequency, amplitude and charge transfer of mEPSCs ($p<0.05$ Kolmogorov Smirnov test). The overall electrophysiological effects mediated by TRPV1 activation and recorded from minocycline treated slices are similar to those of control experiments. Hence these results suggest that changes induced by minocycline on the properties of TRPV1 expressed by microglial should not account for the alteration of neuronal synaptic strength. **A sentence on this point has been added in the Discussion**

What is the effect of LPS on TRPV1?

We thank the reviewer for raising this point.

We reason that LPS in particular its receptor TLR4 may share some steps with TRPV1 signalling, based on our own and on literature evidence:

a) TRPV1 activation induces NF-kB-dependent production and release of TNF α from WT microglia (Fig 6; Sappington RM and Calkins, DJ Invest Ophthalmol Vis Sci 2008);

b) Similarly to TRPV1 activation, LPS binding to TLR4 produces the same signalling cascade (Chandel NS et al., J Immunol. 2000);

c) we show in this paper that both LPS and capsaicin enhance glutamatergic neurotransmission and their effects are occluded (Fig 4 and Fig 7);

d) we observed a significant reduction of LPS effects in TRPV1^{-/-} cortical slices (Fig 4 and Suppl Fig 11).

Mechanistically, the reduction of LPS effect in the TRPV1^{-/-} tissue could derive from the persistently reduced level of NF-kB, which holds microglia in a “refractory” state in the brain of TRPV null genetic background (please for details see our response to Reviewer 1 point 5).

Therefore, in conclusion we posit that LPS/TLR4 and capsaicin/TRPV1 use much the same signaling pathways, probably upstream to or at the level of NF-kB of TRPV1/TLR4 signalling.

p. 17, 1st sentence: ...ones pain became chronic." It would be good to show corresponding behavioural data as graphs.

We have added the behavioral graph in the new manuscript version as Suppl. Fig 13

Most of the electrophysiology data from slices display tiny differences with overlapping SEM values. Even though significant, it is difficult to appreciate what the physiological impact would be caused by such small changes in mEPSC.

For example, the effect of LPA is not convincing in Figure 4c (is BrPLPA different from ctrl? +LPA is unlikely different from ctrl?, what statistical method was used?).

We agree with the referee that the electrophysiological effects that we report are not quantitatively striking, although they express real and consistent modifications occurring in the entire neuronal population, according to the inferential statistical analysis that we adopted. Indeed, it is hard to predict the extent of the impact that such a small increase in glutamate release probability is going to have in the entire inflammatory response. However, even assuming it might be marginal, such electrophysiological modification is only one experimental readout of TRPV1 activation. For instance, the inflammatory activity of capsaicin is not only expressed as an enhancement of excitatory neurotransmission, but also as a stimulation of microglial potassium channels and elevation of microvesicle release. Likewise, using glutamate neurotransmission as a highly sensitive electrophysiological readout, we provide evidence that TRPV1 is activated by LPA. However, the proinflammatory effects of LPA through TRPV1 activation may well not be limited to such increase of glutamatergic neurotransmission, but may for instance activate potassium conductances in microglial cells or increase the microglial microvesicles shedding, as previously demonstrated (Schilling T et al., Eur.J. of Neurosci 2004; Duc Bach Nguyen et al., Cell Physiol Biochem 2016).

We have improved the weakness of LPA effects by performing new experiments that are now part of the new manuscript version. Please, read our answer to the Reviewer 2 in the

section “*Appropriate use of statistics*” and to Reviewer 1, in point 6.

Also, with regard to the mentioned study on FAAH and microglia function (p. 16), other endovanilloids would have been more relevant to study.

We agree with this point, the endocannabinoid anandamide and/or other FAAH substrates such as palmitoylethanolamide and oleoylethanolamide could be potential endovanilloids involved in this TRPV1 signalling. The reasons for having investigated on the bioactive lipid LPA instead of anandamide and congeners are the following:

-in our previous studies we showed that the endocannabinoid anandamide activated TRPV1 only at high concentration (30 μ M) and during TRPV1 sensitization (Marinelli S., et al J Neurosci 2003).

-other evidence show that anandamide activated TRPV1 only when FAAH was inhibited (Starowicz K et al., 2013 Plos One).

-Palmitoylethanolamide and/or oleoylethanolamide, similarly to anandamide, besides their high affinity receptors PPAR γ , stimulated both TRPV1 and CB1, in a FAAH activity-dependent fashion.

-While in our study TRPV1 stimulation increases the production and release of TNF α , other evidence found reduced levels of this cytokine induced by palmitoylethanolamide (Costa B et al., Pain 2008).

-LPA similarly to capsaicin activates potassium conductances in microglial cells and increase the microglial microvesicles shedding (Schilling T et al., Eur.J. of Neurosci 2004; Duc Bach Nguyen et al., Cell Physiol Biochem 2016).

Although our results promotes LPA as a possible endovanilloid able to tune excitatory neurotransmission, we do not exclude other suitable partners in the TRPV1 signalling

A sentence on this point was added to the Discussion

Taken together, statistics are not clearly described. Each statistical test should be clearly stated in the text including figure legends. Statistical significance is normally *P<0.05, **P<0.01 and ***P<0.001, but not in legend to Figure 4 and 7 where **p<0.001. Also *p{less than or equal to}0.05 in figure legend 8 is not informative.

In this revised version all statistics employed are now specified in the figure legends.

Ref 60 is incomplete.

Thank you, we have corrected it

Supplementary Figure 12. This figure should be moved to the main text as it summarizes very well the authors' thoughts.

We agree with the Reviewer that the cartoon mechanism would more visible if moved to the main text but the limited number of Main Figures (maximum 8), prompt us to leave it as supplementary. We leave this point to Editor decision.

Summary, last sentence: TRPV1 is already identified as a detector of harmful stimuli.

Yes but at peripheral level not in the brain. Indeed the present data clarify the TRPV1 role in the brain with respect to that held by the same receptors in the peripheral somatosensory system. TRPV1s expressed in the main immune cells of the brain may similarly function as detectors of nociceptive and inflammatory stimuli/agents

Discussion, last sentence: How could TRPV1 be a potential biomarker? By removing brain tissues from humans?

We substituted this sentence with the following: “our study suggests that future strategies aimed at imaging TRPV1 cellular expression and distribution may provide differential diagnostic tools for pathologies associated with an inflammatory brain component”. Our idea is that the mechanism disclosed by our study, and the shift in expression from microglia to neurons, may be useful to identify neuropathologies with inflammatory components and may differentiate between different pathologies. Admittedly, methods to achieve this must still be developed. We trust the Reviewer will agree with the revised wording.

Reviewers' comments:

Reviewer #1 (Remarks to the Author):

I thank the authors for their careful consideration of my comments. The novel experiments strengthen their claims that TRPV1 is expressed by, and regulates the function of cortical microglia and astrocyte, and increases the impact of the study.

Given the very large number of investigations on TRPV1, it remains puzzling that no one had previously reported TRPV1 expression and function in cortical glia. The authors have performed a number of adequate control experiments, including using TRPV1 null mice, that are difficult to fault. There is considerable controversy regarding TRPV1 distribution in the CNS, and this article would undoubtedly fuel this controversy. The possible explanations provided by the authors in their response regarding the discrepancies between their findings and those derived from the use of TRPV1 reporter mice, are interesting but clearly do not fully explain these discrepancies, and will not convince all researchers in the field. It is thus essential that the discussion adequately represent the literature on the subject. For example, it is not appropriate to indicate "Hence, those results from TRPV1 reporter mice are entirely consistent with ours". Clearly, this is not the case. I rather suggest to include in the manuscript, in some form, the complete discussion sections on important and controversial points provided by the authors to the reviewers in their response, rather than isolated sentences.

Some critics have not been addressed adequately. In particular, the immunohistochemistry images provided in DRG and spinal cord are not convincing. Whether immunoreactivity generated by the anti-TRPV1 Mab is observed in peptidergic unmyelinated nociceptors, spinal cord interneurons or projection neurons, remains unclear. This issue could be addressed easily by providing both low magnification images of whole DRG and dorsal horn and high magnifications images of co-labeling experiments with adequate markers (e.g. CGRP for DRG).

Reviewer #2 (Remarks to the Author):

In the first version of the manuscript, this reviewer ask for 5 main modifications and raised a concern about a conclusion.

1-for the sake of clarity, the authors could have focussed on one story (TrpV and EPSC or TrpV and pain) and to reduce the number of supplemental figures . The authors chose not to follow this recommendation.

2-Functional categories cannot be based on morphological analysis. The authors have changed some of their descriptions but chose to keep a lot of functional descriptions based on morphology. eg: page 13 "Morphometric analyses showed that the majority of microglial cells from acute slices incubated in ACSF were in a ramified/surveillance resting status"; page 13 "...capsaicin-induced morphological changes were absent in sections from TRPV1-/- mice (Fig. 6f,h). In the latter, microglia appeared already in activated state under baseline conditions.";see also figure 6 and supp figure 11

3-Expression in astrocytes. The authors have added convincing functional results using FAc showing that astrocytes are not involved in the change in frequency induced by Capsaicine. However, the authors base their conclusion of a lower expression of Trpv1 in astrocytes on the Pearson correlation. In microglia, Trpv1 a membrane protein is compared to Iba1 another membrane protein. In astrocytes, Trpv1 is compared to GFAP which is a cytoskeleton protein. Comparing the pattern of a membrane protein to a cytoskeleton protein cannot give accurate results.

points 4 and 5 were errors of the authors that have been corrected.

In the first version, this reviewer noticed that the conclusion "Capsaicin increases synaptic activity by promoting shedding of microglial MV which in turn fosters sphingosine metabolism in neurons and enhances presynaptic release probability" was not supported by data but only by correlations. The authors have now added an experiment in which they show that blocking p38kinase prevents the capsaicin-induced modulation of EPSC. Based on this experiment, stick to their conclusion. However, p38 kinase acts downstream of Trpv1 and also prevents the MV shedding. This is still a correlation that does not support the conclusion.

In conclusion, the authors have only partially answered to the questions raised in the first round of review.

In addition, in supp figure 2, the authors show convincingly that two Trpv1 polyclonal antibodies display artifactual neuronal labelling (the company selling these antibodies should be indicated). However, neuronal TRPV1 immunoreactivity observed upon CCI is strongly reminiscent of this non specific immunoreactivity shown in supp figure 2. This reviewer could not see any control labelling in Trpv KO upon CCI. Can the authors exclude that upon CCI, a non specific immunoreactivity appears on neurons? The "sporadic" expression of Trpv1 described p9 second para favors the hypothesis of a non specific expression.

There are still minor remarks :

It is not clear why the authors indicate the size of the sample for non quantified results. In addition, the nature of the "n" is not specified. For instance p8 n=19, 7 mice; p9 : n=3 form 3 mice. This increases the confusion and make the reading more difficult.

Figure 6 : what does "the percentage of TNF α production" means ?

Material and methods: morphometric analyses refer to supp figure 7. This is supp figure 6 in this version of the manuscript.

typos:

p29 : "following imagine acquisition"

p30 first lane : "ipertrophied"

Reviewer #3 (Remarks to the Author):

I appreciate very much the rebuttal letter and the extensively revised study by Marrone and colleagues. I congratulate the authors to their impressive study that certainly is worth publishing in Nature Communications.

No doubt, the role of TRPV1 in the brain as well as the identity of endovanilloids can raise emotions. However, as advised by nature editors (doi:10.1038/nchembio.310) citing well is key and, without diminishing the findings by the authors it would be generous to add a few references to balance their view:

Introduction (p. 3): Please add Zygmunt et al Plos One 2013 clearly showing that monoacylglycerols

directly activate TRPV1, and indeed 2-arachidonoylglycerol is very potent when TRPV1 is not desensitized.

Results (p. 8): There are additional studies (Mallet et al Plos One 2010; Barriere et al Plos One 2013; Zygmunt et al Plos One 2013; Kerckhove et al Pain 2014; Silva et al Neuropharmacol 2016) showing that TRPV1 activators can affect nociceptive processing also in the brain. Thus, there is not only sparse evidence that TRPV1 in the brain is involved in nociceptive processing, and these references should be included.

Discussion (p. 18, 2nd paragraph): Please add Zygmunt et al Plos One 2013 clearly showing that monoacylglycerols directly activate TRPV1, and add monoacylglycerols as candidate endovanilloids.

On a final note, the authors' choice of endovanilloid (LPA) can be justified without reducing the role of N-acylethanolamides due to potency differences. In fact LPA is not more potent than anandamide as an activator of the purified TRPV1 (Cao et al Neuron 2013). Also, anandamide in contrast to LPA at submicromolar concentrations lowered the TRPV1 temperature threshold allowing TRPV1 to fully respond to physiological relevant temperatures.

Response to Reviewers' comments:

Reviewer #1 (Remarks to the Author):

I thank the authors for their careful consideration of my comments. The novel experiments strengthen their claims that TRPV1 is expressed by, and regulates the function of cortical microglia and astrocyte, and increases the impact of the study.

Given the very large number of investigations on TRPV1, it remains puzzling that no one had previously reported TRPV1 expression and function in cortical glia.

The authors have performed a number of adequate control experiments, including using TRPV1 null mice, that are difficult to fault. There is considerable controversy regarding TRPV1 distribution in the CNS, and this article would undoubtedly fuel this controversy. The possible explanations provided by the authors in their response regarding the discrepancies between their findings and those derived from the use of TRPV1 reporter mice, are interesting but clearly do not fully explain these discrepancies, and will not convince all researchers in the field. It is thus essential that the discussion adequately represent the literature on the subject. For example, it is not appropriate to indicate "Hence, those results from TRPV1 reporter mice are entirely consistent with ours". Clearly, this is not the case. I rather suggest to include in the manuscript, in some form, the complete discussion sections on important and controversial points provided by the authors to the reviewers in their response, rather than isolated sentences.

Actually, as outlined in the introduction and in result section, there are previous studies showing that TRPV1 expression also in microglia cells is functional, in particular in vitro experiments (Kim et al., 2006; Schilling and Eder 2009; Miyake et al., 2015) and ex vivo studies in the retina of rat and mice (Sappington and Calkins, 2008; Sappington et al., 2009). Therefore, our study does not represent the first evidence on the functional expression of TRPV1 in microglial cells, rather it demonstrates that in ex vivo brain preparations, in a circuit context where microglia-to-neuron communication is preserved, this channel, in physiological or acute pathological conditions, is expressed and is functional in microglia rather than in neurons: it controls neuroinflammation and, once activated, it indirectly modulates neurotransmission, likely by promoting microglial microvesicles shedding. On the other hand, we showed that in chronic pathological conditions, TRPV1 is expressed also in neurons and directly regulates their excitability. Our study provides a different insight on the brain role of TRPV1, proposing a role that appears more similar to that fulfilled in the sensory neurons and spinal cord. In particular TRPV1 may represent a strategic player between neuroinflammation and synaptic transmission alterations.

We ran into the TRPV1 on microglia evidence by chance, by using a monoclonal anti-TRPV1 antibody, during the screening of different TRPV1 antibodies for our urgent need to detect a fiber staining that might support the electrophysiological results, i.e. the presynaptic modulation of neurotransmission.

The majority of the ex vivo studies (including our own previous ones) on TRPV1 in

the brain has been centered on the immunohistochemical results obtained using

polyclonal antibodies, which only stain neurons and sparse astrocytes and accordingly, by patch-clamp recordings of neurons. Without anatomical evidence of microglia TRPV1 expression, there were no reasons to characterize capsaicin-mediated currents in brain microglia cells, although a previous study investigated the electrophysiological and proinflammatory properties of this ion channel in the CD4⁺ T cells (Bertin S et al, Nat Immunolgy 2014).

Finally, the possibility that microglia express a splicing form of TRPV1, still to be discovered, that is not detected by the previously used polyclonal antibodies or that would escape the reporter system described in Cavanaugh et al, cannot be excluded. This might provide an alternative explanation of why no one had previously reported microglia TRPV1 regulation of neurotransmission, and the modulation of some forms of synaptic plasticity by postsynaptic TRPV1 in naïve mice.

Following the Reviewer suggestions, in this new revised manuscript we have added in the Discussion some points to fully discuss our evidence in relation to the vast literature on this subject, including arguments we provided to the Reviewer in the previous rebuttal.

Some critics have not been addressed adequately. In particular, the immunohistochemistry images provided in DRG and spinal cord are not convincing. Whether immunoreactivity generated by the anti-TRPV1 Mab is observed in peptidergic unmyelinated nociceptors, spinal cord interneurons or projection neurons, remains unclear. This issue could be addressed easily by providing both low magnification images of whole DRG and dorsal horn and high magnifications images of co-labeling experiments with adequate markers (e.g. CGRP for DRG).

Along with the Reviewers' new requests, we performed experiments, this time using specific cellular subset markers, for both DRG and spinal cord from old mice (P60-P80).

A new suppl. Fig 8 and legend on DRG neurons and cellular markers (CGRP, IB4 and NF200) is provided and it is also showed below. **The old data illustrating**

DRG from TRPV1-/- sections are now represented in the last panel of the

new figure. In line with the results obtained by Cavanaugh et al by using Trpv1

reporter mice, we found that among CGRP+ neurons nearly 32% were

immunoreactive for the anti-TRPV1 MAb and 14.5% of IB4+ cells were TRPV1

immunopositive, while the percentage of NF200+ neurons expressing TRPV1

was slightly higher (14% compared to the 4.5% of nlacZ+NF200).

Figure B. 40x confocal images (left panels) and detail magnification (right panels) of DRG sections. Double staining of anti-TRPV1 MAb (green) with anti-CGRP (a), anti-IB4 (b) anti-NF200 (c, all in red) in DRG neurons of WT animals (n=4) indicates that part of neurons identified into each subgroup expressed TRPV1 (yellow; 32% of CGRP⁺ neurons are TRPV1⁺, 14.5 % of IB4⁺ neurons are TRPV1⁺, 14 % NF100⁺ cells were TRPV1⁺). (d) In TRPV1^{-/-} DRG no cells show labelling for TRPV1 MAb (n=3)

The results from the new experiments on spinal cord carried out using glutamatergic (CAMKII) and GABAergic markers (GAD65/67) are reported below. Both inhibitory and excitatory neurons from dorsal and ventral horn were immunopositive for the anti-TRPV1 MAb. Right now, we did not integrate these new data within the Suppl Fig 7 (TRPV1 expression in astrocytes, microglia and WT/TRPV1ko neurons) but we provided them only in the rebuttal. However if the Reviewer considers worthwhile adding these new data as supplementary new figure, we will certainly add it.

Figure B. TRPV1 is expressed in both glutamatergic (a,b) and GABAergic neurons (c,d) of the dorsal and ventral horn. (a,b) Photomicrographs of immunofluorescence for the anti-TRPV1 MAb (green) and anti-CAMKII (red) in dorsal (DH) and ventral horn (VH). CAMKII is highly and diffusely expressed in DH and less in VH of naïve animal spinal cord. The anti-TRPV1 MAb is highly expressed and partially co-localizes with CAMKII present in body cells of both horns (merged panels - yellow). **(b,c)** Photomicrographs of double immunofluorescence for anti-GAD65/67 (red) and anti-TRPV1 MAb (green). GAD65/67 is expressed as dots in DH and VH of naïve animal

spinal cord. The anti-TRPV1 MAb is highly expressed and partially co-localizes with GAD65/67 present in body cells (merged panels - yellow).

Reviewer #2:

In the first version of the manuscript, this reviewer asked for 5 main modifications and raised a concern about a conclusion.

1) For the sake of clarity, the authors could have focussed on one story (TrpV and EPSC or TrpV and pain) and to reduce the number of supplemental figures. The authors chose not to follow this recommendation.

We regret that the Reviewer thinks it was our choice keeping the study with the physiological and pain stories together. As outlined in the previous rebuttal, it was not our decision, although we do like the two stories together, but, rather, we followed an editorial clear recommendation: "You will see that Referee #2 recommended refocusing the manuscript; editorially, we would expect you to keep all current data while expanding the manuscript to address the concerns of all referees".

Moreover, in the previous cover letter we reiterated the Reviewer advice.

2) Functional categories cannot be based on morphological analysis. The authors have changed some of their descriptions but chose to keep a lot of functional descriptions based on morphology. eg: page 13 "Morphometric analyses showed that the majority of microglial cells from acute slices incubated in ACSF were in a ramified/surveillance resting status"; page 13 "...capsaicin-induced morphological changes were absent in sections from TRPV1-/- mice (Fig. 6f,h). In the latter, microglia appeared already in activated state under baseline conditions."; see also figure 6 and supp figure 11

In the previous revised version we expressly left functional descriptions of microglia morphology since they were correlated to the cytokine release measurements. Nevertheless, as requested by the Reviewer, in this new revised version, we have removed all microglial functional descriptions.

3) Expression in astrocytes. The authors have added convincing functional results using FAc showing that astrocytes are not involved in the change in frequency induced by Capsaicine. However, the authors base their conclusion of a lower expression of Trpv1 in astrocytes on the Pearson correlation. In microglia, Trpv1 a membrane protein is compared to Iba1 another membrane protein. In astrocytes, Trpv1 is compared to GFAP which is a cytoskeleton protein. Comparing the pattern of a membrane protein to a cytoskeleton protein cannot give accurate results.

We are pleased to note that the Reviewer agrees with the marginal role played by astrocytic TRPV1, in microglia to neuron communication as revealed by fluoroacetate experiments. Nonetheless, the Reviewer has a point in arguing that an analysis based on Pearson coefficient cannot give accurate information if the two proteins of interest stain different cellular compartments. And we agree with this.

After having screened different antibodies staining the astrocytic membrane compartment, we have carried out new sets of experiments by using two astroglial markers, the GLAST and GLT- 1. The quality and pattern of their staining is very different from what we obtained with the anti- GFAP ab. Nevertheless, the colocalization analyses between the anti-TRPV1 and anti-GLAST or anti-GLT1 provide CCP values even lower than those obtained with the GFAP, confirming the negligible role of astrocytic TRPV1 assessed by FAc experiments. **These new results are now included in the Result section and as suppl Fig. 5 (only the merged panels with their relative enlargements), replacing the old ones obtained with the anti-GFAP.** For the sake of clarity, we also provide split panels relative to the experiments illustrated in suppl. Fig. 5, here below:

Figure C. TRPV1 distribution in astrocytes of the ACC and

hippocampus of adult naïve mice. Representative examples of low magnification (40X) IF images of anti-TRPV1 (green), Glast and GLT-1 (red), and their colocalization (merge, yellow), taken from sections of anterior cingulate cortex (ACC) and hippocampus (HIPPO) of naïve mice.

points 4 and 5 were errors of the authors that have been corrected.

6) In the first version, this reviewer noticed that the conclusion "Capsaicin increases synaptic activity by promoting shedding of microglial MV which in turn fosters sphingosine metabolism in neurons and enhances presynaptic release probability" was not supported by data but only by correlations. The authors have now added an experiment in which they show that blocking p38kinase prevents the capsaicin-induced modulation of EPSC. Based on this experiment, stick to their conclusion.

However, p38 kinase acts downstream of Trpv1 and also prevents the MV shedding. This is still a correlation that does not support the conclusion.

We agree with the Reviewer that the results obtained by these experiments show a correlation between the TRPV1 signalling on microglia and synaptic function. However our results strongly suggest a possible route at the base of microglia-neuron communication. Therefore, **we have edited the results (p12), conclusions (p19) and the legend of supplementary Fig 14** stating that our evidence suggests (although does not formally prove) that TRPV1-induced microglia microvesicles shedding may play a role in the modulation of glutamatergic neurotransmission.

To provide a more causal link on this phenomenon a dedicated study will be performed.

In conclusion, the authors have only partially answered to the questions raised in the first round of review.

In addition, in supp figure 2, the authors show convincingly that two Trpv1 polyclonal antibodies display artifactual neuronal labelling (the company selling these antibodies should be indicated). However, neuronal TRPV1 immunoreactivity observed upon CCI is strongly reminiscent of this non specific immunoreactivity shown in suppl figure 2. This reviewer could not see any control labelling in Trpv KO upon CCI. Can the authors exclude that upon CCI, a non specific immunoreactivity appears on neurons? The "sporadic" expression of Trpv1 described p9 second para favors the hypothesis of a non specific expression.

This claim is new and we respectfully believe it should have been raised during the previous revision. However, we provide new evidence for the lack of cellular and even more neuronal staining of TRPV1 MAb in the cortex of TRPV1^{-/-} mice suffering from neuropathic pain (NP) at both 1 and 4 weeks from the ligation of sciatic nerve (see panels below). Thus, these results exclude the assumption made by the Reviewer on the aspecific immunoreactivity related (relative) to TRPV1 neuronal labeling upon CCI. In addition, we would like to stress the time dependency of TRPV1 expression in ACC neurons from the onset of NP, that is consistent in all

analyzed samples (Fig.2), and its area specificity. Indeed, together with the new data on TRPV1^{-/-}, the “sparse” evidence of TRPV1 neuronal localization in the other painmatrix areas does not depend on the MAb aspecificity, rather to a “biomarker “behavior of this protein in the brain. For example, under other pathological conditions such as neurodegeneration, reproduced by using transgenic mouse that lack sufficient nerve growth factor (NGF) and suffer from an inflammatory/AD-like syndrome, we detected the TRPV1 neuronal expression mainly in the thalamus (Silvia Marinelli’s laboratory, preliminary results).

In the legend of the suppl Fig 2 we have indicated the companies selling the pAbs tested in this study.

Confocal laser scanning photomicrographs of ACC sections containing superficial layers from naïve (a), 1 week (b) and 1 month CCI TRPV1^{-/-} mice (c,d) stained with the anti-TRPV1 MAb (Millipore Chemicon, 1:100) (a-c) or background control of secondary Ab (d). Anti-TRPV1 MAb shows no staining in naïve condition (a, n=4) and neither at 1 week (b, n=3) nor at 4 weeks (c, n=6) after CCI. Background control of secondary Ab shows no false staining (d).

There are still minor remarks:

It is not clear why the authors indicate the size of the sample for non quantified results. In addition, the nature of the "n" is not specified. For instance p8 n=19, 7 mice; p9 : n=3 from 3 mice. This increases the confusion and make the reading more difficult.

In the last paragraph of page 9 we have specified the number of sample for each examined brain areas to give the reader more clear info on the frequency of the phenomenon i.e. TRPV1 neuronal expression. In any case, we have removed this kind of information from the text. Moreover, along with the Reviewer, **in the paragraph Histology and immunofluorescence of the Method section we have added a sentence in which it is specified the nature of both “n” and sample size. The nature of “n” is also recalled in the first IF experiment of the Result section for easy of reading.** In the previous version of the manuscript “n” represented the number of wells, each containing two or three cortical sections, used for each set of experiments. In this revised version, as specified in the Methods, we decided to simply put the number of mice.

Figure 6: what does "the percentage of TNF α production" means ?

Percentage of TNF- α production represents the percentage of cells that are actively producing the cytokine and is the most accepted way of representing cytokine production in flow cytometry. This is achieved by gating the cell population of interest according to its specific cell marker (in this case CD11b) and then showing the percentage of cells that are positive for the cytokine analyzed. **For the sake of clarity we have added in the figure legend the following note: Percentage of TNF α production means the “Percentage of TNF α producing cells”.**

Material and methods: morphometric analyses refer to supp figure 7. This is supp figure 6 in this version of the manuscript.

Thank you! We have corrected it

typos:

p29 : "following imagine acquisition"

p30 first lane : "ipertrophied"

Corrected

Reviewer #3:

I appreciate very much the rebuttal letter and the extensively revised study by Marrone and colleagues. I congratulate the authors to their impressive study that certainly is worth publishing in Nature Communications.

Thank you!!!!

No doubt, the role of TRPV1 in the brain as well as the identity of endovanilloids can raise emotions. However, as advised by nature editors (doi:10.1038/nchembio.310) citing well is key and, without diminishing the findings by the authors it would be generous to add a few references to balance their view:

Introduction (p. 3): Please add Zygmunt et al Plos One 2013 clearly showing that monoacylglycerols directly activate TRPV1, and indeed 2-arachidonoylglycerol is very potent when TRPV1 is not desensitized.

Done!

Results (p. 8): There are additional studies (Mallet et al Plos One 2010; Barriere et al

Plos One 2013; Zygmunt et al Plos One 2013; Kerckhove et al Pain 2014; Silva et al Neuropharmacol 2016) showing that TRPV1 activators can affect nociceptive processing also in the brain. Thus, there is not only sparse evidence that TRPV1 in the brain is involved in nociceptive processing, and these references should be included.

We fully agree with the Reviewer and therefore we have added some of these references and amended the text.

Discussion (p. 18, 2nd paragraph): Please add Zygmunt et al Plos One 2013 clearly showing that monoacylglycerols directly activate TRPV1, and add monoacylglycerols as candidate endovanilloids.

Done!

On a final note, the authors' choice of endovanilloid (LPA) can be justified without reducing the role of N -acylethanolamides due to potency differences. In fact LPA is not more potent than anandamide as an activator of the purified TRPV1 (Cao et al Neuron 2013). Also, anandamide in contrast to LPA at submicromolar concentrations lowered the TRPV1 temperature threshold allowing TRPV1 to fully respond to physiological relevant temperatures.

Maybe there has been a misunderstanding and we have not clearly explained the reasons for LPA choice. We apologize for it. Our decision for LPA investigation rather N-acylethanolamides was not mainly due for the potency differences rather for the analogy of this bioactive lipid with the capsaicin mediated effect on microglia current and on microvesicles shedding. Not less important is the evidence on the increased production and release of LPA in inflammatory conditions. Lastly, LPA was never been investigated as endovanilloid in the brain and we were interested in finding another inflammatory candidate able to modulate brain TRPV1.

REVIEWERS' COMMENTS:

Reviewer #1 (Remarks to the Author):

I thank the authors for their answers. The part of the manuscript that addresses the controversy regarding TRPV1 expression pattern remains weak: what explains that the reporter mice generated by Cavanaugh and colleagues, or the polyclonal antibodies, misrepresent TrpV1 expression in the CNS, in particular in glia? It is the authors' own interest to clarify this question, convince the field, and put the controversy to rest. The new experiments also highlight differences in DRG and spinal cord that remain unexplained and puzzling. The author's Mab labels about three times more NF200+ neurons compared to the reporter mouse. This might be important as these cells might correspond to heat - sensitive myelinated nociceptors; so far TrpV1 was thought to be largely restricted to unmyelinated nociceptors. In the spinal cord, the expression in ventral horn neurons is puzzling: does TrpV1 regulate spinal motor control, and are these Mab+ cells motoneurons? This could represent a major confound for pain studies in which withdrawal reflexes for noxious heat have been used to probe TrpV1 function; for example in TrpV1 KO mice, or in ablation studies in which high doses of capsaicin were injected intrathecally to ablate TrpV1+ central terminals (e.g. studies by Yaksh, Jessell et al PMID: 228392, Hoon et al. PMID: 19853036, Anderson et al. PMID: 19451647). Capsaicin is also injected in the peripheral tissue in pain models; expression in motoneurons could underlie a direct capsaicin action on muscle contraction at the neuromuscular junction. Do all these DRG and spinal cord neurons show capsaicin-induced currents? The new data provided by the authors may transform our understanding of TrpV1 function in spinal processing of somatosensory information; this is potentially extremely important and needs to be watertight. Note that CaMKII is not a useful marker of excitatory neurons in the dorsal horn; instead, TLX3 and NK1R should be used for excitatory interneurons and projection neurons, respectively.

Reviewer #2 (Remarks to the Author):

The authors have done an impressive amount of work. They have convincingly answered to all the concerns raised in the first versions of their manuscripts.

I still believe that such an enormous amount of data somehow hampers the reader to have a good synthetic view of their work, but I acknowledge their answer and understand that they follow the recommendations of the editor.

Responses to Reviewer#1

I thank the authors for their answers. The part of the manuscript that addresses the controversy regarding TRPV1 expression pattern remains weak:

1) what explains that the reporter mice generated by Cavanaugh and colleagues, or the polyclonal antibodies, misrepresent TrpV1 expression in the CNS, in particular in glia? It is the authors' own interest to clarify this question, convince the field, and put the controversy to rest.

Along with the Reviewer we have added new sentences and a new reference (both highlighted in blue) in the Discussion to strengthen the issue on the controversy of brain TRPV1 expression. In particular we have argued that the discrepancy between the conclusions of our study (and other evidence on the expression of TRPV1 in the brain), with those from the TRPV1 reporter mice studies might lie on the fact that reporter mice not always faithfully reproduce the expression pattern of the endogenous gene, as evinced by: i) the variable levels of protein expression downstream of IRES sequences; ii) the bigger size of mRNA reporter (encompassing TRPV1, PLAP, lacZ) compared the endogenous one (TRPV1 alone), with possible effects on mRNA stability and shorter half-life, with respect to that of the endogenous TRPV1 mRNA, as a result of a different secondary and tertiary structure, of a lack or different complement of 3'UTR sequences and/or to the presence of spurious microRNA target sequences; iii) possible chromatin organization alterations due to the larger size of the gene. Finally, we have discussed the different expression patterns between the MAb and the polyclonal antisera, in relation to the experimental technique/conditions and protein state (denaturated/undenaturated, fixed or unfixed). Although not reported in the discussion also a different subcellular microdomain in microglia and neurons may influence the protein distribution epitope conformation and the ensuing exposure to antibodies.

We hope this time to have provided constructive explanations that may convince on the existence of this channel also in the brain and in microglia cells, in physiological conditions. Certainly, we neither claim nor expect to have totally solved the controversy but we are confident that our study can give a solid and compelling insight on the role of TRPV1 in the brain.

2) The new experiments also highlight differences in DRG and spinal cord that remain unexplained and puzzling.

We consider these issues on sensory neurons and motoneurons to be extremely interesting, like all the other issues raised by the Reviewer in each revision. However we also think that the questions raised in this last report are too specific and not central for our manuscript, that is indeed centered on TRPV1 in the brain.

Along with the Reviewer, we are aware that the provided DRG and spinal cord data give not a whole understanding on TRPV1 distribution on these areas. Indeed these experiments have been performed, as requested, as positive "controls" for our evidence on TRPV1 in the brain. Nonetheless, we are

interested in addressing the issue on sensory and spinal neurons in a future dedicate study, even better in collaboration with experts of these areas.

a- The author's Mab labels about three times more NF200+ neurons compared to the reporter mouse. This might be important as these cells might correspond to heat-sensitive myelinated nociceptors; so far TrpV1 was thought to be largely restricted to unmyelinated nociceptors.

We thank the Reviewer for his valuable insights on our data. To this regard, **we have added a note on the legend of Supplementary Figure 9:**” Note that the NF200+ neurons might correspond to the heat-sensitive myelinated nociceptors (Ringkamp M, PMID: 11404433).

Our result indirectly provides another additional piece of evidence in support of the fact that the antibody detects in the brain cells that the reporter mouse fails to identify. In agreement with our evidence and that stemming from the reporter mice, a recent study by Dimitry Usoskin et al (Unbiased classification of sensory neuron types by large-scale single-cell RNA sequencing. Nature Neuroscience 2015) indicates that also a subtype of myelinated sensory neurons (i.e. NF4, belonging to proprioceptors subclass) express TRPV1 gene (see the heat map of fig 4 and supplementary table 6). So far, there is no functional evidence on vanilloid channels in proprioceptors. Therefore to confirm our evidence of TrpV1 expression in these sensory neurons, future studies are mandatory to test if TRPV1 is also functional in these cells; in particular if the same percentage of the NF200+/TRPV1+ myelinated sensory neurons also show capsaicin-induced currents.

b- In the spinal cord, the expression in ventral horn neurons is puzzling: does TrpV1 regulate spinal motor control, and **are these Mab+ cells motoneurons**? This could represent a major confound for pain studies in which withdrawal reflexes for noxious heat have been used to probe TrpV1 function; for example in TrpV1 KO mice, or in ablation studies in which high doses of capsaicin were injected intrathecally to ablate TrpV1+ central terminals (e.g. studies by Yaksh, Jessell et al PMID: 228392, Hoon et al. PMID: 19853036, Anderson et al. PMID: 19451647). Capsaicin is also injected in the peripheral tissue in pain models; expression in motoneurons could underlie a direct capsaicin action on muscle contraction at the neuromuscular junction.

I really thank the Reviewer for his/her comments that enhance my own critical understanding of the data. To my knowledge, there is no direct and functional evidence of TrpV1 regulation of spinal motor control. The only study on a likely expression of TrpV1 in motor neurons is in *Drosophila* (Ching-On Wong et al PMID: 25451193). In addition, as shown in the figure below from the Allen Brain Atlas, TrpV1 ISH assays reveal the expression of this channel also in the ventral horn of the spinal cord and presumably in the motoneurons (blue arrows in the figure taken from the Allen atlas of spinal cord <http://mousespinal.brainmap.org/imageseries/show.html?id=100018297>).

Hence, our IF data on ventral horn represent a preliminary result on a possible involvement of TRPV1 signaling in the nocifensive behavior. In addition, the TRPV1+ neurons in the ventral horn resemble motor neurons in terms of their shape.

In the legend of the new Supplementary Fig 8 we have added the following sentences: "Note that the TRPV1 expression on VH neurons may provide a preliminary evidence on a possible involvement of TRPV1 signaling in the nocifensive behavior".

- Do all these DRG and spinal cord neurons show capsaicin-induced currents? The new data provided by the authors may transform our understanding of TrpV1 function in spinal processing of somatosensory information; this is potentially extremely important and needs to be watertight.

We thank the Reviewer for his/her positive judgment on our new results and we agree with him/her that this pattern distribution of TRPV1 may add a new role of this channel in the somatosensory information processing. For example, in the proprioception information conveyed by the ventral spinocerebellar tract. However, we consider worthwhile to dedicate another study to this issue, since the present manuscript is on the role of TRPV1 in the brain, it has already an enormous amount of data, and lastly, the DRG and spinal cord represent a different story in their own right. We are confident that Reviewer will understand our point.

Note that CaMKII is not a useful marker of excitatory neurons in the dorsal horn; instead, TLX3 and NK1R should be used for excitatory interneurons and projection neurons, respectively.

Thank you for this advice. We will employ these markers for the future investigations. Our antibody choice has been based on the paper by Kim YH et al (PMID- 22632722) for the GAD antibody, while for the CaMK glutamatergic marker because it was already present in our lab.